# Small random initialization is akin to spectral learning: Optimization and generalization guarantees for overparameterized low-rank matrix reconstruction

**Dominik Stöger**
Katholische Universität Eichstätt-Ingolstadt
85072 Eichstätt, Germany
`Dominik.Stoeger@ku.de`

**Mahdi Soltanolkotabi**
University of Southern California
Los Angeles, CA 90089
`soltanol@usc.edu`

## Abstract

Recently there has been significant theoretical progress on understanding the convergence and generalization of gradient-based methods on nonconvex losses with overparameterized models. Nevertheless, many aspects of optimization and generalization and in particular the critical role of small random initialization are not fully understood. In this paper, we take a step towards demystifying this role by proving that small random initialization followed by a few iterations of gradient descent behaves akin to popular spectral methods. We also show that this *implicit spectral bias* from small random initialization, which is provably more prominent for overparameterized models, also puts the gradient descent iterations on a particular trajectory towards solutions that are not only globally optimal but also generalize well. Concretely, we focus on the problem of reconstructing a low-rank matrix from a few measurements via a natural nonconvex formulation. In this setting, we show that the trajectory of the gradient descent iterations from small random initialization can be approximately decomposed into three phases: (I) a *spectral or alignment phase* where we show that that the iterates have an implicit spectral bias akin to spectral initialization allowing us to show that at the end of this phase the column space of the iterates and the underlying low-rank matrix are sufficiently aligned, (II) a *saddle avoidance/refinement phase* where we show that the trajectory of the gradient iterates moves away from certain degenerate saddle points, and (III) a *local refinement phase* where we show that after avoiding the saddles the iterates converge quickly to the underlying low-rank matrix. Underlying our analysis are insights for the analysis of overparameterized nonconvex optimization schemes that may have implications for computational problems beyond low-rank reconstruction.

## 1 Introduction

Many contemporary problems in machine learning and signal estimation spanning deep learning to low-rank matrix reconstruction involve fitting nonlinear models to training data. Despite tremendous empirical progress, theoretical understanding of these problems poses two fundamental challenges. First, from an *optimization* perspective, fitting these models often requires solving highly nonconvex optimization problems and except for a few special cases, it is not known how to provably find globally or approximately optimal solutions. Yet simple heuristics such as running (stochastic) gradient descent from (typically) small random initialization is surprisingly effective at finding globally optimal solutions. A second *generalization* challenge is that many modern learning models including neural network architectures are trained in an overparameterized regime where the parameters of the model exceed the size of the training dataset. It is well understood that in this overparameterized

35th Conference on Neural Information Processing Systems (NeurIPS 2021).

regime, these large models are highly expressive and have the capacity to (over)fit arbitrary training datasets including pure noise. Mysteriously however overparameterized models trained via simple algorithms such as (stochastic) gradient descent when initialized at random continue to predict well or generalize on yet unseen test data. In particular, it has been noted in a number of works that for many modern machine learning architectures, the scale of initialization is important for the generalization/test behavior [1, 2]. It has been noted that stronger generalization performance is typically observed for a smaller scale initialization. Indeed, small random initialization followed by (stochastic) gradient descent iterative updates is arguably the most widely used learning algorithm in modern machine learning and signal estimation.

There has been a large number of exciting results aimed at demystifying both the optimization and generalization aspects over the past few years. We will elaborate on these results in detail in the supplementary, however, we would like to briefly mention the common techniques and their existing limitations. On the optimization front a large body of work has emerged on providing guarantees for nonconvex optimization which can roughly be put into two categories: (I) smart initialization+local convergence and (II) landscape analysis+saddle escaping algorithms. Approaches in (I) focus on showing local convergence of local search techniques from carefully designed spectral initializations [3, 4, 5, 6, 7, 8, 9, 10]. Approaches in (II) focus on showing that in some cases the optimization landscape is benign in the sense that all local minima are global (no spurious local minima) and the saddle points have a direction of strict negative curvature (strict saddle) [11]. Then specialized truncation or saddle escaping algorithms such as trust region, cubic regularization [12, 13], or noisy (stochastic) gradient-based methods [14, 15, 16, 17] are deployed to provably find a global optimum. Both approaches fail to fully explain the typical behavior of local search techniques in practice. Indeed, for many nonconvex problems local search techniques or simple variants, when initialized at random, quickly converge to globally optimal solutions without getting stuck in local optima/saddles without the need for sophisticated initialization or saddle escaping heuristics. We note that while for differentiable losses eventual convergence to local minimizers is known from a random initialization [18] on problems of the form (II), these results cannot rule out exponentially slow cases in the worst-case [19]. Indeed, it has been argued that in general a more granular analysis of the trajectory of gradient descent beyond the landscape may be necessary [20]. For example, some recent advances has been made by analysing the trajectory of gradient descent using a leave-one-out analysis for the phase retrieval problem [21].

Similarly, there has been a lot of exciting progress on the generalization front, especially for neural networks. Specific to generalization capabilities of gradient-based approaches these results broadly fall into two categories: (a) the first category is based on a linearization principle which characterizes the performance of nonlinear models such as neural networks by comparing it to a linearized kernel problem around the initialization (a.k.a. Neural Tangent Kernels) [22, 23, 24, 25, 26, 27, 28]. This has often been referred to as "lazy training". (b) the second category is based on a continuous limit analysis in the limit of width going to infinity and learning rate going to zero (mean-field analysis) [29, 30, 31, 32, 33]. However, these existing analyses contain many idealized and non-realistic assumptions (e.g. requiring large, random initialization in (a), which typically leads to worse generalization than what is observed in practice, or unrealistically large widths in (b)) and therefore cannot fully explain the success of overparameterized models or serve as a guiding principle for practitioners [34].

Despite the aforementioned exciting recent theoretical progress many aspects of optimization and generalization and in particular the role of random initialization remains mysterious. This leads us to the main challenge of this paper

> *Why is **small random initialization** combined with gradient descent updates so effective at finding globally optimal models that generalize well despite the nonconvex nature of the optimization landscape or model overparameterization?*

In this paper we wish to take a step towards addressing the above challenge by demystifying the critical role of small random initialization in gradient-based approaches. Specifically we show that

> ***Small random initialization** followed by a few iterations of gradient descent behaves akin to spectral initialization.*

By that, we mean more precisely, that if the initialization is chosen small enough, then in the initial stage of the training, gradient descent *implicitly* behaves like spectral initialization techniques such as

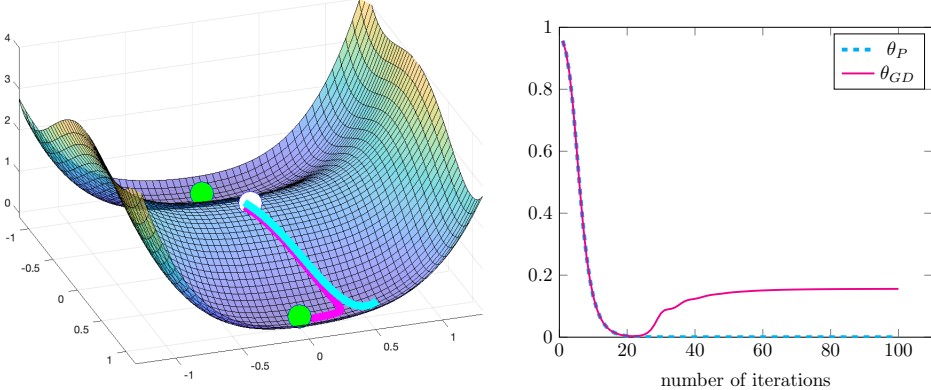

Figure 1: **Gradient descent from small random initialization is akin to spectral initialization.** The left figure depicts the empirical landscape of a low-rank matrix reconstruction problem with the two green circles depicting the two global minima and the white circle the saddle point at the origin. In this figure, we also depict the trajectory of the gradient descent iterations (magenta) together with the power method based on a popular spectral initialization technique (blue). Both gradient descent and power method use the same small initialization near the origin. We see that in the early stage, the two trajectories are almost the same. The figure on the right depicts the angle between the gradient descent (magenta)/power method (blue) iterates and a popular spectral initialization technique, denoted by $\theta_{GD}$ and $\theta_P$ respectively. This figure clearly demonstrates that for the first iterations these angles are practically the same further confirming that the initial trajectory of gradient descent and power methods are similar. See Section 5 for further detail on the experimental setup. (In this figure we have used $r = r_\star = 1$.)

those commonly used in techniques based on the method of moments. This *implicit spectral bias* of gradient descent from random initialization puts the gradient descent iterations on a particular trajectory towards solutions that are not only globally optimal but also generalize well for overparameterized models. We also show that with small random initialization this implicit spectral bias phenomenon is more prominent for more overparameterized models in the sense that it materializes after fewer iterations. This intriguing phenomenon is depicted in Figure 1 in the context of a low-rank reconstruction problem. This figure clearly demonstrates that the first few iterations of gradient descent starting from a small random initialization are virtually identical to that of running power iterations (a popular algorithm to find the spectral initialization, see, e.g. [35]).

Concretely we focus on the problem of low-rank matrix recovery, which appears in many different application areas such as recommendation systems, phase retrieval, and quantum tomography [36]. Here, our goal is to recover a low-rank matrix of the form $XX^T$ from a few linear measurements. We consider a natural, non-convex approach based on matrix factorization, where we minimize the loss function via gradient descent. In this paper, we show that, regardless of the amount of overparameterization used, for small random initialization vanilla gradient descent will always converge towards the low-rank solution. This holds as long as the measurement operator obeys a popular restricted isometry property [37].

Our analysis consists of three phases. The first phase is the aforementioned *spectral* or *alignment* phase where we show gradient descent from small random initialization behaves akin to spectral initialization, which is a key insight of this paper. Indeed, we show that the first few gradient descent iterates can be accurately approximated by power method iterates. Next, we show that after this first *spectral or alignment phase*, gradient descent enters a second phase, which we refer to as *saddle avoidance phase*. In this phase, we show that the trajectory of the gradient iterates moves away from degenerate saddle points, while the iterates maintain almost the same effective rank as $XX^T$. In the third phase, the *local refinement phase*, we show that the iterates approximately converge towards the underlying low-rank matrix $XX^T$ with a geometric rate up to a certain error floor which depends on the initialization scale. In particular, by decreasing the scale of initialization this error threshold can

be made arbitrarily small. While in this paper our main focus is on low-rank matrix reconstruction, we believe that our analysis holds more generally for a variety of contemporary machine learning and signal estimation tasks including neural networks.

Finally we note that while a similar setting has already been studied in [38], our analysis goes beyond it in many important ways. For example, our result holds for any amount of overparameterization and allows for arbitrarily small initialization. Maybe most importantly, we study the spectral phase phenomenon at initialization.

## 2 Low-rank matrix recovery via non-convex optimization

As mentioned earlier in this paper we focus on reconstructing a (possibly overparameterized) Positive Semidefinite (PSD) low rank matrix from a few measurements. In this problem, given $m$ observations of the form

$$y_i = \langle A_i, XX^T \rangle = \text{Tr}\left(A_i XX^T\right) \qquad i = 1, \ldots, m, \tag{1}$$

we wish to reconstruct the unknown matrix $XX^T$. Here, $X \in \mathbb{R}^{n \times r_\star}$ with $1 \le r_\star \le n$ is a factor of the unknown matrix and $\{A_i\}_{i=1}^m$ are known symmetric measurement matrices. A common approach to solving this problem is via minimizing the loss function

$$\min_{\bar{U} \in \mathbb{R}^{n \times r}} f(\bar{U}) := \min_{\bar{U} \in \mathbb{R}^{n \times r}} \frac{1}{4m} \sum_{i=1}^m \left(y_i - \langle A_i, \bar{U}\bar{U}^T \rangle\right)^2,$$

with $r \ge r_\star$. More compactly one can rewrite the optimization problem above in the form

$$\min_{\bar{U} \in \mathbb{R}^{n \times r}} f(\bar{U}) := \min_{\bar{U} \in \mathbb{R}^{n \times r}} \frac{1}{4} \left\| \mathcal{A}\left(\bar{U}\bar{U}^T - XX^T\right) \right\|_{\ell_2}^2, \tag{2}$$

where $\mathcal{A} : \mathbb{R}^{n \times n} \longrightarrow \mathbb{R}^m$ is the measurement operator defined by $[\mathcal{A}(Z)]_i := \frac{1}{\sqrt{m}} \langle A_i, Z \rangle$.

In order to solve the minimization problem (2) we run gradient descent iterations starting from (often small) random initialization. More specifically,

$$U_{t+1} = U_t - \mu \nabla f(U_t) = U_t + \mu \mathcal{A}^* \left[y - \mathcal{A}\left(U_t U_t^T\right)\right] U_t$$
$$= U_t + \mu \left[(\mathcal{A}^* \mathcal{A})\left(XX^T - U_t U_t^T\right)\right] U_t.$$

where we have set $U_0 = \alpha U$ is the initialization matrix, $\mathcal{A}^*$ denotes the adjoint operator of $\mathcal{A}$ and $y = (y_i)_{i=1}^m \in \mathbb{R}^m$ denotes the measurement vector. Here, $U \in \mathbb{R}^{n \times r}$ is a typically random matrix which represents the form of the initialization and $\alpha > 0$ is a scaling parameter.

There are two challenges associated with analyzing such randomly initialized gradient descent updates. The first is an *optimization* challenge. Since $f$ is *non-convex* it is a priori not clear whether gradient descent converges to a global optimum or whether it gets stuck in a local minima and/or saddle. The second challenge is that of *generalization*. This is particularly pronounced in the overparameterized scenario where the number of parameters are larger than the number of data points i.e. $rn \ge m$. In this case, there are infinitely many $\bar{U}$ such that $f(\bar{U}) = 0$, but $\|\bar{U}\bar{U}^T - XX^T\|_F$ is arbitrarily large (see, e.g., [39, Proposition 1]). That is, even if gradient descent converges to a global optimum, i.e. $f(\bar{U}) = 0$, it is a priori not clear whether it has found the low-rank solution $XX^T$ (see also Figure 5).

## 3 Main results

In this section, we present our main results. Stating these results requires a couple of simple definitions. The first definition concerns the measurement operator $\mathcal{A}$.

**Definition 3.1** (Restricted Isometry Property (RIP)). *The measurement operator $\mathcal{A} : \mathbb{R}^{n \times n} \longrightarrow \mathbb{R}^m$ satisfies RIP of rank $r$ with constant $\delta > 0$, if it holds for all matrices $Z$ of rank at most $r$*

$$(1 - \delta) \|Z\|_F^2 \le \|\mathcal{A}(Z)\|_{\ell_2}^2 \le (1 + \delta) \|Z\|_F^2. \tag{3}$$

We note that for a Gaussian measurement operator $\mathcal{A}$ [1], RIP of rank $r$ and constant $\delta > 0$ holds with high probability, if the number of observations satisfies $m \gtrsim nr/\delta^2$ [37, 40].

---

[1]By that, we mean that all the entries of the (symmetric) measurement matrices $\{A_i\}_{i=1}^m$ are drawn i.i.d. with distribution $\mathcal{N}(0, 1)$ on the off-diagonal and distribution $\mathcal{N}(0, 1/\sqrt{2})$ on the diagonal.

The second definition concerns the condition number of the factor $X$.

**Definition 3.2** (condition number). *We denote the condition number of $X \in \mathbb{R}^{n \times r_\star}$ by $\kappa := \frac{\|X\|}{\sigma_{r_\star}(X)}$, where $\sigma_{r_\star}(X)$ denotes $r_\star$-th largest singular value of $X$.*

With these definitions in place we are now ready to state our main results. Due to space limitations in the main paper we focus on the case of $r \geq 2r_\star$. With refer the reader to the supplementary for results covering all $r \geq r_\star$ including two special cases: (1) the fully overparameterized case, i.e., $r = n$ along with comparisons with existing work, in this case [38], and (2) the scenario that $U$ has the same number of parameters as $X$, i.e., $r = r_\star$.

**Theorem 3.3.** *Let $X \in \mathbb{R}^{n \times r_\star}$ and assume we have $m$ measurements of the low rank matrix $XX^T$ of the form $y = \mathcal{A}(XX^T)$ with $\mathcal{A}$ the measurement operator. We assume $\mathcal{A}$ satisfies the restricted isometry property for all matrices of rank at most $2r_\star + 1$ with constant $\delta \leq c\kappa^{-4}r_\star^{-1/2}$. To reconstruct $XX^T$ from the measurements we fit a model of the form $\bar{U} \mapsto \mathcal{A}(\bar{U}\bar{U}^T)$ with $\bar{U} \in \mathbb{R}^{n \times r}$ via running gradient descent iterations of the form $U_{t+1} = U_t - \mu\nabla f(U_t)$ on the objective (2) with a step size obeying $\mu \leq c\kappa^{-4}\|X\|^{-2}$. Here, the initialization is given by $U_0 = \alpha U$, where $U \in \mathbb{R}^{n \times r}$ has i.i.d. entries distributed as $\mathcal{N}(0, 1/\sqrt{r})$. Furthermore, we assume $r \geq 2r_\star$ and that the scale of initialization fulfills*

$$\alpha \lesssim \min\left\{\frac{(\min\{r; n\})^{1/4}}{\kappa^{1/2}n^{3/4}}\left(2\kappa^2\sqrt{\frac{n}{\min\{r; n\}}}\right)^{-6\kappa^2}; \frac{1}{\kappa^7 n}\right\}\|X\|. \tag{4}$$

*Then, after*

$$\hat{t} \lesssim \frac{1}{\mu\sigma_{\min}(X)^2}\ln\left(\frac{C_1 n\kappa}{\min\{r; n\}} \cdot \max\left\{1; \frac{\kappa r_\star}{\min\{r; n\} - r_\star}\right\} \cdot \frac{\|X\|}{\alpha}\right)$$

*iterations we have that*

$$\frac{\|U_{\hat{t}}U_{\hat{t}}^T - XX^T\|_F}{\|X\|^2} \lesssim \frac{n^2\kappa^{81/16}r_\star^{1/8}}{(\min\{r; n\})^{15/16}} \cdot \frac{\alpha^{21/16}}{\|X\|^{21/16}}, \tag{5}$$

*holds with probability at least $1 - Ce^{-\tilde{c}r}$. Here, $c, \tilde{c}, C, C_1 > 0$ are fixed numerical constants.*

Note that the test error $\|U_{\hat{t}}U_{\hat{t}}^T - XX^T\|_F^2$ can be made arbitrarily small by choosing the scale of initialization $\alpha$ small enough. In particular, the dependence of the test error on $\alpha$ is polynomial and the dependence of the number of iterations on $\alpha$ is logarithmic, which means that reducing the test error by scaling down $\alpha$ introduces only modest additional computational cost. Hence, as long as the rank at most $2r_\star + 1$ RIP with constant $\delta \leq c\kappa^{-4}r_\star^{-1/2}$ holds, gradient descent converges to a point in the proximity of the low-rank solution, whenever the initialization is chosen small enough regardless of the choice of $r$. This holds even when the model is overparameterized i.e. $rn \gg m$ and the optimization problem has many global optima many of which do not obey $UU^T \approx XX^T$. This result thus further demonstrates that when initialized with a small random initialization gradient descent has an implicit bias towards solutions of low-rank or small nuclear norm. This is in sharp contrast to Neural Tangent Kernel (NTK)-based theory for low-rank matrix recovery (see [23, Section 4.2]) which will not approximately recover the ground truth matrix $XX^T$ due to the larger scale of initialization required when using that technique.

As discussed in Section 2, the restricted isometry property holds with high probability for a sample complexity $m \gtrsim nr_\star^2\kappa^8$ for Gaussian measurement matrices. Up to constants, this sample complexity is optimal in $n$, while it is sub-optimal in $r_\star$ and $\kappa$ compared to approaches based on nuclear-norm minimization (see, e.g., [37]). While there is numerical evidence that the true scaling of $m$ in $r_\star$ should also be linear in the non-convex case [41], we note that the optimal dependence of the sample complexity on $r_\star$ is a major open problem in the field, as the sample complexities in all theoretical results for non-convex approaches in the literature scale at least quadratically in $r_\star$.

**Interpretation:** Recall from Section 1 that our convergence analysis can be divided into three phases: the spectral phase, the saddle avoidance phase, and the local refinement phase. As it will become clear from the proofs in the supplementary when $r \geq 2r_\star$ the bound on the number of iterations can

be decomposed as follows

$$\hat{t} \lesssim \frac{1}{\mu\sigma_{\min}(X)^2}\left[\underbrace{\ln\left(2\kappa^2\sqrt{\frac{n}{\min\{r;n\}}}\right)}_{\text{Phase I: spectral/alignment phase}} + \underbrace{\ln\left(\frac{\sigma_{\min}(X)}{\alpha}\right)}_{\text{Phase II: saddle avoidance phase}} + \underbrace{\ln\left(\max\left\{1;\frac{\kappa r_\star}{\min\{r;n\}-r_\star}\right\}\frac{\|X\|}{\alpha}\right)}_{\text{Phase III: local refinement phase}}\right].$$

(6)

First, we note that the duration of all three phases scales inversely with $\sigma_{\min}(X)^2$. This is due to the fact that in all three phases the dynamics associated the smallest singular value of $X$ is the slowest one and hence needs the most time to complete.

In the spectral phase, the eigenvectors corresponding to the leading $r_\star$ eigenvalues of $U_t U_t^T$ become aligned with the eigenvectors corresponding to the leading $r_\star$ eigenvalues of $\mathcal{A}^*\mathcal{A}(XX^T)$. We observe in (6) that in the spectral phase increasing $r$, i.e. the amount of parameters, decreases the number of iterations in this phase. As we will explain in the supplementary, the reason is that increasing $r$ decreases the angle between the column space of the initialization $U_0$ and the span of the eigenvectors corresponding to the leading $r_\star$ eigenvalues of $\mathcal{A}^*\mathcal{A}(XX^T)$ used in spectral initialization. As a consequence, gradient descent needs fewer iterations to align these two subspaces.

In the saddle avoidance phase (Phase II), $\sigma_{r_\star}(U_t)$, the $r_\star$th largest singular value of $U_t$, grows geometrically until it is on the order of $\sigma_{\min}(X)$. Hence, this duration depends on the ratio between the $\sigma_{\min}(X)$ and the the scale of initialization $\alpha$. This is clearly reflected in the upper bound on the number of needed iterations in equation (6).

In Phase III, the local refinement phase, the matrix $U_t U_t^T$ converges towards $XX^T$. In particular, at iteration $\hat{t}$ the test error obeys (5). We observe that a smaller $\alpha$ allows for a smaller test error in (5) but per (6) this higher accuracy is achieved with a modest increase in the required iterations.

## 4 A glimpse of our analysis

In our proofs, we show that the trajectory of the gradient descent iterations can be approximately decomposed into three phases: (I) a *spectral or alignment phase* where we show that gradient descent from random initialization behaves akin to spectral initialization allowing us to show that at the end of this phase the column spaces of the iterates $U_t$ and the ground truth matrix $X$ are sufficiently aligned, (II) a *saddle avoidance phase*, where we show that the trajectory of the gradient iterates move away from certain degenerate saddle points, and (III) a *refinement phase*, where the product of the gradient descent iterates $U_t U_t^T$ converges quickly to the underlying low-rank matrix $XX^T$. The latter result holds up to a small error that is commensurate with the scale of the initialization and tends to zero as the scale of the initialization goes to zero.

To formalize the above, we use $L$ and $L_t$ to denote the subspaces spanned by the eigenvectors corresponding to the $r_\star$ largest eigenvalues of the matrix $\mathcal{A}^*\mathcal{A}(XX^T)$, and $U_t U_t^T$, respectively. Moreover, for a subspace $L$ of dimension $r_\star$ we use $V_L \in \mathbb{R}^{n\times r_\star}$ to denote an orthonormal matrix whose columns span the subspace $L$. Note that $L$ is the subspace, which is obtained by commonly used spectral methods. Using this notation, Figure 2a depicts the three phases described above.

In the spectral phase we will prove that we can approximate the iterate $U_t$ by

$$U_t \approx \underbrace{\left(\text{Id} + \mu\mathcal{A}^*\mathcal{A}(XX^T)\right)^t}_{=:Z_t} U_0 = Z_1^t U_0 := \tilde{U}_t. \tag{7}$$

We note that the matrix $Z_1 = \text{Id} + \mu\mathcal{A}^*\mathcal{A}(XX^T)$ is the basis for the commonly used spectral initialization, where typically a factorization of the rank $r_\star$ approximation of this matrix is used as the initialization [6, 5, 42]. Therefore, the approximation (7) suggests that gradient descent iterates modulo the normalization are akin to running power method on $Z_1$. Hence, we expect that at the end of the spectral phase the subspace $L_t$ to be closely aligned with the subspace $L$, i.e. the subspace obtained by commonly used spectral initialization techniques. In particular, this also implies that $L_t$ is also aligned with the subspace $X$. Figure 2b clearly illustrates that the first few iterations of gradient descent behave essentially identical to the power method, confirming our intuition.

The description of the second and third phase is more elaborate and technical in nature and we defer to the supplementary for a more detailed and intuitive explanation. However, to give a brief

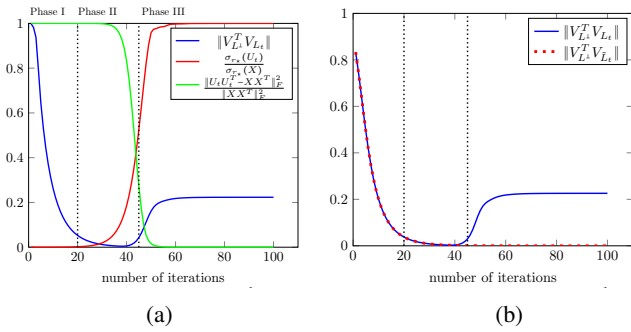

(a)  (b)

Figure 2: **(a) Depiction of the three phases of convergence.** This figure demonstrates that the convergence analysis can be divided into three phases: (I) spectral/alignment phase; (II) saddle avoidance phase and (III) the refinement phase. We see that in the first phase the first $r_\star$ eigenvectors of $U_t U_t^T$ rapidly learn the subspace corresponding to the first $r_\star$ eigenvectors of $\mathcal{A}^* \mathcal{A} (XX^T)$, i.e. the angle $\|V_{L^\perp}^T V_{L_t}\|$ becomes small. The $r_\star$th largest singular value of $U_t$ is still small in this phase and the (normalized) test error $\|U_t U_t - XX^T\|_F^2 / \|XX^T\|_F^2$ has not decreased yet. In Phase (II), however, we see that $\sigma_{r_\star}(U_t)$ is growing, whereas the loss begins to decrease in this phase and the subspaces stay aligned. In Phase (III) we see that the test error is converging towards 0 rapidly, meaning that $U_t U_t^T$ converges to $XX^T$. Consequently, $\sigma_{r_\star}(U_t) / \sigma_{r_\star}(X)$ converges to 1 (red curve). We also see that in this phase the angle $\|V_{L^\perp}^T V_{L_t}\|$ grows again, until it reaches a certain threshold. This is because in this phase the top $r_\star$ eigenvalues of $U_t U_t^T$ become aligned with the eigenvectors of $XX^T$. **(b) Depiction of the spectral alignment phase: in the first few iterations, gradient descent with small initialization behaves like a power method.** Denote by $\tilde{L}_t$ the subspace spanned by the eigenvectors corresponding to the $r_\star$ largest eigenvalues of the matrix $\widetilde{U}_t \widetilde{U}_t^T$. Analogously as before denote by $V_{\tilde{L}_t}$ an orthonormal matrix, whose columns span the subspace $\tilde{L}_t$. In this figure, we observe that in the first iterations $U_t$ and $\widetilde{U}_t$ learn the subspace $L$ at almost exactly the same rate.

description, in these two phases we will decompose the iterates $U_t$ into the sum of two matrices, a "signal" matrix of rank $r_\star$ and a "noise" matrix of rank at most $r - r_\star$. In Phase (II) we will prove that the smallest singular value of the signal term, which is approximately the same as $\sigma_{r_\star}(U_t)$ grows, whereas the spectral norm of the noise matrix grows at a much slower rate. We will also show that in this phase the columns of the signal term stay approximately aligned with the span of the matrix $X$. As soon as the smallest singular value of the signal term of $U_t$ is approximately at the same order than the smallest singular value of $X$ we enter Phase (III). In that Phase we provide a local convergence argument, which shows that the signal term of $U_t$ converges towards $X$ (up to a rotation), whereas the noise term stays small.

## 5 Numerical experiments

In this section, we perform several numerical experiments to corroborate our theoretical results.

**Experimental setup.** For the experiments we set the ground truth matrix $X \in \mathbb{R}^{n \times r_\star}$ to be a random orthogonal matrix with $n = 200$ and $r_\star = 5$. Moreover, we use $m = 10nr_\star = 50n$ random Gaussian measurements. The initialization $U$ is chosen as in Theorem 3.3 and we use a step size of $\mu = 1/4$ which is consistent with these theorems. We note that while all experimental depictions are based on a single trial, in line with the NeurIPS guidelines we have drawn these curves multiple times (not depicted) and the behavior of the plots do not change.

**Depiction of the three phases and the role of overparameterization.** In our first experiment, we want to examine how increasing the number of parameters via increasing the number of the columns $r$ of the matrix $U_t \in \mathbb{R}^{n \times r}$, affects the spectral phase. To this aim we set the scale of initialization to $\alpha = 1/(70n^2)$. Let $L$ denote the subspace spanned by the eigenvectors corresponding to the leading $r_\star$ singular values of $\mathcal{A}^* \mathcal{A}(XX^T)$ and $L_t$ denotes the subspace spanned by the left-singular vectors corresponding to the largest $r_\star$ singular values of $U_t$.

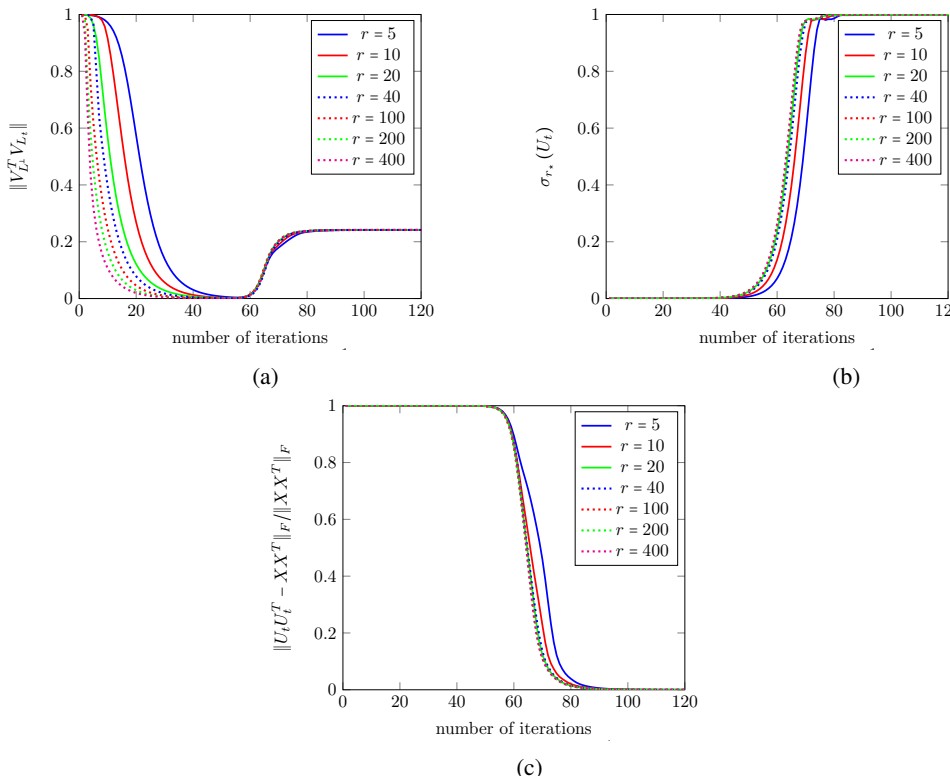

Figure 3: Impact of different levels of overparameterization on (a) the angle $\|V_{L^\perp}^T V_{L_t}\|$ and (b) the $r_\star$th largest singular value, (c) the trajectory of the (normalized) test error $\|U_t U_t^T - XX^T\|_F / \|XX^T\|_F$.

*Spectral phase and alignment under different levels of overparameterization.* First, we examine how the angle between these two subspaces (i.e. $\|V_{L^\perp}^T V_{L_t}\|$) changes in the first few iterations. We depict the results for different $r$ in Figure 3a. We see that in the first few iterations, i.e. in the spectral phase, this angle converges towards zero. This confirms the main conclusion of this paper that the first few iterations of gradient descent from small random initialization indeed behaves akin to running power method for spectral initialization. This experiment also shows that changing the number of columns $r$ of $U_t$ has an interesting effect on the spectral phase. In particular, increasing $r$ allows the gradient descent algorithm to learn the subspace $L$ with fewer iterations, i.e. $\|V_{L^\perp}^T V_{L_t}\|$ becomes small with fewer iterations. This is in accordance with our theory for $r_\star \leq r \leq n$ (see, for example, the first summand on the right-hand side of equation (6)), where we show that more overparameterization allows gradient descent to leave the spectral phase earlier. Interestingly, this improvement continues to hold even when increasing $r$ beyond $n$ allowing for even faster convergence of $\|V_{L^\perp}^T V_{L_t}\|$. This holds even though in this case the rank of $U_0$ is still not larger than $n$. One potential explanation for this phenomenon might be that for such a choice of $r$ the matrix $U_0$ is better conditioned.

*Growth of $\sigma_{r_\star}(U_t)$ and saddle avoidance.* In Figure 3b we depict how $\sigma_{r_\star}(U_t)$ grows during the training for different choices of $r$. We see that the curves look similar, although for smaller $r$ the growth phase sets in at a slightly later time. This is due to the fact that for smaller $r$, as we have seen in Figure 3a, Phase I, the spectral phase takes longer to complete.

*Evolution of the test error and the refinement phase.* Similarly, in Figure 3c we depict how the (normalized) test error $\|U_t U_t^T - XX^T\|_F / \|XX^T\|_F$ evolves during the training for different choices of $r$. We observe that for smaller $r$ the third phase sets in slightly later. Again, this is due to the fact that for smaller $r$ the spectral phase takes slightly longer to complete (see inequality (6)).

**Test error under different scales of initialization.** In the next experiment, we focus on understanding how the scale of initialization $\alpha$ affects the generalization error $\|U_t U_t^T - XX^T\|_F^2$.

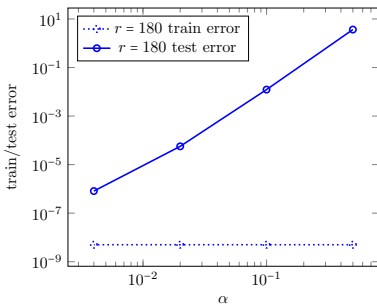

Figure 4: Relative test error $\frac{\|U_t U_t^T - XX^T\|_F}{\|XX^T\|_F}$ for different scales of initialization $\alpha$.

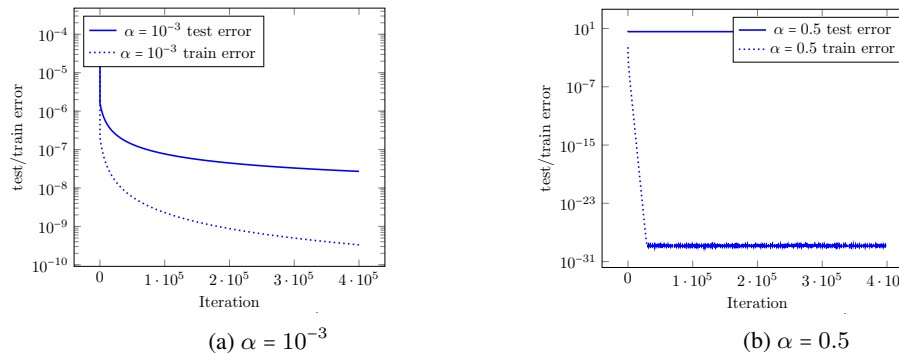

(a) $\alpha = 10^{-3}$        (b) $\alpha = 0.5$

Figure 5: Change of test error $\|U_t U_t^T - XX^T\|_F^2$ and train error $f(U_t)$ for (a) small and (b) large $\alpha$ during training.

For that, we set $r = 180$ and run gradient descent with for different choices of $\alpha$. We stop as soon as the training error becomes small ($f(U_t) \leq 0.5 \cdot 10^{-9}$). We depict the results in Figure 4. We see that the test error decreases as $\alpha$ decreases. In particular, this figure indicates that the test error depends polynomially on the scale of initialization $\alpha$. This is in line with our theory, where we also show that the test error decreases at least with the rate $\alpha^{21/16}$ (see inequality (5) in Theorem 3.3).

**Change of test and train error during training.** In the next experiment, we set $r = 180$ and examine how the test error $\|U_t U_t^T - XX^T\|_F^2$ and the train error $f(U_t)$ changes throughout training and, in particular, how this depends on the scale of initialization. To this aim, we run gradient descent with $4 \cdot 10^5$ iterations. We see that for a small scale of initialization, $\alpha = 10^{-3}$, which is the scenario studied in this paper, both test error and train error decrease throughout training.

We observe that in the beginning, as described our theory, both test and train error decrease rapidly. After that the decrease of both test and train error slows down significantly. Moreover, the train error converges towards zero, in contrast to the test error. One reason for the slow convergence in this phase might be that $U_t$ is ill-conditioned in the sense that $\sigma_{r_\star}(U_t W_t)$ is much larger than $\|U_t W_{t,\perp}\|$. It is an interesting future research direction to extend our theory to this part of the training.

For large scale of initialization $\alpha = 0.5$, we observe a very different behaviour. We see that the train error converges with linear rate until machine precision is reached. However, the test error barely changes throughout the training. This scale of initialization corresponds to the *lazy training regime* [34], where the parameters stay close to the initialization during the training. We depict the results in Figure 5.

**Number of iterations until convergence:** In the last experiment, we set $\alpha = 10^{-3}$ and examine how many iterations are needed until the test error $\|U_t U_t^T - XX^T\|_F^2$ falls below a certain threshold of $10^{-4}$ for different values of $r$ obeying $5 \leq r \leq 30$. For each choice of $r$ we run the experiment ten times and then average the number of iterations for each choice of $r$. The results are depicted in Figure 6. We observe that increasing the number of columns $r$ from 5 to 10, i.e., a small amount of overparameterization, decreases the number of iterations needed. After that the number of iterations

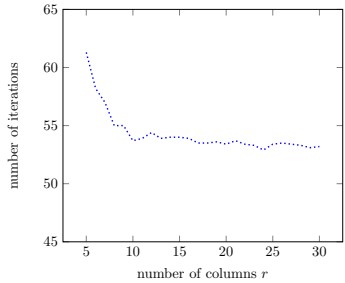

Figure 6: Number of iterations required for the test error to fall below $10^{-4}$ for different levels of overparameterization.

needed stays roughly constant. This observation is in line with Figure 3, where we have seen that overparameterization leads to fast decrease of the test error in the spectral phase (with diminishing speedup as $r$ becomes larger and larger) without affecting the other two phases.

## 6 Conclusion and Broader Impact

In this paper we focused on demystifying the role of initialization when training overparameterized models by showing that small random initialization followed by a few iterations of gradient descent behaves akin to popular spectral methods. We also show that this *implicit spectral bias* from small random initialization, which is provably more prominent for overparameterized models, also puts the gradient descent iterations on a particular trajectory towards solutions that are not only globally optimal but also generalize well.

We think that our results give rise to a number of interesting future research directions. For example, one could extend our results to scenarios where the measurement matrices are more structured such as in matrix completion [43] or in blind deconvolution [44]. Moreover, while our main results, e.g. Theorem 3.3 do require early stopping, our simulations (e.g. Figure 5a) indicate that early stopping is not needed. It would be interesting to examine whether we can remove the early stopping requirement. It is also an interesting future avenue to examine whether the quadratic dependence of the sample complexity $m$ on $r_\star$ in our results is really needed.

Moreover, while in this paper our main focus was on low-rank matrix reconstruction, we believe that our analysis holds more generally for a variety of contemporary overparameterized machine learning and signal estimation tasks including neural network training. This is a tantalizing future research direction.

Despite being theoretical/foundational in nature our results have potential for broader practical impact. In particular, low rank reconstruction problems are an important component of many recommender engines and our insights may guide better algorithm and systems designs for such engines. More broadly, training overparameterized models using stochastic GD starting from small random initialization is the work-horse of modern learning ncluding deep learning and our insights may in the long term help enable more efficient/reliable training with a smaller carbon footprint and improved test accuracy. As with other technologies such insights may potentially also be used nefariously.

## Acknowledgments and Disclosure of Funding

M.S. is supported by the Packard Fellowship in Science and Engineering, a Sloan Research Fellowship in Mathematics, an NSF-CAREER under award #1846369, the Air Force Office of Scientific Research Young Investigator Program (AFOSR-YIP) under award #FA9550-18-1-0078, DARPA Learning with Less Labels (LwLL) and FastNICS programs, and NSF-CIF awards #1813877 and #2008443.

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
