with distribution $\mathcal{N}(0, 1/\sqrt{r})$. With this setting and assumptions the following two statements hold.*

1. *Under the assumption that $r \ge 2r_\star$ and that the scale of initialization fulfills*

$$\alpha \lesssim \min\left\{ \frac{(\min\{r; n\})^{1/4}}{\kappa^{1/2}n^{3/4}} \left(2\kappa^2 \sqrt{\frac{n}{\min\{r; n\}}}\right)^{-6\kappa^2}; \frac{1}{\kappa^7 n} \right\} \|X\|, \tag{4}$$

   *after*

$$\hat{t} \lesssim \frac{1}{\mu\sigma_{\min}(X)^2} \ln\left( \frac{C_1 n\kappa}{\min\{r; n\}} \cdot \max\left\{1; \frac{\kappa r_\star}{\min\{r; n\} - r_\star}\right\} \cdot \frac{\|X\|}{\alpha} \right) \tag{5}$$

   *iterations we have that*

$$\frac{\|U_{\hat{t}}U_{\hat{t}}^T - XX^T\|_F}{\|X\|^2} \lesssim \frac{n^{21/16}\kappa^{81/16}r_\star^{1/8}}{(\min\{r; n\})^{15/16}} \cdot \frac{\alpha^{21/16}}{\|X\|^{21/16}}, \tag{6}$$

   *holds with probability at least $1 - Ce^{-\tilde{c}r}$.*

2. *Assume that $r_\star < r < 2r_\star$ and that the scale of initialization fulfills*

$$\alpha \lesssim \min\left\{ \frac{\varepsilon^{1/2}}{n^{3/4}\kappa^{1/2}} \left(\frac{2\kappa^2\sqrt{rn}}{\varepsilon}\right)^{-6\kappa^2}; \frac{\varepsilon}{n\kappa^7} \right\} \|X\|, \tag{7}$$

   *with $0 < \varepsilon < 1$. Then, after*

$$\hat{t} \lesssim \frac{1}{\mu\sigma_{\min}(X)^2} \ln\left( \frac{C_2\kappa n^2}{\varepsilon^2(r - r_\star)} \cdot \frac{\|X\|}{\alpha} \right)$$

   *iterations we have that*

$$\frac{\|U_{\hat{t}}U_{\hat{t}}^T - XX^T\|_F}{\|X\|^2} \lesssim r_\star^{1/8}(r - r_\star)^{3/8}\kappa^{81/16}\left(\frac{n}{\varepsilon} \cdot \frac{\alpha}{\|X\|}\right)^{21/16} \tag{8}$$

   *holds with probability at least $1 - (\tilde{C}\varepsilon)^{r - r_\star + 1} + \exp(-\tilde{c}r)$.*

---

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

## 3.2 Special case: $r = r_\star$

The following result deals with the scenario $r = r_\star$, that is, the iterates $U_t$ have as many parameters as the ground truth matrix $X$.

**Theorem 3.4.** *Let $X \in \mathbb{R}^{n \times r_\star}$ and assume we have $m$ measurements of the low rank matrix $XX^T$ of the form $y = \mathcal{A}(XX^T)$ with $\mathcal{A}$ the measurement operator. We assume $\mathcal{A}$ satisfies the restricted*

*isometry property for all matrices of rank at most $2r_\star + 1$ with constant $\delta \leq c\kappa^{-4}r_\star^{-1/2}$. To reconstruct $XX^T$ from the measurements we fit a model of the form $\bar{U} \mapsto \mathcal{A}(\bar{U}\bar{U}^T)$ with $\bar{U} \in \mathbb{R}^{n \times r_\star}$ via running gradient descent iterations of the form $U_{t+1} = U_t - \mu\nabla f(U_t)$ on the objective (2) with a step size obeying $\mu \leq c\kappa^{-4}\|X\|^{-2}$. Here, the initialization is given by $U_0 = \alpha U$, where $U \in \mathbb{R}^{n \times r_\star}$ has i.i.d. entries with distribution $\mathcal{N}(0, 1/\sqrt{r_\star})$. Assume that the scale of initialization fulfills*

$$\alpha \lesssim \min\left\{\frac{\varepsilon^{1/2}}{n^{3/4}\kappa^{1/2}}\left(\frac{2\kappa^2\sqrt{rn}}{\varepsilon}\right)^{-6\kappa^2}; \frac{\varepsilon^2}{n\sqrt{r_\star}\kappa^7}\right\}\|X\|,$$

*for some $0 < \varepsilon < 1$. Then with probability at least $1 - C\varepsilon + \exp(-cr_\star)$ after*

$$\hat{t} \lesssim \frac{1}{\mu\sigma_{\min}(X)^2}\ln\left(\frac{8\kappa^3 n^3}{\varepsilon^2}\cdot\frac{\|X\|}{\alpha}\right)$$

*iterations we have that*

$$\frac{\|U_{\hat{t}}U_{\hat{t}}^T - XX^T\|_F}{\|X\|^2} \lesssim r_\star^{1/8}\kappa^{81/16}\left(\frac{n}{\varepsilon}\cdot\frac{\alpha}{\|X\|}\right)^{21/16}. \tag{10}$$

*Here $C, c > 0$ are fixed numerical constants.*

Note that by choosing $\alpha$ small enough we can make the test error in (10) arbitrarily small. In particular, this means that then well-known local convergence results can be applied showing that $U_t U_t^T$ converges linearly to $XX^T$ (see, e.g., [6]).

Thus, this result implies that if the measurement operator fulfills the restricted isometry property, gradient descent with *small, random initialization* will converge to the ground truth matrix $X$ in polynomial time. It is known that under the RIP assumption the loss landscape is benign [42] in the sense that there are no local optima that are not global and all saddles have a direction of negative curvature. However, such results do not imply that vanilla gradient descent converges quickly (i.e. in polynomial time) to a global optimum, as gradient descent may take exponential time to escape from saddle points.

To the best of our knowledge, this is the first in the non-overparameterized setting $r = r_\star$ result which shows the convergence of vanilla gradient descent to the ground truth from a random initialization using only the restricted isometry property in polynomial time. The only other paperin the low-rank matrix recovery literature, which shows fast convergence of vanilla gradient descent to the ground truth from a random initialization, is [21]. In this work, the problem of phase retrieval has been studied, which can be formulated as a low-rank matrix recovery problem with $r = r_\star = 1$. The paper shows that gradient descent converges from a random initialization to the ground truth with a near-optimal number of iterations. However, the proof in this paper leverages the rotation-invariance of the Gaussian measurements vectors via carefully constructed auxiliary sequences. In contrast, Theorem 3.4 above relies only on the restricted isometry property and no further assumptions on $\mathcal{A}$ are needed.

### 3.3 Special case: $r = n$ with orthonormal initialization

In the following result, we study the scenario $r = n$, where the initialization matrix $U \in \mathbb{R}^{n \times n}$ is an orthonormal matrix, i.e. $U^T U = \mathrm{Id}$, instead of a Gaussian matrix as in the previous results in this paper. This is the same setting as in [38, Theorem 1.1], and we include this special case so as to explain how our results improves upon prior work in this special case.

**Theorem 3.5.** *Let $X \in \mathbb{R}^{n \times r_\star}$ and assume we have $m$ measurements of the low rank matrix $XX^T$ of the form $y = \mathcal{A}(XX^T)$ with $\mathcal{A}$ the measurement operator. We assume $\mathcal{A}$ satisfies the restricted isometry property for all matrices of rank at most $2r_\star + 1$ with constant $\delta \leq c\kappa^{-4}r_\star^{-1/2}$. To reconstruct $XX^T$ from the measurements we fit a model of the form $\bar{U} \mapsto \mathcal{A}(\bar{U}\bar{U}^T)$ with $\bar{U} \in \mathbb{R}^{n \times n}$ via running gradient descent iterations of the form $U_{t+1} = U_t - \mu\nabla f(U_t)$ on the objective (2) with a step size obeying $\mu \leq c\kappa^{-4}\|X\|^{-2}$. Here, the initialization is given by $U_0 = \alpha U$, where $U \in \mathbb{R}^{n \times n}$ can be any orthonormal matrix. Assume that the scale of initialization satisfies $\alpha \leq c\frac{\sigma_{\min}(X)}{\kappa^2 n}$. Then, after*

$$\hat{t} \lesssim \frac{1}{\mu\sigma_{\min}(X)^2}\ln\left(\max\left\{1; \frac{\kappa r_\star}{n - r_\star}\right\}\frac{\|X\|}{\alpha}\right)$$

*iterations we have that*

$$\frac{\|U_{\hat{t}}U_{\hat{t}}^T - XX^T\|_F}{\|X\|^2} \lesssim \frac{r_\star^{1/8} n^{3/8}}{\kappa^{3/16}} \cdot \frac{\alpha^{21/16}}{\|X\|^{21/16}}.$$

*Here $c > 0$ is a fixed numerical constant.*

Note that this result improves over [38, Theorem 4.1] in several aspects. First of all, in [38] it is assumed that the measurement operator $\mathcal{A}$ has the rank-$4r$ restricted isometry property with constant $\delta \lesssim \kappa^{-6} r_\star^{-1/2} \log^{-2} \frac{n}{\alpha}$. In particular, this suggests that this result cannot handle the scenario that the scale of initialization $\alpha$ becomes arbitrarily small, as this would also require that the restricted isometry constant $\delta$ becomes arbitrarily small as well. This in turn would require an arbitarily large sample size. Moreover, [38] requires a step size of at most $\mu \lesssim \kappa^{-6} r_\star^{-1/2} \log^2 n \|X\|^{-2}$, whereas the above theorem only needs the weaker assumption $\mu \lesssim \kappa^{-2} \|X\|^{-2}$. These improvements aside the main difference between our result and this prior work is that we can handle any $r$ by formalizing an intriguing connection between small random initialization and spectral learning.

## 4   Related work

**Global convergence guarantees for nonconvex low-rank matrix recovery**: As mentioned earlier in Section 1, there is a large body of work on developing global convergence guarantees for nonconvex problems. In the context of low-rank matrix recovery, several papers have demonstrated that low-rank reconstruction problems in a variety of domains can be solved via nonconvex gradient descent starting from spectral initialization. More precisely, this has been shown for phase retrieval [3, 4, 5], matrix sensing [43], blind deconvolution [7, 8], and matrix completion [44]. However, in practice often random initialization is used in lieu of specialized spectral initialization techniques. To remedy this issue, more recent literature [45, 46, 47], focusses on studying the loss landscape of such problems. These papers show that despite their non-convexity under certain assumptions these loss landscapes are benign in the sense that there are no *spurious local minima*, (i.e. all minimizers are global minima) and saddles points have a strict direction of negative curvature (a.k.a. strict saddle) [11]. Then specialized truncation or saddle escaping algorithms such as trust region, cubic regularization [12, 13] or noisy (stochastic) gradient-based methods [14, 15, 16, 17] are deployed to provably find a global optimum. These papers however do not directly develop global convergence for gradient descent (without any additional modification) from a random initialization. For differentiable losses eventual convergence to local minimizers is known from random initialization [18] but these results do not provide convergence rates and only guarantee eventual convergence. Indeed, gradient descent may converge exponentially slowly in the worst-case [19]. In contrast to the above literature our result in Theorem 3.4 (in the case of $r = r_\star$) shows that gradient descent from a small random initialization converges rather quickly to the global optima. As mentioned earlier, we are able to establish this result by demonstrating that in the initial phase gradient descent iterates are intimately connected to the spectral initialization techniques discussed above. Furthermore, the above spectral initialization followed by local convergence or landscape analysis techniques cannot be directly applied in in the overparameterized case ($r > r_\star$) whereas our analysis works regardless of model overparameterization.

We would like to mention that even more recently the paper [21] proves the convergence of gradient descent starting from a random initialization for low-rank recovery problems via an interesting leave-one-out analysis. To the best of our knowledge, this is the only existing result, which provides convergence guarantees for vanilla gradient descent from random initialization for low-rank matrix recovery problems in the non-overparameterized setting $r = r_\star$. However, the leave-one-out analysis heavily relies on the independence and the rotation invariance of the measurements. Also similar to the above this analysis does not seem to easily lend to generalization in the overparameterized regime. In contrast, our proof techniques rely on standard restricted isometry assumptions without requiring the independence of the measurements and does provide generalization guarantees with model overparameterization ($r > r_\star$). Moreover, in [48] it has been shown that Riemannian gradient descent converges with nearly linear rate to the true solution from a random initialization in the population loss scenario.

**Overparameterization in low-rank matrix recovery**: In the influential work [49] it has been conjectured and in the special case that the measurement matrices commute proven that gradient

descent on overparameterized matrix factorization converges to the solution with the minimal nuclear norm. This phenomenon is now often referred to as *implicit regularization*. In [20], evidence is provided that adding depth even increases the tendency of gradient descent to converge to low-rank solution. In [50] it has been shown that there are certain scenarios where the conjecture in [49] does not hold. In [51] theoretical and empirical evidence has been provided that gradient flow with infinitesimal initialization is equivalent to a certain rank-minimization heuristic.

In this paper, we shed further light on the implicit regularization of gradient descent. In particular, we provide a precise analysis of the initial stage and relate it to the power method and our analysis explains how overparameterization is beneficial in the initial stages. Closest to our work is the paper [38], which studies a special case of the problem analysed in this paper. More precisely, this paper considers the special case $r = n$ with orthonormal initialization. We also applied our theory to this exact same setting, see Section 3.3, where we include a detailed comparison for this special scenario. Most importantly our theory is able to handel the case $\alpha \to 0$, which the result in [38] seems not to be able to. Moreover, analysing the full range of possible choise of $r$ requires a careful analysis of the spectral phase, which is one key novely of this paper compared to [38].

In [52, 53] it has been shown that in certain scenarios, where the measurement matrices $A_i$ are positive semidefinite (PSD), the equation $y = \mathcal{A}\left(UU^T\right)$ has a unique low-rank solution. This means that in these scenarios the PSD constraint by itself might lead to a low-rank matrix recovery, which makes implicit regularization by gradient descent meaningless in this setting. However, note that these results not apply to the scenario studied in this paper, as we assume the measurement matrices $A_i$ to be Gaussian, which, in particular, means that they are not positive semidefinite. In particular, in our setting it can be shown that there are infinitely many solutions to the equation $y = \mathcal{A}\left(UU^T\right)$ with arbitrarily large test error [39].

**Gradient-based generalization guarantees for overparameterized tensors and neural networks:**
A recent line of work is concerned with connecting the analysis of neural network training with the so-called neural tangent kernel (NTK) [22, 23, 24, 25, 26]. The key idea is that for a large enough initialization, it suffices to consider a linearization of the neural network around the origin. This allows connecting the analysis of neural networks with the well-studied theory of kernel methods. This is also sometimes referred to as *lazy training*, as with such an initialization the parameters of the neural networks stay close to the parameters at initialization. However, there is a line of work, which suggests that NTK-analysis might not be sufficient to completely explain the success of neural networks in practice. The paper [34] provides empirical evidence that by choosing a smaller initialization the test error of the neural network decreases. A similar performance gap between the performance of the NTK and neural networks has been observed in [2], where it has been shown that the performance gap is larger if the covariance matrix is isotropic.

There is also a line of work [29, 30, 31, 32, 33], which is concerned with the mean-field analysis of neural networks. The insight is that for sufficiently large width the training dynamics of the neural network can be coupled with the evolution of a probability distribution described by a PDE. These papers use a smaller initialization than in the NTK-regime and, hence, the parameters can move away from the initialization. However, these results do not provide explicit convergence rates and require an unrealistically large width of the neural network.

For the problem of tensor decomposition it has also been shown that gradient descent with small initialization is able to leverage low-rank structure [54]. This is relevant to neural network analysis, since in [55] a relationship between tensor decomposition and training neural networks has been established. In [56] it has been shown that neural networks with ReLU function and trained by SGD can outperform any kernel method. One crucial element in their analysis is that the early stage of the training is connected with learning the first and second moment of the data.

While in this paper we do not study overparameterized tensor or neural network models we note that the NTK-theory can also be applied to low-rank matrix recovery (see [23, Section 4.2]). This means that if the scale of initialization is chosen large enough and the number of parameters is larger than the number of measurements, i.e. $nr \gtrsim m$, then gradient descent will converge linearly to a global minimizer with zero loss. However, since for this approach the parameters will stay close to the initialization, this approach will not recover the ground truth matrix $XX^T$. Hence, an NTK analysis will not yield good generalization. In contrast in this paper we have seen that choosing a small initialization is a remedy for low-rank matrix recovery. So in this sense our result can be

viewed as going beyond the lazy training in NTK theory. In fact we believe that similar analysis to the one developed in this paper for low-rank recovery can be used to analyze a much broader class of overparameterized models including the analysis of neural networks. We defer this to a future paper.

**Linear neural networks:** In [57, 58, 59, 60, 61] the convergence of gradient flow and gradient descent is studied for (deep) linear neural networks of the form

$$\min_{W_1, W_2, \ldots, W_N} \sum_{i=1}^{m} \left\| W_N \ldots W_2 W_1 x_i - y_i \right\|^2.$$

However, note that this model is different from the one studied in this paper. In [62] it is shown that gradient descent for convolutional linear neural networks has a bias towards the $\ell_p$-norm, where $p$ depends on the depth of the network.

# 5 Overview and key ideas of the proof

In this section, we briefly discuss the key ideas and techniques in our proof. We begin by discussing a simple decomposition, which is utilized throughout our proofs. Next, in Sections 5.2 and 5.3 we show that the trajectory of the gradient descent iterations can be approximately decomposed into three phases: (I) a *spectral or alignment phase* where we show that gradient descent from random initialization behaves akin to spectral initialization allowing us to show that at the end of this phase the column spaces of the iterates $U_t$ and the ground truth matrix $X$ are sufficiently aligned, (II) a *saddle avoidance phase*, where we show that the trajectory of the gradient iterates move away from certain degenerate saddle points, and (III) a *refinement phase*, where the product of the gradient descent iterates $U_t U_t^T$ converges quickly to the underlying low-rank matrix $XX^T$. The latter result holds up to a small error that is commensurate with the scale of the initialization and tends to zero as the scale of the initialization goes to zero. Figure 2 depicts these three phases.

Let us remark that the proof in the related work [38] decomposes the convergence analysis into two phases, which roughly correspond to Phase II and Phase III in our proof. However, the proof details are quite different since we use a different decomposition into signal and noise term, see Section 5.1.

## 5.1 Decomposition of $U_t$ into "signal" and "noise" matrices

A key idea in our proof is to decompose the matrix $U_t$ into the sum of two matrices. The first matrix, which is of rank $r_\star$, can be thought of as the "signal" term. We will show that the product of this matrix with its transpose converges towards the ground truth low-rank matrix $XX^T$. The second matrix, will have rank at most $r - r_\star$ and will have column span orthogonal to the column span of the ground truth matrix $X$. We will show that the spectral norm of this matrix will remain relatively small depending on the scale of initialization $\alpha$. Hence, this term can be interpreted as the "noise" term.

We now formally introduce our decomposition. To this aim, consider the matrix $V_X^T U_t \in \mathbb{R}^{r_\star \times r}$ and denote its singular value decomposition by $V_X^T U_t = V_t \Sigma_t W_t^T$ with $W_t \in \mathbb{R}^{r \times r_\star}$. Similarly, we shall use $W_{t,\perp} \in \mathbb{R}^{r \times (r-r_\star)}$ to denote the orthogonal matrix, whose column space is orthogonal to the column space of $W_t$ (i.e. the basis of the subspace orthogonal to the span of $W_t$). We then can decompose $U_t$ into

$$U_t = \underbrace{U_t W_t W_t^T}_{\text{signal term}} + \underbrace{U_t W_{t,\perp} W_{t,\perp}^T}_{\text{noise term}}.$$

This decomposition has the following two simple properties, which will be useful throughout our proofs.

**Lemma 5.1** (Properties of signal-noise decomposition)**.**

1. *The column space of the noise term is orthogonal to the column span of $X$, i.e. $V_X^T U_t W_{t,\perp} = 0$.*

2. *When $V_X^T U_t$ is full rank, then the signal term has rank $r_\star$ and the noise term has rank at most $r - r_\star$.*

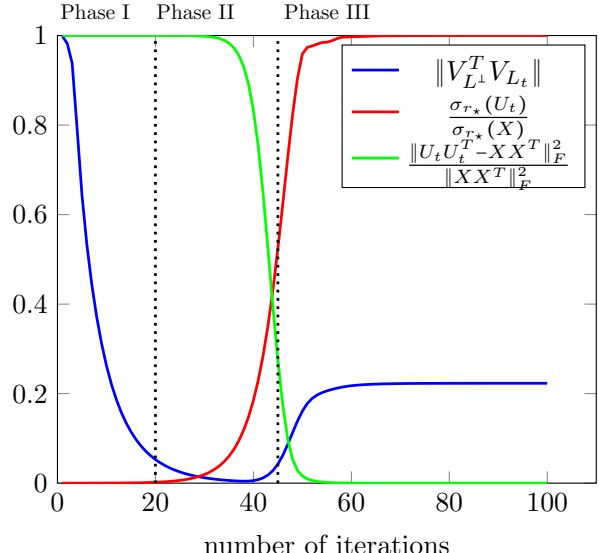

Figure 2: **Depiction of the three phases of convergence.** Let $L$ denote the subspace spanned by the eigenvectors corresponding to the $r_\star$ largest eigenvalues of the matrix $\mathcal{A}^\star \mathcal{A}\left(XX^T\right)$ and $L_t$ denote the subspace spanned by the eigenvectors corresponding to the $r_\star$ largest eigenvalues of the matrix $U_t U_t^T$. This figure demonstrates that the convergence analysis can be divided into three phases: (I) spectral/alignment phase; (II) saddle avoidance phase and (III) the refinement phase. We see that in the first phase the first $r_\star$ eigenvectors of $U_t U_t^T$ rapidly learn the subspace corresponding to the first $r_\star$ eigenvectors of $\mathcal{A}^\star \mathcal{A}\left(XX^T\right)$, i.e. the angle $\|V_{L^\perp}^T V_{L_t}\|$ becomes small. The $r_\star$th largest singular value of $U_t$ is still small in this phase and the (normalized) test error $\|U_t U_t - XX^T\|^2 / \|XX^T\|^2$ has not decreased yet. In Phase (II), however, we see that $\sigma_{r_\star}(U_t)$ is growing, whereas the loss begins to decrease in this phase and the subspaces stay aligned. In Phase (III) we see that the test error is converging towards 0 rapidly, meaning that $U_t U_t^T$ converges to $XX^T$. Consequently, $\sigma_{r_\star}(U_t)/\sigma_{r_\star}(X)$ converges to 1 (red curve). We also see that in this phase the angle $\|V_{L^\perp}^T V_{L_t}\|$ grows again, until it reaches a certain threshold. This is because in this phase the top $r_\star$ eigenvalues of $U_t U_t^T$ become aligned with the eigenvectors of $XX^T$.

*Proof.* The first statement follows directly from the observation $V_X^T U_t W_{t,\perp} W_{t,\perp}^T = V_X^T U_t\left(\mathrm{Id} - W_t W_t^T\right) = 0$. The second statement is a direct consequence of the definition of $W_t$. $\qquad\square$

We would like to note that decomposing $U_t$ into two terms has appeared in prior work such as [38] as well as in earlier work in the compressive sensing literature. However, [38] uses a different decomposition. A key advantage of our decomposition is that it only depends on $U_t$ and $X$, whereas the decomposition in [38] depends on all previous iterates $U_0, U_1, \ldots, U_{t-1}$.

## 5.2   The spectral/alignment phase

In this section we turn our attention to giving an overview of the key ideas and proofs of the spectral/alignment phase. More specifically, we will argue that in the first few iterations gradient descent implicitly performs a form of spectral initialization. By that, we mean that after the first few iterations the column span of the signal term $U_t W_t W_t^T$ is aligned with the column span of $X$ and that $\|U_t W_{t,\perp}\|$ is relatively small compared to $\sigma_{\min}(U_t W_t)$, meaning that the signal term dominates the noise term.

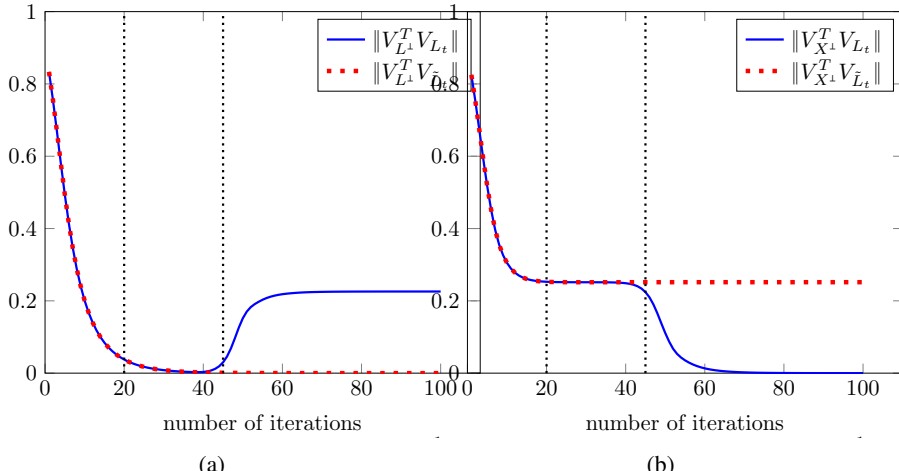

(a)                              (b)

Figure 3: **Depiction of the spectral alignment phase: in the first few iterations, gradient descent with small initialization behaves like a power method.** Here, $L$ denotes the subspace spanned by the eigenvectors corresponding to the $r_\star$ largest eigenvalues of the matrix $\mathcal{A}^*\mathcal{A}(XX^T)$. $L_t$ denotes the subspace spanned by the eigenvectors corresponding to the $r_\star$ largest eigenvalues of the matrix $U_t U_t^T$. Moreover, $\tilde{L}_t$ denotes the subspace spanned by the eigenvectors corresponding to the $r_\star$ largest eigenvalues of the matrix $\tilde{U}_t \tilde{U}_t^T$, where $\tilde{U}_t = \left(\mathrm{Id} + \mu\mathcal{A}^*\mathcal{A}(XX^T)\right)^t U_0$. In Figure 3a we see that in the first iterations $U_t$ and $\tilde{U}_t$ learn the subspace $L$ at the same rate. In Figure 3b we observe that also the angle between $V_X$ and $L_t$, respectively $\tilde{L}_t$, decreases monotonically in the spectral phase and then both angles stay constant in the saddle-avoidance phase. We see that in the local convergence phase the angle between $V_X$ and $L_t$ converges to 0 as expected since $U_t$ converges to $X$ up to a rotation.

We now provide the main intuition behind the analysis in our spectral/alignment phase. Our starting point is the observation that for the gradient at the initialization $U_0 = \alpha U$ it holds that

$$\nabla f(U_0) = -\left[\mathcal{A}^*\mathcal{A}(XX^T - U_0 U_0^T)\right]U_0$$
$$= -\alpha\left[\mathcal{A}^*\mathcal{A}(XX^T)\right]U + \alpha^3\left[\mathcal{A}^*\mathcal{A}(UU^T)\right]U.$$

In particular, we observe that for $\alpha > 0$ sufficiently small the second term is negligible. Hence, we have that

$$U_1 = U_0 - \mu\nabla f(U_0)$$
$$= \left(\mathrm{Id} + \mu\left[\mathcal{A}^*\mathcal{A}(XX^T)\right]\right)U_0 - \alpha^2\left[\mathcal{A}^*\mathcal{A}(UU^T)\right]U_0$$
$$= \left(\mathrm{Id} + \mu\mathcal{A}^*\mathcal{A}(XX^T)\right)U_0 + O\left(\alpha^2\|U_0\|\right)$$

In the first few iterations (i.e. small $t$) we expect the matrix $U_t$ to be small and continue to scale commensurately with $\alpha$ and we expect that a similar approximation holds for the first iterations. Hence, for $\alpha$ sufficiently small we can approximate $U_t$ by

$$U_t \approx \underbrace{\left(\mathrm{Id} + \mu\mathcal{A}^*\mathcal{A}(XX^T)\right)^t}_{=:Z_t} U_0 := \tilde{U}_t. \tag{11}$$

Figure 3 clearly illustrates that the first few iterations of gradient descent behave essentially identical to (11) confirming our intuition and proofs.

We indeed formally prove that such an approximation holds in Section 8. We note that the matrix $Z_1 = \mathrm{Id} + \mu\mathcal{A}^*\mathcal{A}(XX^T)$ is the basis for the commonly used spectral initialization, where typically a factorization of the rank $r_*$ approximation of this matrix is used as the initialization [6, 5, 44]. Therefore, the approximation (11) suggests that gradient descent iterates modulo the normalization are akin to running power method on $Z_1$. Therefore, we expect the column space of the signal term

at the end of the spectral phase to be closely aligned with those of the commonly used spectral initialization techniques and in turn the column space of $X$ as we formalize below.

To be more precise about the aforementioned alignment with $X$, let the singular value decomposition of $\mathcal{A}^* \mathcal{A} (XX^T)$ be given by $\mathcal{A}^* \mathcal{A} (XX^T) = \sum_{i=1}^{n} \lambda_i v_i v_i^T$. It follows that

$$U_t \approx \left[ \sum_{i=1}^{n} (1 + \mu \lambda_i)^t v_i v_i^T \right] U_0. \tag{12}$$

It is well-known that when the operator $\mathcal{A}$ obeys the restricted isometry property we have

$$\mathcal{A}^* \mathcal{A} (XX^T) \approx XX^T.$$

In particular, we have that

$$\lambda_{r_\star + 1} (\mathcal{A}^* \mathcal{A} (XX^T)) \ll \lambda_{r_\star} (\mathcal{A}^* \mathcal{A} (XX^T)).$$

Hence, it follows from

$$Z_t = \sum_{i=1}^{n} (1 + \mu \lambda_i)^t v_i v_i^T$$

that $\lambda_{r_\star} (Z_t) / \lambda_{r_\star + 1} (Z_t)$ grows exponentially. In particular, this means that

$$Z_t \approx \sum_{i=1}^{r_\star} (1 + \mu \lambda_i)^t v_i v_i^T$$

and, by (11),

$$U_t \approx \left[ \sum_{i=1}^{r_\star} (1 + \mu \lambda_i)^t v_i v_i^T \right] U_0.$$

Since $U_0$ is a *random* Gaussian matrix, for an appropriate choice of $t$, we will be able to show that the matrix $U_t$ has the following two properties with high probability, where $L = \mathrm{span}\{v_1; \ldots; v_{r_\star}\}$ and $L_t$ is the projection of $U_t$ onto its best rank-$r_\star$ approximation:

- There is a sufficiently large gap between $\sigma_{r_\star} (U_t)$ and $\sigma_{r_\star + 1} (U_t)$, i.e., $\frac{\sigma_{r_\star}(U_t)}{\sigma_{r_\star+1}(U_t)} \geq \Delta > 1$, where $\Delta$ is an appropriately chosen constant.
- We have that $\|V_{L^\perp}^T V_{L_t}\|$ is small. Since the column space of $\mathcal{A}^* \mathcal{A} (XX^T)$ is aligned with the column space of $X$, this also implies that $\|V_{X^\perp}^T V_{L_t}\|$ is small.

This confirms that in the first few iterations, gradient descent indeed implicitly performs akin to spectral initialization with the column space of $U_t$ aligned with the column space of $X$. However, this does not yet fully complete our analysis for the spectral/alignment phase, since critical to the analysis of second phase we need certain properties to hold for the signal and noise terms $U_t W_t$ and $U_t W_{t,\perp}$ (see Section 5.1) rather than the singular value decomposition of $U_t$. However, using the properties of the SVD of $U_t$, which are listed above, we will establish the following properties of $U_t W_t$ and $U_t W_{t,\perp}$ .

- The column space of $U_t W_t$ is aligned with the column space of $X$: $\|V_{X^\perp}^T V_{U_t W_t}\| \leq c \kappa^{-2}$.
- The spectral norm of the noise term is not too large compared to the minimum singular value of the signal term, i.e., $2 \sigma_{\min} (U_t W_t) \geq \|U_t W_{t,\perp}\|$.
- The spectral norm of the noise term is bounded from above in the sense that i.e., $\|U_t W_{t,\perp}\| \ll \sigma_{\min} (X)$.
- The spectral norm of $U_t$ is bounded, i.e., $\|U_t\| \leq 3\|X\|$.

### 5.3 The saddle avoidance phase and the refinement phase

In the next two phases, we will show that the signal term $U_t W_t W_t^T U_t^T$ converges towards $XX^T$, whereas the spectral norm of the noise term, i.e. $\|U_t W_{t,\perp}\|$, stays small. For that, we show that throughout this process the columns of the matrices $X$ and $U_t W_t$ stay approximately aligned, i.e., the angle $\|V_{X^\perp}^T V_{U_t W_t}\|$ stays small. This latter property also ensures that after the spectral phase the iterates are not too close to well known saddle points of the optimization landscape (it is known that

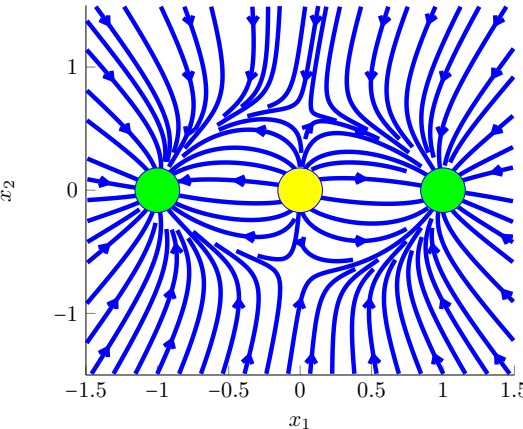

Figure 4: **Depiction of saddle avoidance and local refinement phases.** In this figure we depict the gradient field of the loss function $f$ with $n = 2$, $r_\star = r = 1$, $m = 15$, and $X = (1\ \ 0)$. The green circles depict the two generalizable global minima of $f$, namely $(1\ \ 0)$ and $(-1\ \ 0)$. The red circle depicts the saddle point $(0\ \ 0)$. As this figure demonstrates starting from small random initialization after a while the trajectory moves away from the saddle (i.e. avoids it) and then converges to one of the two generalizable global optima (i.e. the local refinement phase).

this problem may have degenerate saddle points at a point $\bar{U}$ obeying $\text{rank}(\bar{U}) < r_\star$ [41]). See Figure 4 for a depiction of the gradient flows of the landscape when $r_\star = 1$.

Next we sketch the proofs of Phase II and Phase III in more detail.

**Phase II:** In this phase, we will show that the minimal singular value of the signal term, $\sigma_{\min}(U_t W_t)$ grows exponentially, until it holds that $\sigma_{\min}(U_t W_t) \geq \frac{\sigma_{\min}(X)}{\sqrt{10}}$. To this aim, we show that

$$\sigma_{\min}(U_{t+1}W_{t+1}) \geq \sigma_{\min}(V_X^T U_{t+1}) \geq \sigma_{\min}(V_X^T U_t)\left(1 + \frac{1}{4}\mu\sigma_{\min}^2(X) - \mu\sigma_{\min}^2(V_X^T U_t)\right)$$

holds under suitable assumptions (see Lemma 9.1). In order to show that the spectral norm of the noise term $\|U_t W_{t,\perp}\|$ grows much slower than $\sigma_{\min}(U_{t+1}W_{t+1})$, we establish the inequality

$$\begin{aligned}&\|U_{t+1}W_{t+1,\perp}\|\\&\leq\left(1 - \frac{\mu}{2}\|U_t W_{t,\perp}\|^2 + 9\mu\|V_{X^\perp}^T V_{U_t W_t}\|\|X\|^2 + 2\mu\|(\mathcal{A}^*\mathcal{A} - \text{Id})(XX^T - U_t U_t^T)\|\right)\|U_t W_{t,\perp}\|\end{aligned} \tag{13}$$

(see Lemma 9.2). The next inequality (see Lemma 9.3) shows that $\|V_{X^\perp}^T V_{U_t W_t}\|$ stays sufficiently small

$$\begin{aligned}&\|V_{X^\perp}^T V_{U_{t+1}W_{t+1}}\|\\&\leq\left(1 - \frac{\mu}{4}\sigma_{\min}^2(X)\right)\|V_{X^\perp}^T V_{U_t W_t}\| + 100\mu\|(\text{Id} - \mathcal{A}^*\mathcal{A})(XX^T - U_t U_t^T)\| + 500\mu^2\|XX^T - U_t U_t^T\|^2.\end{aligned}$$

As mentioned above, this implies in particular, that $U_t$ stays sufficiently far away from saddle points $\bar{U}$, which are rank-deficient, e.g., $\text{rank}(\bar{U}) < r_\star$.

**Phase III:** After we have shown that $\sigma_{\min}(U_t W_t) \geq \frac{\sigma_{\min}(X)}{\sqrt{10}}$ holds for some $t$, we enter the *local refinement phase*. We start by observing that the error $\|XX^T - U_t U_t^T\|_F$ can be decomposed into two summands, i.e.

$$\|U_t U_t^T - XX^T\|_F \leq 4\|V_X^T(XX^T - U_t U_t^T)\|_F + \|U_t W_{t,\perp}W_{t,\perp}^T U_t^T\|_F. \tag{14}$$

(see Lemma B.4). We will bound the second summand by using inequality (13), which is also valid for the third phase. We will show that the first summand decreases at a linear rate. For that, we

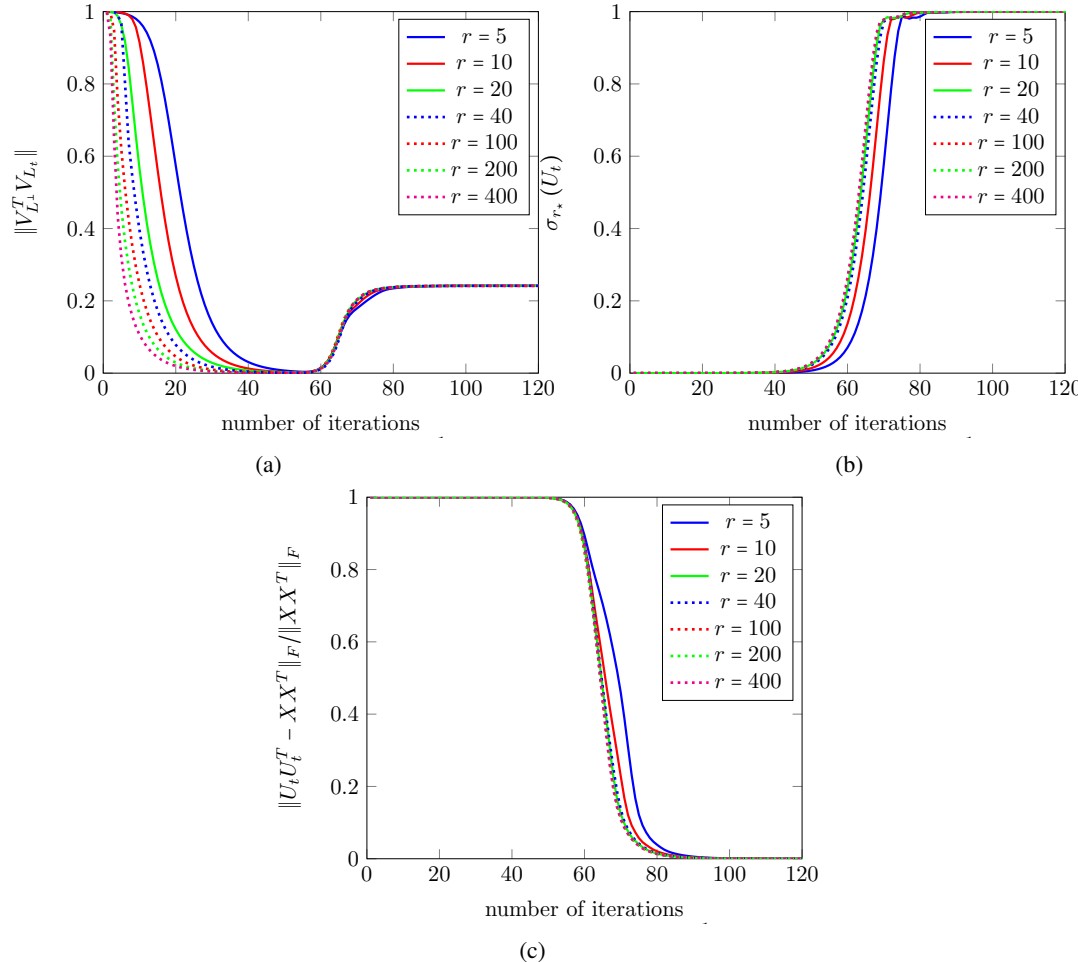

Figure 5: Impact of different levels of overparameterization on (a) the angle $\|V_{L^\perp}^T V_{L_t}\|$ and (b) the $r_\star$th largest singular value, (c) the trajectory of the (normalized) test error $\|U_t U_t^T - XX^T\|_F / \|XX^T\|_F$.

establish the inequality

$$\|V_X^T \left( XX^T - U_{t+1}U_{t+1}^T \right)\|_F$$
$$\leq \left( 1 - \frac{\mu}{200} \sigma_{\min}(X)^2 \right) \|V_X^T \left( XX^T - U_t U_t^T \right)\|_F + \mu \frac{\sigma_{\min}^2(X)}{100} \|U_t W_{t,\perp} W_{t,\perp}^T U_t^T\|_F.$$

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

# 7 Preliminaries

Before we are going into the details of the proof, we are collecting some useful definitions.

## 7.1 Notation

For any matrix $A \in \mathbb{R}^{n_1 \times n_2}$ we denote its spectral norm by $\|A\|$ and the Frobenius norm by $\|A\|_F = \sqrt{\mathrm{Tr}(AA^T)}$. By $\|A\|_*$ we denote its nuclear norm, i.e. the sum of the singular values. Moreover, for two symmetric matrices $A, B \in S^d$ we define the Hilbert-Schmidt inner product by $\langle A, B \rangle = \mathrm{Tr}(AB)$.

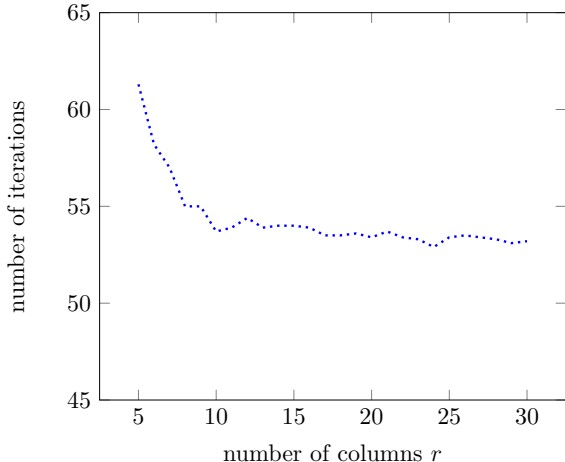

Figure 8: Number of iterations required for the test error to fall below $10^{-4}$ for different levels of overparameterization.

For a positive semidefinite matrix $A$ we denote its square root by $A^{1/2}$, i.e., the unique positive semidefinite matrix $B$ for which it holds that $B^2 = A$. We also set $A^{-1/2} = \left(A^{1/2}\right)^{-1}$.

For any matrix $A \in \mathbb{R}^{d_1 \times d_2}$ we will denote its singular value decomposition by $A = V_A \Sigma_A W_A^T$ with $V_A \in \mathbb{R}^{d_1 \times \tilde{r}}$, $W_A \in \mathbb{R}^{d_2 \times \tilde{r}}$, $\Sigma_A \in \mathbb{R}^{\tilde{r} \times \tilde{r}}$, where $\tilde{r}$ denotes the rank of $A$. Moreover, by $V_{A^\perp} \in \mathbb{R}^{(d_1 - \tilde{r}) \times d_1}$ we denote an orthogonal matrix, whose column span is orthogonal to the column span of the matrix $V_A$. Similarly, if $U \subset \mathbb{R}^n$ is a subspace of dimension $\tilde{r}$, we will denote by $V_U \in \mathbb{R}^{n \times \tilde{r}}$ a matrix, whose column span is the subspace $U$. Similarly as before, we will denote by $V_{U^\perp} \in \mathbb{R}^{n \times (n - \tilde{r})}$ a matrix whose column span is orthogonal to the column span of $U$.

We will measure the angle between two subspaces $U_1, U_2 \subset \mathbb{R}^n$ by $\|V_{U_1^\perp}^T V_{U_2}\|$. Moreover, we will also several times rely on the well-known identity (see, e.g., [63, Section 2])

$$\|V_{U_1^\perp}^T V_{U_2}\| = \|V_{U_1} V_{U_1}^T - V_{U_2} V_{U_2}^T\|.$$

### 7.2 Restricted isometry property and related properties

As discussed in Section 2, we are going to assume that the measurement operator $\mathcal{A}$ satisfies the restricted isometry property. However, as it turns out, the following two slightly weaker properties will suffice for our proof.

**Definition 7.1.** *The measurement operator $\mathcal{A} : S^n \longrightarrow \mathbb{R}^m$ satisfies the spectral-to-spectral restricted isometry property of rank $r$ with constant $\delta > 0$, if it holds for all symmetric matrices $Z$ of rank at most $r$ that*

$$\| \left(Id - \mathcal{A}^* \mathcal{A}\right)(Z) \| \leq \delta \|Z\|.$$

**Definition 7.2.** *The measurement operator $\mathcal{A} : S^n \longrightarrow \mathbb{R}^m$ satisfies the spectral-to-nuclear restricted isometry property with constant $\delta > 0$, if it holds for all symmetric matrices $Z$ hat*

$$\| \left(Id - \mathcal{A}^* \mathcal{A}\right)(Z) \| \leq \delta \|Z\|_*.$$

The following lemma shows that these two properties are induced by the standard restricted isometry property (Definition 3.1).

**Lemma 7.3.** *Let $\mathcal{A} : S^n \longrightarrow \mathbb{R}^m$ be a linear measurement operator. Then the following two statements hold.*

1. *Suppose that $\mathcal{A}$ has the restricted isometry property as in (3) for all matrices of rank $r + 1$ with constant $\delta_1 < 1$. Then $\mathcal{A}$ has the spectral-to-spectral restricted isometry property of rank $r$ with constant $\sqrt{r}\delta_1$.*

2. *Suppose that $\mathcal{A}$ has the restricted isometry property as in (3) for all matrices of rank 2 with constant $\delta_2 < 1$. Then $\mathcal{A}$ has the spectral-to-nuclear restricted isometry property with constant $\delta_2$.*

*Proof.* From [37] it follows that if $\mathcal{A}$ has the restricted isometry property of rank $r + r'$ with constant $\delta < 0$, then it holds for all matrices $Z$ with rank at most $r$ and all matrices $Y$ with rank at most $r'$ that

$$\left| \langle (\mathrm{Id} - \mathcal{A}^* \mathcal{A}) (Z), Y \rangle \right| = \left| \langle \mathcal{A}^* \mathcal{A} (Z), Y \rangle - \langle Z, Y \rangle \right| \leq \delta_1 \|Z\|_F \|Y\|_F. \tag{15}$$

In order to prove the first statement it suffices to note that there is a vector $v \in \mathbb{R}^n$ with $\|v\|_{\ell_2} = 1$ such that

$$\| (\mathrm{Id} - \mathcal{A}^* \mathcal{A}) (Z) \| = \langle (\mathrm{Id} - \mathcal{A}^* \mathcal{A}) (Z), vv^T \rangle.$$

The claim follows from (15), $\|v\|_{\ell_2} = 1$, and from $\|Z\|_F \leq \sqrt{r} \|Z\|$.

In order to show the second claim consider the eigenvalue decomposition $Z = \sum_{i=1}^n \lambda_i v_i v_i^T$. We compute that

$$\| (\mathrm{Id} - \mathcal{A}^* \mathcal{A}) (Z) \| \leq \sum_{i=1}^n |\lambda_i| \| (\mathrm{Id} - \mathcal{A}^* \mathcal{A}) (v_i v_i^T) \|$$

$$\leq \delta_2 \sum_{i=1}^n |\lambda_i| \|v_i v_i^T\|$$

$$= \|Z\|_*,$$

where in the second inequality we used the spectral-to-spectral restricted isometry property, which holds due to the first part of this proof. This finishes the proof of the second statement. $\qquad\square$

# 8 Analysis of the spectral phase

In the following we will provide an analysis of the spectral phase, where the proofs of the technical lemmas are deferred to Appendix A. Our first goal is to show that in the first few iterations $U_t$ can be approximated by

$$\widetilde{U}_t := \left( \mathrm{Id} + \mu \underbrace{\mathcal{A}^* \mathcal{A} \left( XX^T \right)}_{=:M} \right)^t U_0 =: Z_t U_0,$$

where we have set

$$Z_t = \left( \mathrm{Id} + \mu M \right)^t.$$

Next, we define

$$t^\star := \min \left\{ i \in \mathbb{N} : \|\widetilde{U}_{i-1} - U_{i-1}\| > \|\widetilde{U}_{i-1}\| \right\}.$$

The next lemma shows how well $U_t$ can be approximated by $\widetilde{U}_t$ for $t \leq t^\star$. To formulate it, we set $E_t = U_t - \widetilde{U}_t$.

**Lemma 8.1.** *Suppose that $\mathcal{A}$ satisfies the rank-1 RIP with constant $\delta_1$. For all integers $t$ such that $1 \leq t \leq t^\star$ it holds that*

$$\|E_t\| = \|U_t - \widetilde{U}_t\| \leq \frac{4}{\lambda_1 (M)} \alpha^3 \min \{r; n\} (1 + \delta_1) (1 + \mu \lambda_1 (M))^{3t} \|U\|^3.$$

The next lemma gives a lower bound for $t^\star$. In particular, this shows how long the approximation in Lemma 8.1 is valid.

**Lemma 8.2.** *Let $\widetilde{U}_t$ be as defined before and consider the eigenvalue decomposition $\mathcal{A}^* \mathcal{A} \left( XX^T \right) = \sum_{i=1}^n \lambda_i v_i v_i^T$. Then we have that*

$$t^\star \geq \left\lceil \frac{\ln \left( \frac{\lambda_1 (M)}{4\alpha^2 (1+\delta_1) \|U\|^3} \left( \frac{\|U_0^T v_1\|_{\ell_2}}{\alpha \min\{r;n\}} \right) \right)}{2 \ln (1 + \mu \lambda_1 (M))} \right\rceil.$$

Next, recall the relation
$$U_t = \widetilde{U}_t + E_t = Z_t U_0 + E_t$$
and denote by $L$ the subspace spanned by the eigenvectors, which correspond to the largest $r_\star$ eigenvalues of the matrix $M = \mathcal{A}^\star \mathcal{A}(XX^T)$. Note that $L$ is also the subspace spanned by the eigenvectors corresponding to the largest $r_\star$ eigenvalues of the matrix $Z_t$. Denote by $L_t$ the subspace spanned by the left-singular vectors of $U_t = Z_t U_0 + E_t$, which corresponds to the largest $r_\star$ singular values.

Since $Z_t$ is computed via a power method we expect for $t$ large enough that $\lambda_{r_\star}(Z_t) \gg \lambda_{r_\star+1}(Z_t)$. Moreover, if, in addition, $\|E_t\|$ is sufficiently small, we expect in this case that the subspace $L$ is aligned with the subspace $L_t$. This is made precise by the following lemma.

**Lemma 8.3.** *Let $Z_t \in \mathbb{R}^{n \times n}$ be a symmetric matrix. Let $U \in \mathbb{R}^{n \times r}$ be a matrix and let $E_t \in \mathbb{R}^{n \times r}$. Set $U_0 = \alpha U$ for some $\alpha > 0$. Moreover, assume that*

$$\sigma_{r_\star+1}(Z_t) \|U\| + \frac{\|E_t\|}{\alpha} < \sigma_{r_\star}(Z_t) \sigma_{\min}(V_L^T U). \tag{16}$$

*Then the following three inequalities hold.*

$$\sigma_{r_\star}(Z_t U_0 + E_t) \geq \alpha \sigma_{r_\star}(Z_t) \sigma_{\min}(V_L^T U) - \|E_t\|, \tag{17}$$

$$\sigma_{r_\star+1}(Z_t U_0 + E_t) \leq \alpha \sigma_{r_\star+1}(Z_t) \|U\| + \|E_t\|, \tag{18}$$

$$\|V_{L^\perp}^T V_{L_t}\| \leq \frac{\alpha \sigma_{r_\star+1}(Z_t) \|U\| + \|E_t\|}{\alpha \sigma_{r_\star}(Z_t) \sigma_{\min}(V_L^T U) - \alpha \sigma_{r_\star+1}(Z_t) \|U\| - \|E_t\|}. \tag{19}$$

Recall that we are interested in bounds for the quantities $\sigma_{r_\star}(U_t W_t)$, $\|V_{X^\perp}^T V_{U_t W_t}\|$, and $\|U_t W_{t,\perp}\|$, i.e., properties of the signal and noise term. However, in the lemma above, we have obtained instead bounds for $\sigma_{r_\star}(U_t)$, $\|V_{X^\perp}^T V_{L_t}\|$, and $\sigma_{r_\star+1}(U_t)$, i.e. for the singular value decomposition of $U_t$. However, if $\|V_{X^\perp}^T V_{L_t}\|$ is small, these quantities are closely related to each other, as the next lemma shows.

**Lemma 8.4.** *Assume that $\|V_{X^\perp}^T V_{L_t}\| \leq \frac{1}{8}$ for some $t \geq 1$. Then it holds that*

$$\sigma_{r_\star}(U_t W_t) \geq \frac{1}{2}\sigma_{r_\star}(U_t), \tag{20}$$

$$\|V_{X^\perp}^T V_{U_t W_t}\| \leq 7\|V_{X^\perp}^T V_{L_t}\|, \tag{21}$$

$$\|U_t W_{t,\perp}\| \leq 2\sigma_{r_\star+1}(U_t). \tag{22}$$

By combining Lemma 8.3 and Lemma 8.4, we obtain the following technical result.

**Lemma 8.5.** *Let $XX^T$ be a low-rank matrix of rank $r_\star$. Assume that*

$$M := \mathcal{A}^\star \mathcal{A}(XX^T) = XX^T + \tilde{E},$$

*with $\|\tilde{E}\| \leq \delta \lambda_{r_\star}(XX^T)$, where $\delta \leq \widetilde{c}_1$ where $\widetilde{c}_1 > 0$ is a sufficiently small absolute constant. Furthermore, set $E_t = U_t - \tilde{U}_t$. Moreover, assume that*

$$\gamma := \frac{\alpha \sigma_{r_\star+1}(Z_t) \|U\| + \|E_t\|}{\alpha \sigma_{r_\star}(Z_t) \sigma_{\min}(V_L^T U)} \leq \widetilde{c}_2 \kappa^{-2}, \tag{23}$$

*where $\widetilde{c}_2 > 0$ is a sufficiently small, absolute constant. Then it holds that*

$$\sigma_{\min}(U_t W_t) \geq \frac{\alpha}{4}\sigma_{r_\star}(Z_t) \sigma_{\min}(V_L^T U), \tag{24}$$

$$\|U_t W_{t,\perp}\| \leq \frac{\kappa^{-2}}{8}\alpha \sigma_{r_\star}(Z_t) \sigma_{\min}(V_L^T U), \tag{25}$$

$$\|V_{X^\perp}^T V_{U_t W_t}\| \leq 56(\delta + \gamma). \tag{26}$$

In order to utilize Lemma 8.5, we need to insert bounds for the approximation error $\|E_t\|$, which we have derived in the Lemmas 8.1 and 8.2. This yields the following lemma.

**Lemma 8.6.** *Fix a sufficiently small constant $c > 0$. Let $U \in \mathbb{R}^{n \times r}$. Assume that $\mathcal{A}$ has the spectral-to-nuclear restricted isometry property for some constant $\delta_1 < 1$. Moreover, assume that*

$$M := \mathcal{A}^* \mathcal{A} \left( X X^T \right) = X X^T + \tilde{E},$$

*with $\| \tilde{E} \| \le \delta \lambda_{r_\star} \left( X X^T \right)$, where $\delta \le c_1 \kappa^{-2}$. Denote by $L$ the subspace spanned by the eigenvectors corresponding to the $r_\star$ largest eigenvalues of the matrix $\mathcal{A}^* \mathcal{A} \left( X X^T \right)$. Let $U_0 = \alpha U$, where*

$$\alpha^2 \le \frac{c_2 \|X\|^2}{32 \min \{r; n\} \kappa \|U\|^3} \left( \frac{2 \kappa^2 \|U\|}{c_3 \sigma_{\min} \left( V_L^T U \right)} \right)^{-12 \kappa^2} \min \left( \sigma_{\min} \left( V_L^T U \right); \left\| U^T v_1 \right\|_{\ell_2} \right), \qquad (27)$$

*where $v_1$ denotes the eigenvector corresponding to a leading eigenvalue of the matrix $\mathcal{A}^* \mathcal{A} \left( X X^T \right)$. Assume that the step size satisfies $\mu \le c_2 \kappa^{-2} \|X\|^{-2}$. Then after*

$$t_\star \asymp \frac{1}{\mu \sigma_{r_\star} \left( X \right)^2} \cdot \ln \left( \frac{2 \kappa^2 \|U\|}{c_3 \sigma_{\min} \left( V_L^T U \right)} \right)$$

*iterations it holds that*

$$\| U_{t_\star} \| \le 3 \|X\|, \qquad (28)$$

$$\sigma_{\min} \left( U_{t_\star} W_{t_\star} \right) \ge \frac{\alpha \beta}{4}, \qquad (29)$$

$$\| U_{t_\star} W_{t_\star, \perp} \| \le \frac{\kappa^{-2}}{8} \alpha \beta, \qquad (30)$$

$$\| V_{X^\perp}^T V_{U_{t_\star} W_{t_\star}} \| \le c \kappa^{-2}, \qquad (31)$$

*where $\beta > 0$ satisfies*

$$\sigma_{\min} \left( V_L^T U \right) \lesssim \beta \lesssim \sigma_{\min} \left( V_L^T U \right) \left( \frac{\kappa^2 \|U\|}{c_3 \sigma_{\min} \left( V_L^T U \right)} \right)^2 = \frac{\kappa^4 \|U\|^2}{c_3^2 \sigma_{\min} \left( V_L^T U \right)}. \qquad (32)$$

*Here $c_1, c_2, c_3 > 0$ are absolute constants only depending on the choice of $c$.*

Note that the result above holds for any initialization $U$. To complete the proof we are going to utilize the fact that $U$ is a random matrix with Gaussian entries. This yields the following lemma, which is the main result of this section.

**Lemma 8.7.** *Fix a sufficiently small constant $c > 0$. Let $U \in \mathbb{R}^{n \times r}$ be a random matrix with i.i.d. entries with distribution $\mathcal{N} \left( 0, 1/\sqrt{r} \right)$ and let $0 < \varepsilon < 1$. Assume that $\mathcal{A}$ has the spectral-to-nuclear restricted isometry property for some constant $\delta_1 < 1$. Moreover, assume that*

$$M := \mathcal{A}^* \mathcal{A} \left( X X^T \right) = X X^T + \tilde{E},$$

*with $\| \tilde{E} \| \le \delta \lambda_{r_\star} \left( X X^T \right)$, where $\delta \le c_1 \kappa^{-2}$. Let $U_0 = \alpha U$, where*

$$\alpha^2 \lesssim \begin{cases} \frac{\sqrt{\min\{r;n\}} \|X\|^2}{\kappa n^{3/2}} \left( 2 \kappa^2 \sqrt{\frac{n}{\min\{r;n\}}} \right)^{-12 \kappa^2} & \text{if } r \ge 2 r_\star \\ \frac{\|X\|^2}{n^{3/2} \kappa} \left( \frac{2 \kappa^2 \sqrt{rn}}{\varepsilon} \right)^{-12 \kappa^2} \varepsilon & \text{if } r < 2 r_\star \end{cases}. \qquad (33)$$

*Assume that the step size satisfies $\mu \le c_2 \kappa^{-2} \|X\|^2$. Then with probability at least $1 - p$, where*

$$p = \begin{cases} O \left( \exp \left( -\tilde{c} r \right) \right) & \text{if } r \ge 2 r_\star \\ \left( \tilde{C} \varepsilon \right)^{r - r_\star + 1} + \exp \left( -\tilde{c} r \right) & \text{if } r < 2 r_\star \end{cases}$$

*the following statement holds. After*

$$t_\star \lesssim \begin{cases} \frac{1}{\mu \sigma_{r_\star} (X)^2} \cdot \ln \left( 2 \kappa^2 \sqrt{\frac{n}{\min\{r;n\}}} \right) & \text{if } r \ge 2 r_\star \\ \frac{1}{\mu \sigma_{r_\star} (X)^2} \cdot \ln \left( \frac{2 \kappa^2 \sqrt{rn}}{\varepsilon} \right) & \text{if } r < 2 r_\star \end{cases}$$

*iterations it holds that*

$$\|U_{t_\star}\| \le 3\|X\|, \tag{34}$$

$$\sigma_{\min}(U_{t_\star} W_{t_\star}) \ge \frac{\alpha\beta}{4}, \tag{35}$$

$$\|U_{t_\star} W_{t_\star,\perp}\| \le \frac{\kappa^{-2}}{8}\alpha\beta, \tag{36}$$

$$\|V_{X^\perp}^T V_{U_{t_\star} W_{t_\star}}\| \le c\kappa^{-2}, \tag{37}$$

*where $\beta > 0$ satisfies*

$$\beta \lesssim \begin{cases} \frac{n\kappa^4}{c_3^2(\min\{r;n\})} & \text{if } r \ge 2r_\star \\ \frac{n\kappa^4}{c_3^2\varepsilon} & \text{if } r < 2r_\star \end{cases}$$

*as well as*

$$\beta \gtrsim \begin{cases} 1 & \text{if } r \ge 2r_\star \\ \frac{\varepsilon}{r} & \text{if } r < 2r_\star \end{cases}.$$

*Here $c_1, c_2, c_3 > 0$ are absolute constants only depending on the choice of c. Moreover, $\tilde{C}, \tilde{c} > 0$ are absolute numerical constants.*

The proof of Lemma 8.7 requires the following theorem, which gives a non-asymptotic lower bound for the smallest singular value of a Gaussian matrix.

**Theorem 8.8.** *[64] Let $G \in \mathbb{R}^{r_\star \times r}$ with $r_\star \le r$ and i.i.d. Gaussian entries with distribution $\mathcal{N}(0, 1/\sqrt{r})$. Then for every $\varepsilon > 0$ we have with probability at least $1 - (C\varepsilon)^{r-r_\star+1} - \exp(-cr)$ that*

$$\sigma_{\min}(G) \ge \varepsilon \frac{\sqrt{r} - \sqrt{r_\star - 1}}{\sqrt{r}}.$$

*The constants $C, c > 0$ are universal.*

With this theorem in place we can prove Lemma 8.7.

*Proof of Lemma 8.7.* We will deduce this statement from Lemma 8.6. For that we need to estimate $\|U\|$, $\sigma_{\min}(V_L^T U)$, and $\|U^T v_1\|_{\ell_2}$. It is well-known (see, e.g. [40, Section 4]) that with probability at least $1 - O(\exp(-c\max\{r;n\}))$ it holds that

$$\|U\| \lesssim \sqrt{\max\{r;n\}/r} = \sqrt{\frac{n}{\min\{r;n\}}}. \tag{38}$$

Next, note that again due to rotation invariance of the Gaussian measure the vector $U^T v_1 \in \mathbb{R}^r$ has i.i.d. entries with distribution $\mathcal{N}(0, 1/\sqrt{r})$. Hence, with probability at least $1 - O(\exp(-cr))$ it holds that

$$\|U^T v_1\|_{\ell_2} \asymp 1. \tag{39}$$

Next, we note that due to rotation invariance of the Gaussian distribution the matrix $V_L^T U \in \mathbb{R}^{r_\star \times r}$ has i.i.d. entries with distribution $\mathcal{N}(0, 1/\sqrt{r})$. Moreover, note that using the elementary inequality $\sqrt{1-x} \le 1 - \frac{1}{2x}$ we obtain that

$$\frac{\sqrt{r} - \sqrt{r_\star - 1}}{\sqrt{r}} \ge \frac{\sqrt{r} - \sqrt{r_\star}\left(1 - \frac{1}{2r_\star}\right)}{\sqrt{r}} \gtrsim \begin{cases} 1 & \text{if } r \ge 2r_\star \\ \frac{1}{r} & \text{else} \end{cases}. \tag{40}$$

In order to proceed we are going to distinguish the following two cases.

**Case 1:** $r \ge 2r_\star$
Note that by choosing $\varepsilon > 0$ appropriately, we obtain from Theorem 8.8 combined with inequality (40) that with probability at least $1 - O(\exp(-cr))$ it holds that

$$\sigma_{\min}(V_L^T U) \gtrsim 1. \tag{41}$$

By combining the inequalities (38), (39), and (41) with Lemma 8.6 the claim follows in the case that $r \geq 2r_\star$.

**Case 2:** $r_\star \leq r \leq 2r_\star$

Similar to the first case, we note that by choosing $\varepsilon > 0$ appropriately, we obtain by applying Theorem 8.8 combined with inequality (40) that with probability at least $1 - (C\varepsilon)^{r - r_\star + 1} - \exp(-cr)$ it holds that

$$\sigma_{\min}\left(V_L^T U\right) \gtrsim \frac{\varepsilon}{r}. \tag{42}$$

By combining the inequalities (38), (39), and (42) with Lemma 8.6 the claim follows. $\square$

## 9 Analysis of the saddle avoidance and refinement phases

Before stating and proving the main result of this section, Theorem 9.6, we will first collect some useful lemmas. Their proofs are deferred to Appendix B.

In Phase II we will show that $\sigma_{\min}(U_t W_t)$ grows until it reaches $\sigma_{\min}(U_t W_t) \geq \frac{\sigma_{\min}(X)}{\sqrt{10}}$. For that, we note

$$
\begin{aligned}
\sigma_{\min}(U_t W_t) &\overset{(a)}{=} \sigma_{\min}\left(U_t W_t W_t^T\right) \\
&\geq \sigma_{\min}\left(V_X^T U_t W_t W_t^T\right) \\
&\overset{(b)}{=} \sigma_{\min}\left(V_X^T U_t\right),
\end{aligned}
$$

where $(a)$ and $(b)$ follow from the definition of $W_t$. Hence, in order to show that $\sigma_{\min}(U_t W_t) \geq \frac{\sigma_{\min}(X)}{\sqrt{10}}$ it suffices to show that $\sigma_{\min}\left(V_X^T U_t\right) \geq \frac{\sigma_{\min}(X)}{\sqrt{10}}$. For that, we will use the next lemma, which shows that $\sigma_{\min}\left(V_X^T U_t\right)$ grows exponentially.

**Lemma 9.1.** *Assume that* $\mu \leq c\|X\|^{-2}\kappa^{-2}$, $\|U_t\| \leq 3\|X\|$, *and that* $\|V_{X^\perp}^T V_{U_t W_t}\| \leq c\kappa^{-1}$. *Moreover, suppose that*

$$\|\left(\mathcal{A}^*\mathcal{A} - Id\right)\left(XX^T - U_t U_t^T\right)\| \leq c\sigma_{\min}^2(X). \tag{43}$$

*Furthermore, assume that* $V_X^T U_t$ *has full rank. Then it holds that*

$$\sigma_{\min}\left(V_X^T U_{t+1}\right) \geq \sigma_{\min}\left(V_X^T U_{t+1} W_t\right) \geq \sigma_{\min}\left(V_X^T U_t\right)\left(1 + \frac{1}{4}\mu\sigma_{\min}^2(X) - \mu\sigma_{\min}^2\left(V_X^T U_t\right)\right).$$

*Here* $c > 0$ *is constant, which is chosen small enough.*

The next lemma will allow us to show that the noise term $\|U_t W_{t,\perp}\|$ is growing slower than $\sigma_{\min}\left(V_X^T U_{t+1}\right)$.

**Lemma 9.2.** *Assume that* $\mu \leq c\min\left\{\|X\|^{-2}; \|\left(\mathcal{A}^*\mathcal{A} - Id\right)\left(XX^T - U_t U_t^T\right)\|^{-1}\right\}$ *and that* $\|U_t\| \leq 3\|X\|$. *Moreover, suppose that* $V_X^T U_{t+1} W_t$ *has full rank and that* $\|V_{X^\perp}^T V_{U_t W_t}\| \leq c\kappa^{-1}$. *Then it holds that*

$$\|U_{t+1} W_{t+1,\perp}\|$$
$$\leq \left(1 - \frac{\mu}{2}\|U_t W_{t,\perp}\|^2 + 9\mu\|V_{X^\perp}^T V_{U_t W_t}\|\|X\|^2 + 2\mu\|\left(\mathcal{A}^*\mathcal{A} - Id\right)\left(XX^T - U_t U_t^T\right)\|\right)\|U_t W_{t,\perp}\|.$$

*Here,* $c > 0$ *is an absolute constant chosen small enough.*

The next lemma shows that the angle between the column space of the signal term $U_t W_t$ and column space of $X$ stays sufficiently small.

**Lemma 9.3.** *Assume that* $\|U_t W_{t,\perp}\| \leq 2\sigma_{\min}(U_t W_t)$ *and* $\|U_t\| \leq 3\|X\|$ *holds. Moreover, assume that*

$$\|\left(Id - \mathcal{A}^*\mathcal{A}\right)\left(XX^T - U_t U_t^T\right)\| \leq c\sigma_{\min}^2(X), \tag{44}$$

$$\|V_{X^\perp}^T V_{U_t W_t}\| \leq c, \tag{45}$$

$$\mu \leq c\kappa^{-2}\|X\|^{-2}, \tag{46}$$

$$\|U_t W_{t,\perp}\| \leq c\kappa^{-2}\|X\|. \tag{47}$$

*where $c > 0$ is a small enough constant. Then it holds that*

$$\|V_{X^\perp}^T V_{U_{t+1} W_{t+1}}\|$$
$$\leq \left(1 - \frac{\mu}{4}\sigma_{\min}^2(X)\right)\|V_{X^\perp}^T V_{U_t W_t}\| + 100\mu\|(Id - \mathcal{A}^*\mathcal{A})(XX^T - U_t U_t^T)\| + 500\mu^2\|XX^T - U_t U_t^T\|^2.$$

The next lemma will show that we have $\|U_t\| \leq 3\|X\|$ for all $t$, a technical assumption which is needed in the above lemmas.

**Lemma 9.4.** *Assume that $\|U_t\| \leq 3\|X\|$, $\mu \leq \frac{1}{27\|X\|^2}$, and*

$$\|(\mathcal{A}^*\mathcal{A} - Id)(XX^T - U_t U_t^T)\| \leq \|X\|^2.$$

*Then it also holds that $\|U_{t+1}\| \leq 3\|X\|$.*

With these lemmas in place, we will be able to show that $\sigma_{\min}(U_t W_t) \geq \frac{\sigma_{\min}(X)}{\sqrt{10}}$ holds after sufficiently many iterations. Hence, we can enter Phase III, the *local refinement phase*.

The next lemma is concerned with this third phase. It shows that $U_t W_t W_t^T U_t^T$ converges towards $XX^T$, when projected onto the column space of $X$. We are going to provide a somewhat more general version of the lemma than what is needed in the proofs of our main results, since it may be of independent interest. For that, let $\|\cdot\|$ be a matrix norm, which satisfies $\|ABC\| \leq \|A\|\|B\|\|C\|$ for all matrices $A, B, C$. Furthermore, we assume that $\|A\| = \|A^T\|$ for all matrices $A$. For example, this property is fulfilled by all Schatten-$p$ norms.

**Lemma 9.5.** *Assume that $\|U_t\| \leq 3\|X\|$ and that $\sigma_{\min}(U_t W_t) \geq \frac{1}{\sqrt{10}}\sigma_{\min}(X)$. Moreover, assume that $\mu \leq c\kappa^{-2}\|X\|^{-2}$, $\|V_{X^\perp}^T V_{U_t W_t}\| \leq c\kappa^{-2}$, and*

$$\max\left\{\|(Id - \mathcal{A}^*\mathcal{A})(XX^T - U_t U_t^T)\|, \|(Id - \mathcal{A}^*\mathcal{A})(XX^T - U_t U_t^T)\|\right\} \leq c\kappa^{-2}\|XX^T - U_t U_t^T\|, \tag{48}$$

*where the constant $c > 0$ is chosen small enough. Then it holds that*

$$\|V_X^T(XX^T - U_{t+1}U_{t+1}^T)\| \leq \left(1 - \frac{\mu}{200}\sigma_{\min}(X)^2\right)\|V_X^T(XX^T - U_t U_t^T)\| + \mu\frac{\sigma_{\min}^2(X)}{100}\|U_t W_{t,\perp} W_{t,\perp}^T U_t^T\|.$$

When applying this lemma in our proof, we are going to set $\|\cdot\| = \|\cdot\|_F$. However, we believe that this lemma might be of independent interest, as it shows that $U_t U_t^T$ converges linearly towards $XX^T$ with respect to several different norms.

Having collected all the necessary ingredients, we can state and prove the main theorem of this section.

**Theorem 9.6.** *Let $\{U_t\} \subset \mathbb{R}^{d \times r}$ be the sequence created by the gradient descent algorithm. Assume that $\mu \leq c_1\kappa^{-4}\|X\|^{-2}$ for a sufficiently small constant $c_1$. Moreover, assume that $\mathcal{A}$ satisfies the restricted isometry property for rank-$(2r_\star + 1)$ matrices with constant $\delta \leq c_1\kappa^{-4}/\sqrt{r_\star}$. Let $\gamma > 0$ and choose the iteration count $t_\star$ such that $\sigma_{\min}(U_{t_\star} W_{t_\star}) \geq \gamma$. Furthermore, assume that the following conditions hold:*

$$\|U_{t_\star} W_{t_\star,\perp}\| \leq 2\gamma, \tag{49}$$

$$\|U_{t_\star}\| \leq 3\|X\|, \tag{50}$$

$$\gamma \leq c_2\frac{\sigma_{\min}(X)}{\min\{r;n\}\kappa^2}, \tag{51}$$

$$\|V_{X^\perp}^T V_{U_{t_\star} W_{t_\star}}\| \leq c_2\kappa^{-2}. \tag{52}$$

*Then after*

$$\hat{t} - t_\star \lesssim \frac{1}{\mu\sigma_{\min}(X)^2}\ln\left(\max\left\{1; \frac{\kappa r_\star}{\min\{r;n\} - r_\star}\right\}\frac{\|X\|}{\gamma}\right) \tag{53}$$

*iterations it holds that*

$$\frac{\|U_{\hat{t}} U_{\hat{t}}^T - XX^T\|_F}{\|X\|^2} \lesssim \frac{r_\star^{1/8}(\min\{r;n\} - r_\star)^{3/8}}{\kappa^{3/16}} \cdot \frac{\gamma^{21/16}}{\|X\|^{21/16}}.$$

**Remark 9.7.** *The proof of Theorem 9.6 shows that the number of iterations needed to complete Phase II is smaller than*

$$t_1 - t_\star \lesssim \frac{1}{\mu \sigma_{\min}^2(X)} \ln\left(\frac{\sigma_{\min}(X)}{\gamma}\right)$$

*and that the number of iterations needed to complete Phase III is smaller than*

$$\hat{t} - t_1 \lesssim \frac{1}{\mu \sigma_{\min}^2(X)} \ln\left(\max\left\{1; \frac{\kappa r_\star}{\min\{r; n\} - r_\star}\right\} \frac{\|X\|}{\gamma}\right).$$

*Proof of Theorem 9.6.* **Phase II:** In this phase, we will prove that $\sigma_{\min}\left(V_X^T U_t\right)$ is growing exponentially until it is at larger than $\frac{\sigma_{\min}(X)}{\sqrt{10}}$, while $\|U_t W_{t,\perp}\|$ stays grows much slower. For that, set

$$t_1 := \min\left\{t \geq t_\star : \sigma_{\min}\left(V_X^T U_t\right) \geq \frac{\sigma_{\min}(X)}{\sqrt{10}}\right\}.$$

We will prove by induction that for $t_\star \leq t \leq t_1$ the following inequalities hold:

$$\sigma_{\min}\left(V_X^T U_t\right) \geq \frac{1}{2}\left(1 + \frac{1}{8}\mu \sigma_{\min}^2(X)\right)^{t-t_\star} \gamma, \tag{54}$$

$$\|U_t W_{t,\perp}\| \leq 2\left(1 + 80\mu c_2 \sigma_{\min}^2(X)\right)^{t-t_\star} \gamma, \tag{55}$$

$$\|U_t\| \leq 3\|X\|, \tag{56}$$

$$\|V_{X^\perp}^T V_{U_t W_t}\| \leq c_2 \kappa^{-2}. \tag{57}$$

Note that when the inequalities above hold, then from the definition of $t_1$ above and inequality (54) we can derive that

$$t_1 - t_\star \leq \frac{16}{\mu \sigma_{\min}^2(X)} \ln\left(\sqrt{\frac{5}{2}} \cdot \frac{\sigma_{\min}(X)}{\gamma}\right). \tag{58}$$

For $t = t_\star$, we first note that inequalities (55), (56), and (57) follow directly from our assumptions. In order to prove inequality (54) we note that

$$\sigma_{\min}\left(V_X^T U_{t_\star}\right) \geq \sigma_{\min}\left(V_X^T V_{U_{t_\star} W_{t_\star}}\right) \sigma_{\min}\left(U_{t_\star} W_{t_\star}\right) \overset{(a)}{\geq} \frac{1}{2}\sigma_{\min}\left(U_{t_\star} W_{t_\star}\right) \overset{(b)}{\geq} \frac{\gamma}{2},$$

where inequality $(a)$ is a consequence of assumption (52) and inequality $(b)$ follows from the definition of $\gamma$. Assume now that we have shown these four inequalities for $t$. In order to prove them for $t + 1$ we note first that

$$\begin{aligned}
&\left\|\left(\mathcal{A}^*\mathcal{A} - \mathrm{Id}\right)\left(XX^T - U_t U_t^T\right)\right\| \\
&\overset{(a)}{\leq} \left\|\left(\mathcal{A}^*\mathcal{A} - \mathrm{Id}\right)\left(XX^T - U_t W_t W_t^T U_t^T\right)\right\| + \left\|\left(\mathcal{A}^*\mathcal{A} - \mathrm{Id}\right)\left(U_t W_{t,\perp} W_{t,\perp}^T U_t^T\right)\right\| \\
&\overset{(b)}{\leq} \delta\sqrt{r_\star}\|XX^T - U_t W_t W_t^T U_t^T\| + \delta\|U_t W_{t,\perp} W_{t,\perp}^T U_t^T\|_* \\
&\leq \delta\sqrt{r_\star}\left(\|X\|^2 + \|U_t W_t\|^2\right) + \delta\|U_t W_{t,\perp} W_{t,\perp}^T U_t^T\|_* \\
&\overset{(c)}{\leq} 10\delta\sqrt{r_\star}\|X\|^2 + \delta\left(\min\{r; n\} - r_\star\right)\|U_t W_{t,\perp}\|^2 \\
&\overset{(d)}{\leq} 10c_1 \kappa^{-2} \sigma_{\min}(X)^2 + 4\delta\left(\min\{r; n\} - r_\star\right)\left(1 + 80\mu c_2 \sigma_{\min}(X)^2\right)^{2(t-t_\star)} \gamma^2 \\
&\overset{(e)}{\leq} 10c_1 \kappa^{-2} \sigma_{\min}(X)^2 + 8\delta\left(\min\{r; n\} - r_\star\right)\sigma_{\min}(X)^{1/4}\gamma^{7/4} \\
&\overset{(f)}{\leq} 40c_1 \kappa^{-2} \sigma_{\min}(X)^2.
\end{aligned} \tag{59}$$

In inequality $(a)$ we applied the triangle inequality and for inequality $(b)$ we used the restricted isometry property as well as Lemma 7.3. Inequality $(c)$ is due to the induction assumption (56). In inequality $(d)$ we used the assumption $\delta \leq c_1 \kappa^{-4}$ as well as the induction assumption (55). For inequality $(e)$ we used $t \leq t_1$ as well as (58) and for inequality $(f)$ we used (51).

Next, we observe that by Lemma 9.1 we have that

$$
\begin{aligned}
\sigma_{\min}\left(V_X^T U_{t+1} W_{t+1}\right) &= \sigma_{\min}\left(V_X^T U_{t+1}\right) \\
&\geq \sigma_{\min}\left(V_X^T U_{t+1} W_t\right) \\
&\geq \sigma_{\min}\left(V_X^T U_t\right)\left(1 + \frac{1}{4}\mu\sigma_{\min}^2(X) - \mu\sigma_{\min}^2\left(V_X^T U\right)\right) \\
&\overset{(a)}{\geq} \sigma_{\min}\left(V_X^T U_t\right)\left(1 + \frac{1}{8}\mu\sigma_{\min}^2(X)\right).
\end{aligned}
$$

In $(a)$ we have used that $\sigma_{\min}\left(V_X^T U_t\right) \leq \frac{\sigma_{\min}(X)}{\sqrt{10}}$, which follows from $t \leq t_1$. Using the induction assumption, this implies inequality (54). Moreover, the inequality chain above shows that $V_X^T U_{t+1} W_{t+1}$ has full rank. This allows us to apply Lemma 9.2, which implies that

$$
\begin{aligned}
\|U_{t+1} W_{t+1,\perp}\| &\leq \left(1 - \frac{\mu}{2}\|U_t W_{t,\perp}\|^2 + 9\mu\|V_{X^\perp}^T V_{U_t W}\|\|X\|^2 + 2\mu\|\left(\mathcal{A}^*\mathcal{A} - \mathrm{Id}\right)\left(XX^T - U_t U_t^T\right)\|\right)\|U_t W_{t,\perp}\| \\
&\overset{(a)}{\leq} \left(1 + 80\mu c_2 \sigma_{\min}^2(X)\right)\|U_t W_{t,\perp}\| \\
&\leq 2\left(1 + 80\mu c_2 \sigma_{\min}^2(X)\right)^{t+1-t_\star}\gamma,
\end{aligned}
$$

where in inequality $(a)$ we used (57) as well as (59) and that the constant $c_1$ is chosen sufficiently small. This shows inequality (55). Next, due to inequality (59), our induction assumptions, and Lemma 9.4 we obtain that $\|U_{t+1}\| \leq 3\|X\|$, which shows inequality (56).

Next, we note that by Lemma 9.3 we have that

$$
\begin{aligned}
&\|V_{X^\perp}^T V_{U_{t+1} W_{t+1}}\| \\
&\leq \left(1 - \frac{\mu}{4}\sigma_{\min}^2(X)\right)\|V_{X^\perp}^T V_{U_t W_t}\| + 100\mu\|\left(\mathrm{Id} - \mathcal{A}^*\mathcal{A}\right)\left(XX^T - U_t U_t^T\right)\| + 500\mu^2\|XX^T - U_t U_t^T\|^2 \\
&\overset{(a)}{\leq} \left(1 - \frac{\mu}{4}\sigma_{\min}^2(X)\right)\|V_{X^\perp}^T V_{U_t W_t}\| + 2000 c_1 \mu\kappa^{-2}\sigma_{\min}(X)^2 + 50000\mu^2\|X\|^4 \\
&\overset{(b)}{\leq} \left(1 - \frac{\mu}{4}\sigma_{\min}^2(X)\right)\|V_{X^\perp}^T V_{U_t W_t}\| + 2000 c_1 \mu\kappa^{-2}\sigma_{\min}(X)^2 + 50000 c_1 \mu\kappa^{-2}\sigma_{\min}^2(X) \\
&\overset{(c)}{\leq} \left(1 - \frac{\mu}{4}\sigma_{\min}^2(X)\right)c_2\kappa^{-2} + 2000 c_1 \mu\kappa^{-2}\sigma_{\min}(X)^2 + 50000 c_1 \mu\kappa^{-2}\sigma_{\min}^2(X),
\end{aligned}
$$

where in inequality $(a)$ we used the induction hypothesis (56) as well as (59). Inequality $(b)$ follows from inequality (59) and our assumption on the step size $\mu$. In inequality $(c)$ we used the induction assumption $\|V_{X^\perp}^T V_{U_t W_t}\| \leq c_2\kappa^{-2}$. By choosing the constant $c_1 > 0$ small enough, this implies inequality (57) and, hence, finishes the induction step.

Note that from the definition of $t_1$ and from inequality (54) the inequality (58) follows. Hence, we obtain that

$$
\begin{aligned}
\|U_{t_1} W_{t_1,\perp}\| &\overset{(a)}{\leq} 2\left(1 + 80\mu c_2 \sigma_{\min}^2(X)\right)^{t_1-t_\star}\gamma \\
&\overset{(b)}{\leq} 2\left(\sigma_{\min}(X)/\gamma\right)^{80 c_2}\gamma \\
&\overset{(c)}{\leq} 2\left(\frac{\sigma_{\min}(X)}{\gamma}\right)^{1/8}\gamma \\
&= 2\sigma_{\min}(X)^{1/8}\gamma^{7/8},
\end{aligned}
\tag{60}
$$

where inequality $(a)$ follows from inequality (55) and inequality $(b)$ follows from (58). Inequality $(c)$ follows from choosing $c_2 > 0$ small enough. This finishes the proof of the second phase.

**Phase III:** In the third phase, we analyse the refinement of the signal $U_t$. For that, we set

$$
\hat{t} := t_1 + \left\lfloor \frac{300}{\mu\sigma_{\min}(X)^2}\ln\left(\frac{5}{8}\kappa^{1/4}\sqrt{\frac{r_\star}{\min\{r;n\} - r_\star}}\frac{\|X\|^{7/4}}{\gamma^{7/4}}\right)\right\rfloor.
\tag{61}
$$

Similar as in Phase II, we are going to show inductively that the following inequalities are fulfilled for $t_1 \le t \le \hat{t}$

$$\sigma_{\min}(U_t W_t) \ge \sigma_{\min}(V_X^T U_t) \ge \frac{\sigma_{\min}(X)}{\sqrt{10}}, \tag{62}$$

$$\|U_t W_{t,\perp}\| \le \left(1 + 80\mu c_2 \sigma_{\min}^2(X)\right)^{t-t_1} \|U_{t_1} W_{t_1,\perp}\|, \tag{63}$$

$$\|U_t\| \le 3\|X\|, \tag{64}$$

$$\|V_{X^\perp}^T V_{U_t W_t}\| \le c_2 \kappa^{-2}, \tag{65}$$

$$\|V_X^T \left(XX^T - U_t U_t^T\right)\|_F \le 10\sqrt{r_\star} \left(1 - \frac{\mu}{400}\sigma_{\min}^2(X)\right)^{t-t_1} \|X\|^2. \tag{66}$$

For $t = t_1$ we note the inequalities (62), (64), and (65) follow from the results in Phase 1. The inequality (63) follows directly from setting $t = t_1$. For $t = t_1$, inequality (66) follows from the observation that

$$\begin{aligned}
\|V_X^T \left(XX^T - U_{t_1} U_{t_1}^T\right)\|_F &= \|V_X^T \left(XX^T - U_{t_1} W_{t_1} W_{t_1}^T U_{t_1}^T\right)\|_F \\
&\le \|XX^T\|_F + \|U_{t_1} W_{t_1} W_{t_1}^T U_{t_1}^T\|_F \\
&\le \sqrt{r_\star} \left(\|XX^T\| + \|U_{t_1} W_{t_1} W_{t_1}^T U_{t_1}^T\|\right) \\
&\le 10\sqrt{r_\star}\|X\|^2,
\end{aligned}$$

where we have used that $\|U_{t_1} W_{t_1}\| \le \|U_{t_1}\| \le 3\|X\|$ by induction assumption (64).

For the induction step from $t$ to $t + 1$, we note first that with similar arguments as in Phase 1 we can show that

$$\| (\mathcal{A}^*\mathcal{A} - \mathrm{Id}) \left(XX^T - U_t U_t^T\right) \|$$

$$\le 10\delta\sqrt{r_\star}\|X\|^2 + \delta\left(\min\{r; n\} - r_\star\right)\|U_t W_{t,\perp}\|^2$$

$$\overset{(a)}{\le} 10 c_1 \kappa^{-2}\sigma_{\min}(X)^2 + 4\delta\left(\min\{r; n\} - r_\star\right)\left(1 + 80\mu c_2 \sigma_{\min}(X)^2\right)^{2(t-t_1)}\sigma_{\min}(X)^{1/4}\gamma^{7/4}$$

$$\overset{(b)}{\le} 10 c_1 \kappa^{-2}\sigma_{\min}(X)^2 + 4\delta\left(\min\{r; n\} - r_\star\right)\left(\frac{5}{8}\kappa^{1/4}\sqrt{\frac{r_\star}{\min\{r; n\} - r_\star}}\frac{\|X\|^{7/4}}{\gamma^{7/4}}\right)^{O(c_2)}\sigma_{\min}(X)^{1/4}\gamma^{7/4}$$

$$\overset{(c)}{\le} 40 c_1 \kappa^{-2}\sigma_{\min}(X)^2,$$

where in inequality $(a)$ follows from (63). Inequality $(b)$ follows from (61) as well as the elementary inequality $\ln(1 + x) \le x$. Inequality $(c)$ follows from the assumption $\gamma \le c_2 \frac{\sigma_{\min}(X)}{\min\{r; n\}\kappa}$. This puts us in a position to apply our technical lemmas. We note that by Lemma 9.1 we have that

$$\sigma_{\min}(U_{t+1} W_{t+1}) \ge \sigma_{\min}(V_X^T U_{t+1}) \ge \sigma_{\min}(V_X^T U_{t+1} W_t)$$

$$\ge \sigma_{\min}(V_X^T U_t) \underbrace{\left(1 + \frac{1}{4}\mu\sigma_{\min}(X)^2 - \mu\sigma_{\min}(V_X^T U_t)^2\right)}_{=(*)}.$$

Note that for $\sigma_{\min}(V_X^T U_t) \le \frac{1}{2}\sigma_{\min}(X)$ it holds that $(*) \ge 1$ and thus it follows that (62) holds for $t + 1$ in this case. In the case of $\frac{1}{2}\sigma_{\min}(X) \le \sigma_{\min}(V_X^T U_t)$ we obtain that

$$(*) \ge 1 - \mu\sigma_{\min}(V_X^T U_t)^2 \overset{(a)}{\ge} 1 - 9\mu\|X\|^2 \overset{(b)}{\ge} 4/5,$$

where in inequality $(a)$ we used the induction hypothesis (56) and in inequality $(b)$ we used the assumption $\mu \le c_1 \kappa^{-2}\|X\|^{-2}$. Hence, we have shown that also in this case the inequality (62) holds for $t + 1$.

Note that the previous inequality chain also implies that $V_X^T U_{t+1} W_t$ is invertible. Hence, in a similar way as in Phase II for inequality (55) we can verify that (63) holds for $t + 1$.

Note that from Lemma 9.4, induction assumption (56), and the assumption on the step size $\mu$ it follows that $\|U_{t+1}\| \le 3\|X\|$. Moreover, inequality (65) can be shown analogously as in Phase 1. Next, we note that due to the restricted isometry property we have that

$$\| (\mathrm{Id} - \mathcal{A}^*\mathcal{A}) \left(XX^T - U_t U_t^T\right) \|_F \le c_1 \kappa^{-2}\|XX^T - U_t U_t^T\|_F,$$

which shows that inequality (48) is fulfilled (with $\|\cdot\|$ being the Frobenius norm $\|\cdot\|_F$). Hence, we obtain from Lemma 9.5 that

$$\|V_X^T \left( XX^T - U_{t+1}U_{t+1}^T \right)\|_F$$

$$\leq \left(1 - \frac{\mu}{200}\sigma_{\min}(X)^2\right) \|V_X^T \left( XX^T - U_t U_t^T \right)\|_F + \mu\frac{\sigma_{\min}(X)^2}{100}\|U_t W_{t,\perp}W_{t,\perp}^T U_t^T\|_F$$

$$\leq 10\sqrt{r_\star}\left(1 - \frac{\mu}{200}\sigma_{\min}(X)^2\right)\left(1 - \frac{\mu}{400}\sigma_{\min}(X)^2\right)^{t-t_1}\|X\|^2 + \mu\frac{\sigma_{\min}(X)^2}{100}\|U_t W_{t,\perp}W_{t,\perp}^T U_t^T\|_F,$$

where in the last inequality we used the induction assumption (66). We note that this shows (66) holds for $t+1$, if we can show that

$$\|U_t W_{t,\perp}W_{t,\perp}^T U_t^T\|_F \leq \frac{5}{2}\sqrt{r_\star}\left(1 - \frac{\mu}{400}\sigma_{\min}(X)^2\right)^{t-t_1}\|X\|^2. \tag{67}$$

For that, we note that

$$\|U_t W_{t,\perp}W_{t,\perp}^T U_t^T\|_F \leq \sqrt{\min\{r;n\} - r_\star}\|U_t W_{t,\perp}\|^2$$

$$\leq 4\sqrt{\min\{r;n\} - r_\star}\left(1 + 80\mu c_2\sigma_{\min}(X)^2\right)^{2(t-t_1)}\sigma_{\min}(X)^{1/4}\gamma^{7/4},$$

where in the last inequality we used (60) and (63). Hence, for $c_2 > 0$ small enough, inequality (67) is implied by

$$\frac{8}{5}\sqrt{\frac{\min\{r;n\} - r_\star}{r_\star}}\sigma_{\min}(X)^{1/4}\gamma^{7/4} \leq \left(1 - \frac{\mu}{350}\sigma_{\min}(X)^2\right)^{t-t_1}\|X\|^2.$$

By rearranging terms and using the elementary inequality $\ln(1+x) \geq \frac{x}{1-x}$, we see that this in turn is implied by

$$t - t_1 \leq \frac{300}{\mu\sigma_{\min}(X)^2}\ln\left(\frac{5}{8}\sqrt{\frac{r_\star}{\min\{r;n\} - r_\star}} \cdot \frac{\|X\|^2}{\gamma^{7/4}\sigma_{\min}(X)^{1/4}}\right).$$

Hence, (61) shows (67), which shows inequality (66) for $t+1$. This finishes the induction step.

**Conclusion:** In order to finish the proof we note that

$$\|U_{\hat{t}}U_{\hat{t}}^T - XX^T\|_F \overset{(a)}{\leq} 4\|V_X^T\left(XX^T - U_{\hat{t}}U_{\hat{t}}^T\right)\|_F + \|U_{\hat{t}}W_{\hat{t},\perp}W_{\hat{t},\perp}^T U_{\hat{t}}^T\|_F$$

$$\overset{(b)}{\lesssim} \sqrt{r_\star}\left(1 - \frac{\mu}{400}\sigma_{\min}(X)^2\right)^{\hat{t}-t_1}\|X\|^2$$

$$\overset{(c)}{\lesssim} \sqrt{r_\star}\left(\frac{5}{8}\kappa^{1/4}\sqrt{\frac{r_\star}{\min\{r;n\} - r_\star}}\frac{\|X\|^{7/4}}{\gamma^{7/4}}\right)^{-3/4}\|X\|^2$$

$$\lesssim r_\star^{1/8}\kappa^{-3/16}\left(\min\{r;n\} - r_\star\right)^{3/8}\gamma^{21/16}\|X\|^{11/16},$$

where inequality $(a)$ follows from the triangle inequality and the definition of $W_t$. Inequality $(b)$ follows from (66) and inequality (67). In $(c)$ we used the definition of $\hat{t}$.

In order to finish the proof we need to show (53). For that we note that

$$t - t_1 \leq \frac{300}{\mu\sigma_{\min}(X)^2}\ln\left(\frac{5}{8}\sqrt{\frac{r_\star}{\min\{r;n\} - r_\star}} \cdot \frac{\|X\|^2}{\gamma^{7/4}\sigma_{\min}(X)^{1/4}}\cdot\right)$$

$$= \frac{300}{\mu\sigma_{\min}(X)^2}\ln\left(\frac{5}{8}\kappa^{1/4}\sqrt{\frac{r_\star}{\min\{r;n\} - r_\star}}\frac{\|X\|^{7/4}}{\gamma^{7/4}}\right)$$

$$\leq \frac{300}{\mu\sigma_{\min}(X)^2}\ln\left(\min\left\{1; \frac{\kappa r_\star}{\min\{r;n\} - r_\star}\right\}\frac{\|X\|^{7/4}}{\gamma^{7/4}}\right)$$

$$\lesssim \frac{1}{\mu\sigma_{\min}(X)^2}\ln\left(\min\left\{1; \frac{\kappa r_\star}{\min\{r;n\} - r_\star}\right\}\frac{\|X\|}{\gamma}\right).$$

Combining this with inequality (58) shows (53). $\qquad\square$

# 10 Proof of the main results

## 10.1 Proof of Theorem 3.3

*Proof of Theorem 3.3.* Set $\tilde{E} = \mathcal{A}^*\mathcal{A}\left(XX^T\right) - XX^T$. From the spectral-to-spectral restricted isometry property, which follows from Lemma 7.3 as well as from our assumption on the restricted isometry property, it follows that

$$\|\tilde{E}\| = \|\left(\mathrm{Id} - \mathcal{A}^*\mathcal{A}\right)\left(XX^T\right)\| \le c\kappa^{-4}\|X\|^2 = c\kappa^{-2}\sigma_{\min}\left(X\right)^2. \tag{68}$$

In order to finish the proof we will distinguish two cases:

**Case $r \ge 2r_\star$:** Due to (4) and (68) we can apply Lemma 8.7. Hence, with probability at least $1 - O\left(\exp\left(-cr\right)\right)$ after

$$t_\star \lesssim \frac{1}{\mu\sigma_{\min}\left(X\right)^2} \cdot \ln\left(2\kappa^2\sqrt{\frac{n}{\min\{r;n\}}}\right)$$

iterations we have that

$$\|U_{t_\star}\| \le 3\|X\|, \tag{69}$$

$$\sigma_{\min}\left(U_{t_\star}W_{t_\star}\right) \ge \frac{\alpha\beta}{4}, \tag{70}$$

$$\|U_{t_\star}W_{t_\star,\perp}\| \le \frac{\kappa^{-2}}{8}\alpha\beta, \tag{71}$$

$$\|V_{X^\perp}^T V_{U_{t_\star}W_{t_\star}}\| \le c\kappa^{-2} \tag{72}$$

with $1 \lesssim \beta \lesssim \frac{n\kappa^4}{\min\{r;n\}}$. Our goal is to apply Theorem 9.6 with $\gamma = \frac{\alpha\beta}{4}$. For that we need to check that

$$\frac{c_2\sigma_{\min}\left(X\right)}{\min\{r;n\}\kappa^2} \ge \gamma = \frac{\alpha\beta}{4} \tag{73}$$

holds. Note that since

$$\frac{4c_2\|X\|}{\min\{r;n\}\kappa^3\beta} \gtrsim \frac{\|X\|}{\kappa^7 n} \gtrsim \alpha$$

condition (73) is fulfilled, when the constant in (4) is chosen sufficiently small. Hence, by Theorem 9.6 after

$$\hat{t} - t_\star \lesssim \frac{1}{\mu\sigma_{\min}\left(X\right)^2}\ln\left(\max\left\{1; \frac{\kappa r_\star}{\min\{r;n\} - r_\star}\right\}\frac{4\|X\|}{\alpha\beta}\right)$$

iterations it holds that

$$\begin{aligned}
\frac{\|U_{\hat{t}}U_{\hat{t}}^T - XX^T\|_F}{\|X\|^2} &\lesssim \frac{r_\star^{1/8}\left(\min\{r;n\} - r_\star\right)^{3/8}}{\kappa^{3/16}} \cdot \frac{\gamma^{21/16}}{\|X\|^{21/16}} \\
&\lesssim \frac{r_\star^{1/8}\left(\min\{r;n\} - r_\star\right)^{3/8}}{\kappa^{3/16}}\frac{\left(\alpha\beta\right)^{21/16}}{\|X\|^{21/16}} \\
&\lesssim \frac{\kappa^{81/16}n^{21/16}r_\star^{1/8}\left(\min\{r;n\} - r_\star\right)^{3/8}}{\left(\min\{r;n\}\right)^{21/16}} \cdot \frac{\alpha^{21/16}}{\|X\|^{21/16}} \\
&\le \frac{n^{21/16}\kappa^{81/16}r_\star^{1/8}}{\left(\min\{r;n\}\right)^{15/16}} \cdot \frac{\alpha^{21/16}}{\|X\|^{21/16}}.
\end{aligned}$$

Note that for the total amount of iterations we have that

$$\hat{t} \lesssim \frac{1}{\mu\sigma_{\min}(X)^2}\left(\ln\left(2\kappa^2\sqrt{\frac{n}{\min\{r;n\}}}\right) + \ln\left(\max\left\{1;\frac{\kappa r_\star}{\min\{r;n\}-r_\star}\right\}\frac{4\|X\|}{\alpha\beta}\right)\right)$$

$$= \frac{1}{\mu\sigma_{\min}(X)^2}\ln\left(8\kappa^3\sqrt{\frac{n}{\min\{r;n\}}}\cdot\max\left\{1;\frac{\kappa r_\star}{\min\{r;n\}-r_\star}\right\}\cdot\frac{\|X\|}{\alpha\beta}\right)$$

$$\overset{(b)}{\leq} \frac{1}{\mu\sigma_{\min}(X)^2}\ln\left(C_1\kappa^3\sqrt{\frac{n}{\min\{r;n\}}}\cdot\max\left\{1;\frac{\kappa r_\star}{\min\{r;n\}-r_\star}\right\}\cdot\frac{\|X\|}{\alpha}\right)$$

$$\lesssim \frac{1}{\mu\sigma_{\min}(X)^2}\ln\left(\frac{C_1\kappa n}{\min\{r;n\}}\cdot\max\left\{1;\frac{\kappa r_\star}{\min\{r;n\}-r_\star}\right\}\cdot\frac{\|X\|}{\alpha}\right),$$

where in inequality $(b)$ we have used $\beta \gtrsim 1$ and chosen the constant $C_1 > 0$ large enough. This finishes the proof of the first part.

**Case $r_\star < r < 2r_\star$:** As in the first case, we can apply Lemma 8.7. Hence, with probability at least $1 - (C\varepsilon)^{r-r_\star+1} + O\left(\exp\left(-cr\right)\right)$ after

$$t_\star \lesssim \frac{1}{\mu\sigma_{\min}(X)^2}\cdot\ln\left(\frac{2\kappa^2\sqrt{rn}}{\varepsilon}\right)$$

iterations the inequalities (34), (35), (36), and (37) hold with $\frac{\varepsilon}{r} \lesssim \beta \lesssim \frac{\kappa^4 n}{\varepsilon}$. Again, we want to apply Theorem 9.6 with $\gamma = \frac{\alpha\beta}{4}$. For that we need to check that

$$\frac{c_2\sigma_{\min}(X)}{\min\{r;n\}\kappa^2} \geq \gamma = \frac{\alpha\beta}{4} \tag{74}$$

holds. Note that since

$$\frac{4c_2\|X\|}{\min\{r;n\}\kappa^3\beta} \gtrsim \frac{\varepsilon r\|X\|}{\min\{r;n\}n\kappa^7} = \frac{\varepsilon\|X\|}{n\kappa^7} \gtrsim \alpha$$

condition (74) holds true, when the constant in (7) is chosen sufficiently small. Hence, by Theorem 9.6 after

$$\hat{t} - t_\star \lesssim \frac{1}{\mu\sigma_{\min}(X)^2}\ln\left(\max\left\{1;\frac{\kappa r_\star}{r-r_\star}\right\}\frac{4\|X\|}{\alpha\beta}\right)$$

iterations it holds that

$$\frac{\|U_{\hat{t}}U_{\hat{t}}^T - XX^T\|_F}{\|X\|^2} \lesssim \frac{r_\star^{1/8}(r-r_\star)^{3/8}}{\kappa^{3/16}}\cdot\frac{\gamma^{21/16}}{\|X\|^{21/16}}$$

$$\lesssim \frac{r_\star^{1/8}(r-r_\star)^{3/8}}{\kappa^{3/16}}\cdot\frac{(\alpha\beta)^{21/16}}{\|X\|^{21/16}}$$

$$\leq r_\star^{1/8}(r-r_\star)^{3/8}\kappa^{81/16}\left(\frac{n}{\varepsilon}\cdot\frac{\alpha}{\|X\|}\right)^{21/16}.$$

Note that for the total amount of iterations we have that

$$\hat{t} \lesssim \frac{1}{\mu\sigma_{\min}(X)^2}\left(\ln\left(\frac{2\kappa^2\sqrt{rn}}{\varepsilon}\right) + \ln\left(\max\left\{1;\frac{\kappa r_\star}{r-r_\star}\right\}\frac{4\|X\|}{\alpha\beta}\right)\right)$$

$$\overset{(a)}{=} \frac{1}{\mu\sigma_{\min}(X)^2}\ln\left(\frac{8\kappa^3 r_\star\sqrt{rn}}{\varepsilon(r-r_\star)}\cdot\frac{\|X\|}{\alpha\beta}\right)$$

$$\overset{(b)}{\leq} \frac{1}{\mu\sigma_{\min}(X)^2}\ln\left(\frac{C_2\kappa^3 r_\star r\sqrt{rn}}{\varepsilon^2(r-r_\star)}\cdot\frac{\|X\|}{\alpha}\right)$$

$$\lesssim \frac{1}{\mu\sigma_{\min}(X)^2}\ln\left(\frac{C_2\kappa n^2}{\varepsilon^2(r-r_\star)}\cdot\frac{\|X\|}{\alpha}\right),$$

where in equality $(a)$ we have used $\frac{r_\star}{r-r_\star} \geq 1$, which follows from $r_\star < r \leq 2r_\star$. Inequality $(b)$ follows from $\frac{\varepsilon}{r} \lesssim \beta$ as well as from choosing $C_2 > 0$ large enough. This finishes the proof.

$\square$

## 10.2 Proof of Theorem 3.4

*Proof of Theorem 3.4.* As in the proof of the second part of Theorem 3.3 we can show that with probability at least $1 - C\varepsilon + O\left(\exp\left(-cr_\star\right)\right)$ after

$$t_\star \lesssim \frac{1}{\mu\sigma_{\min}\left(X\right)^2} \cdot \ln\left(\frac{2\kappa^2\sqrt{n}}{\varepsilon}\right)$$

iterations the inequalities (34), (35), (36), and (37) hold with $\frac{\varepsilon}{r_\star} \lesssim \beta \lesssim \frac{\kappa^4 n}{\varepsilon}$. Now define the matrix $\widehat{U}_{t_\star}$ by adding a zero column to $U_{t_\star}$, i.e.,

$$\widehat{U}_{t_\star} = \begin{pmatrix} U_{t_\star} & 0 \end{pmatrix} \in \mathbb{R}^{n\times(r_\star+1)}.$$

Clearly, we can run gradient descent on $\widehat{U}_{t_\star}$ instead of $U_{t_\star}$ with the same step size, which gives us a sequence $\widehat{U}_{t_\star}, \widehat{U}_{t_\star+1}, \widehat{U}_{t_\star+2}, \ldots$ to which we can apply Theorem 9.6 with $\gamma = \frac{\alpha\beta}{4}$ and $r = r_\star + 1$. However, note that the last column always stays zero, which means that the results of this theorem also apply to $U_{t_\star}, U_{t_\star+1}, U_{t_\star+2}, \ldots$. Hence, after

$$\hat{t} - t_\star \lesssim \frac{1}{\mu\sigma_{\min}\left(X\right)^2} \ln\left(4\kappa r_\star \frac{\|X\|}{\alpha\beta}\right) \lesssim \frac{1}{\mu\sigma_{\min}\left(X\right)^2} \ln\left(4\kappa r_\star^2 \frac{\|X\|}{\alpha\varepsilon}\right)$$

iterations it holds that

$$\frac{\|U_{\hat{t}}U_{\hat{t}}^T - XX^T\|_F}{\|X\|^2} \lesssim \frac{r_\star^{1/8}}{\kappa^{3/16}} \cdot \frac{(\alpha\beta)^{21/16}}{\|X\|^{21/16}}$$

$$\lesssim \frac{r_\star^{1/8}}{\kappa^{3/16}} \left(\frac{\kappa^4 n}{\varepsilon} \cdot \frac{\alpha}{\|X\|}\right)^{21/16}$$

$$= r_\star^{1/8} \kappa^{81/16} \left(\frac{n}{\varepsilon} \cdot \frac{\alpha}{\|X\|}\right)^{21/16}.$$

Note that for the total amount of iterations we have that

$$\hat{t} \lesssim \frac{1}{\mu\sigma_{\min}\left(X\right)^2} \left(\ln\left(\frac{2\kappa^2\sqrt{n}}{\varepsilon}\right) + \ln\left(4\kappa r_\star^2 \frac{\|X\|}{\alpha\varepsilon}\right)\right)$$

$$= \frac{1}{\mu\sigma_{\min}\left(X\right)^2} \ln\left(\frac{8\kappa^3 r_\star^2 \sqrt{n}\|X\|}{\alpha\varepsilon^2}\right)$$

$$\leq \frac{1}{\mu\sigma_{\min}\left(X\right)^2} \ln\left(\frac{8\kappa^3 n^3}{\varepsilon^2} \cdot \frac{\|X\|}{\alpha}\right).$$

This finishes the proof.

$\square$

## 10.3 Proof of Theorem 3.5

We start by noting that in the special case $r = n$, the required assumptions for Theorem 9.6 are already fulfilled at the initialization $t_0 = 0$. This means that in this special case we do not need to analyze the spectral phase. This is shown by the following lemma.

**Lemma 10.1.** *Assume that $r = n$ and let $U_0 = \alpha U$, where $U \in \mathbb{R}^{n\times n}$ is an orthonormal matrix. Then it holds that*

$$\|V_{X^\perp}^T V_{U_0 W_0}\| = 0,$$
$$\sigma_{\min}\left(U_0 W_0\right) = \alpha,$$
$$\|U_0\| = \alpha.$$

*Proof.* Note that $V_X^T U \in \mathbb{R}^{r_\star \times n}$ is an isometric embedding. Hence, a feasible choice for $W_0$ is given by $W_0 = U^T V_X$, which implies that

$$U_0 W_0 = \alpha U U^T V_X = \alpha V_X.$$

It follows that $\|V_{X^\perp}^T V_{U_0 W_0}\| = 0$, which verifies the first equality. In order to see that the second equality holds we note that

$$\sigma_{\min}(U_0 W_0) = \sigma_{\min}(\alpha V_X) = \alpha.$$

The third equality follows directly from the definition of $U_0$. This finishes the proof. $\qquad\square$

Now we are in a position to give a proof of Theorem 3.5.

*Proof of Theorem 3.5.* By Lemma 10.1 we have that $\|V_{X^\perp}^T V_{U_0 W_0}\| = 0$, $\sigma_{\min}(U_0 W_0) = \alpha$, and $\|U_0\| = \alpha$. This allows us to apply Theorem 9.6 with $t_0 = 0$ and $\gamma = \alpha$, which yields that after

$$\hat{t} \lesssim \frac{1}{\mu \sigma_{\min}(X)^2} \ln\left(\max\left\{1; \frac{\kappa r_\star}{n - r_\star}\right\} \frac{\|X\|}{\alpha}\right)$$

iterations we have that

$$\frac{\|U_{\hat{t}} U_{\hat{t}}^T - XX^T\|_F}{\|X\|^2} \lesssim \frac{r_\star^{1/8}(n - r_\star)^{3/8}}{\kappa^{3/16}} \cdot \frac{\alpha^{21/16}}{\|X\|^{21/16}}$$

$$\leq \frac{r_\star^{1/8} n^{3/8}}{\kappa^{3/16}} \cdot \frac{\alpha^{21/16}}{\|X\|^{21/16}}.$$

This finishes the proof. $\qquad\square$

## 11 Conclusion

In this paper we focused on demystifying the role of initialization when training overparameterized models by showing that small random initialization followed by a few iterations of gradient descent behaves akin to popular spectral methods. We also show that this *implicit spectral bias* from small random initialization, which is provably more prominent for overparameterized models, also puts the gradient descent iterations on a particular trajectory towards solutions that are not only globally optimal but also generalize well.

We think that our results give rise to a number of interesting future research directions. For example, one could extend our results to scenarios where the measurement matrices are more structured such as in matrix completion [65] or in blind deconvolution [66]. Moreover, while our main results, e.g. Theorem 3.3 do require early stopping, our simulations (e.g. Figure 7a) indicate that early stopping is not needed. It would be interesting to examine whether we can remove the early stopping requirement. It is also an interesting future avenue to examine whether the quadratic dependence of the sample complexity $m$ on $r_\star$ in our results is really needed.

Moreover, while in this paper our main focus was on low-rank matrix reconstruction, we believe that our analysis holds more generally for a variety of contemporary overparameterized machine learning and signal estimation tasks including neural network training. This is a tantalizing future research direction.

## Acknowledgments and Disclosure of Funding

M.S. is supported by the Packard Fellowship in Science and Engineering, a Sloan Research Fellowship in Mathematics, an NSF-CAREER under award #1846369, the Air Force Office of Scientific Research Young Investigator Program (AFOSR-YIP) under award #FA9550-18-1-0078, DARPA Learning with Less Labels (LwLL) and FastNICS programs, and NSF-CIF awards #1813877 and #2008443.

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

# A Proofs for the spectral phase

## A.1 Proof of Lemma 8.1

*Proof of Lemma 8.1.* We are first going to derive a formula for $\tilde{U}_t - U_t$.

**Claim:** Set $\hat{E}_i := \mu \mathcal{A}^* \mathcal{A} \left( U_{i-1} U_{i-1}^T \right) U_{i-1}$. Then, for $t \geq 1$ it holds that

$$\tilde{U}_t - U_t = \sum_{i=1}^{t} \left( \mathrm{Id} + \mu \mathcal{A}^* \mathcal{A} \left( X X^T \right) \right)^{t-i} \hat{E}_i. \tag{75}$$

**Proof of the claim:** We will prove the claim by induction. For $t = 1$ we note that

$$\begin{aligned}
U_1 &= \left( \mathrm{Id} + \mu \mathcal{A}^* \mathcal{A} \left( X X^T - U_0 U_0^T \right) \right) U_0 \\
&= \left( \mathrm{Id} + \mu \mathcal{A}^* \mathcal{A} \left( X X^T \right) \right) U_0 - \mu \mathcal{A}^* \mathcal{A} \left( U_0 U_0^T \right) U_0 \\
&= \tilde{U}_1 - \hat{E}_1,
\end{aligned}$$

which proves the claim for $t = 1$. Now suppose that the claim holds for some $t$. We obtain that

$$\begin{aligned}
U_{t+1} &= \left( \mathrm{Id} + \mu \mathcal{A}^* \mathcal{A} \left( X X^T - U_t U_t^T \right) \right) U_t \\
&= \left( \mathrm{Id} + \mu \mathcal{A}^* \mathcal{A} \left( X X^T \right) \right) U_t - \mu \mathcal{A}^* \mathcal{A} \left( U_t U_t^T \right) U_t \\
&= \left( \mathrm{Id} + \mu \mathcal{A}^* \mathcal{A} \left( X X^T \right) \right) U_t - \hat{E}_{t+1},
\end{aligned}$$

where the last line follows from the definition of $\hat{E}_{t+1}$. By using the induction hypthesis we obtain that

$$\begin{aligned}
U_{t+1} &= \left( \mathrm{Id} + \mu \mathcal{A}^* \mathcal{A} \left( X X^T \right) \right) \left( \tilde{U}_t - \sum_{i=1}^{t} \left( \mathrm{Id} + \mu \mathcal{A}^* \mathcal{A} \left( X X^T \right) \right)^{t-i} \hat{E}_i \right) - \hat{E}_{t+1} \\
&= \tilde{U}_{t+1} - \sum_{i=1}^{t} \left( \mathrm{Id} + \mu \mathcal{A}^* \mathcal{A} \left( X X^T \right) \right)^{t+1-i} \hat{E}_i - \hat{E}_{t+1} \\
&= \tilde{U}_{t+1} - \sum_{i=1}^{t+1} \left( \mathrm{Id} + \mu \mathcal{A}^* \mathcal{A} \left( X X^T \right) \right)^{t+1-i} \hat{E}_i,
\end{aligned}$$

which shows the claimed equation (75).

In order to estimate $\| U_t - \tilde{U}_t \|$ we note first that

$$\begin{aligned}
\| \hat{E}_i \| &= \mu \| \mathcal{A}^* \mathcal{A} \left( U_{i-1} U_{i-1}^T \right) U_{i-1} \| \\
&\leq \mu \| \mathcal{A}^* \mathcal{A} \left( U_{i-1} U_{i-1}^T \right) \| \| U_{i-1} \| \\
&\leq (1 + \delta_1) \mu \| U_{i-1} U_{i-1}^T \|_* \| U_{i-1} \| \\
&= (1 + \delta_1) \mu \| U_{i-1} \|_F^2 \| U_{i-1} \|.
\end{aligned}$$

Moreover, we observe that

$$\begin{aligned}
\left\| \left( \mathrm{Id} + \mu \mathcal{A}^* \mathcal{A} \left( X X^T \right) \right)^{t-i} \hat{E}_i \right\| &\leq \left\| \mathrm{Id} + \mu \mathcal{A}^* \mathcal{A} \left( X X^T \right) \right\|^{t-i} \| \hat{E}_i \| \\
&\leq \left( 1 + \mu \| \mathcal{A}^* \mathcal{A} \left( X X^T \right) \| \right)^{t-i} \| \hat{E}_i \| \\
&\leq \left( 1 + \mu \lambda_1 (M) \right)^{t-i} \| \hat{E}_i \|,
\end{aligned}$$

where in the first line we used the submultiplicativity of the operator norm and in the second line we used the triangle inequality. In the third line we used that $\| \mathcal{A}^* \mathcal{A} \left( X X^T \right) \| = \lambda_1 (M)$. Hence, we have shown that

$$\| U_t - \tilde{U}_t \| \leq \sum_{i=1}^{t} \left( 1 + \mu \lambda_1 (M) \right)^{t-i} (1 + \delta_1) \mu \| U_{i-1} \|_F^2 \| U_{i-1} \|. \tag{76}$$

Note that for all $1 \leq i \leq t^\star$ we have that $\|\tilde{U}_{i-1} - U_{i-1}\| \leq \|\tilde{U}_{i-1}\|$, which implies that

$$
\begin{aligned}
\|U_{i-1}\|_F^2 \|U_{i-1}\| &\leq \min\{r; n\} \|U_{i-1}\|^3 \\
&\leq \min\{r; n\} \left( \|\tilde{U}_{i-1}\| + \|\tilde{U}_{i-1} - U_{i-1}\| \right)^3 \\
&\leq 8\min\{r; n\} \|\tilde{U}_{i-1}\|^3 \\
&\leq 8\min\{r; n\} \|\mathrm{Id} + \mu \mathcal{A}^* \mathcal{A}\left(XX^T\right)\|^{3(i-1)} \|U_0\|^3 \\
&\leq 8\min\{r; n\} \left(1 + \mu\lambda_1(M)\right)^{3i-3} \alpha^3 \|U\|^3.
\end{aligned}
$$

In order to proceed assume now $t \leq t^\star$. Then by inequality (76) and the previous inequality we obtain that

$$
\begin{aligned}
\|U_t - \tilde{U}_t\| &\leq \sum_{i=1}^{t} \left(1 + \mu\lambda_1\right)^{t-i} \left(1 + \delta_1\right) \mu \|U_{i-1}\|_F^2 \|U_{i-1}\| \\
&\leq 8 \sum_{i=1}^{t} \left(1 + \mu\lambda_1(M)\right)^{t-i} \left(1 + \delta_1\right) \mu r \left(1 + \mu\lambda_1\right)^{3i-3} \alpha^3 \|U\|^3 \\
&= 8\alpha^3 \mu \min\{r; n\} \left(1 + \delta_1\right) \left(1 + \mu\lambda_1(M)\right)^{t-1} \sum_{i=1}^{t} \left(1 + \mu\lambda_1\right)^{2(i-1)} \|U\|^3 \qquad (77) \\
&= 8\alpha^3 \mu \min\{r; n\} \left(1 + \delta_1\right) \left(1 + \mu\lambda_1(M)\right)^{t-1} \frac{\left(1 + \mu\lambda_1(M)\right)^{2t} - 1}{\left(1 + \mu\lambda_1(M)\right)^2 - 1} \|U\|^3 \\
&\leq \frac{4}{\lambda_1(M)} \alpha^3 \min\{r; n\} \left(1 + \delta_1\right) \left(1 + \mu\lambda_1(M)\right)^{3t} \|U\|^3.
\end{aligned}
$$

This shows the claim. $\qquad\qquad\qquad\qquad\qquad\qquad\qquad\qquad\qquad\qquad\qquad\qquad\qquad$ $\square$

## A.2  Proof of Lemma 8.2

*Proof of Lemma 8.2.* First, we note that $\|\tilde{U}_t\| \geq \left\|\tilde{U}_t^T v_1\right\|_{\ell_2}$. Then, we observe that

$$
\begin{aligned}
\tilde{U}_t^T v_1 &= U_0^T \left(\mathrm{Id} + \mu \mathcal{A}^* \mathcal{A}\left(XX^T\right)\right)^t v_1 \\
&= U_0^T \left(\sum_{i=1}^{n} \left(1 + \mu\lambda_i\right) v_i v_i^T\right)^t v_1 \\
&= U_0^T \sum_{i=1}^{n} \left(1 + \mu\lambda_i\right)^t v_i v_i^T v_1 \\
&= \left(1 + \mu\lambda_1\right)^t U_0^T v_1.
\end{aligned}
$$

This proves that $\|\tilde{U}_t\| \geq \left(1 + \mu\lambda_1(M)\right)^t \left\|U_0^T v_1\right\|_{\ell_2}$. From this observation together with Lemma 8.1 it follows for all $t < t^\star$ that

$$
\frac{\|U_t - \tilde{U}_t\|}{\|\tilde{U}_t\|} \leq \frac{4}{\lambda_1(M)} \alpha^2 \left(\frac{\alpha \min\{r; n\}}{\left\|U_0^T v_1\right\|_{\ell_2}}\right) \left(1 + \delta_1\right) \left(1 + \mu\lambda_1(M)\right)^{2t} \|U\|^3.
$$

In order to finsh the proof, we are going to derive a lower bound for $t^\star$. First we note that by the definition of $t^*$ followed by elementary algebraic manipulations for $t < t^*$ we have

$$
\begin{aligned}
&\frac{4}{\lambda_1(M)} \alpha^2 \left(\frac{\alpha \min\{r; n\}}{\left\|U_0^T v_1\right\|_{\ell_2}}\right) \left(1 + \delta_1\right) \left(1 + \mu\lambda_1(M)\right)^{2t} \|U\|^3 < 1 \\
\Longleftrightarrow \quad &\left(1 + \mu\lambda_1(M)\right)^{2t} \|U\|^3 < \frac{\lambda_1(M)}{4\alpha^2 \left(1 + \delta_1\right)} \left(\frac{\left\|U_0^T v_1\right\|_{\ell_2}}{\alpha \min\{r; n\}}\right) \\
\Longleftrightarrow \quad &t < \frac{\ln\left(\frac{\lambda_1(M)}{4\alpha^2(1+\delta_1)\|U\|^3} \left(\frac{\left\|U_0^T v_1\right\|_{\ell_2}}{\alpha \min\{r; n\}}\right)\right)}{2\ln\left(1 + \mu\lambda_1(M)\right)}.
\end{aligned}
$$

Therefore, we must have

$$t^\star \geq \left\lceil \frac{\ln\left(\frac{\lambda_1(M)}{4\alpha^2(1+\delta_1)\|U\|^3}\left(\frac{\|U_0^T v_1\|_{\ell_2}}{\alpha\min\{r;n\}}\right)\right)}{2\ln\left(1+\mu\lambda_1(M)\right)} \right\rceil.$$

$\square$

### A.3 Proof of Lemma 8.3

*Proof of Lemma 8.3.* **Proof of inequality** (17): Due to Weyl's inequality we have that

$$\sigma_{r_\star}(Z_t U_0 + E_t) \geq \sigma_{r_\star}(Z_t U_0) - \|E_t\| \geq \sigma_{r_\star}(V_L^T Z_t U_0) - \|E_t\|,$$

where the second inequality follows from the Courant-Fisher minimax theorem (see, e.g., [67, Appendix A]). Now we note that

$$\begin{aligned}
\sigma_{r_\star}(V_L^T Z_t U_0) &= \sigma_{\min}(V_L^T Z_t V_L V_L^T U_0) \\
&\geq \sigma_{\min}(V_L^T Z_t V_L)\,\sigma_{\min}(V_L^T U_0) \\
&= \sigma_{r_\star}(Z_t)\,\sigma_{\min}(V_L^T U_0) \\
&= \alpha\sigma_{r_\star}(Z_t)\,\sigma_{\min}(V_L^T U).
\end{aligned}$$

This shows the second statement.

**Proof of inequality** (18): From Weyl's inequality it follows that

$$\sigma_{r_\star+1}(Z_t U_0 + E_t) \leq \sigma_{r_\star+1}(Z_t U_0) + \|E_t\|. \tag{78}$$

Denote by $U = V_U \Sigma_U W_U^T$ the singular value decomposition of $U$. Then we can compute that

$$\begin{aligned}
\sigma_{r_\star+1}(Z_t U_0) &= \alpha \max_{\mathcal{V},\dim\mathcal{V}=r_\star+1} \min_{x\in\mathcal{V},\|x\|=1} \|Z_t U x\| \\
&= \alpha \max_{\mathcal{V},\dim\mathcal{V}=r_\star+1} \min_{x\in\mathcal{V},\|x\|=1} \|Z_t V_U V_U^T U x\| \\
&= \alpha \max_{\mathcal{V},\dim\mathcal{V}=r_\star+1} \min_{x\in\mathcal{V},\|x\|=1} \|Z_t V_U x\| \|U\| \\
&\leq \alpha \max_{\mathcal{V},\dim\mathcal{V}=r_\star+1} \min_{x\in\mathcal{V},\|x\|=1} \|Z_t x\| \|U\| \\
&= \alpha\sigma_{r_\star+1}(Z_t)\|U\|.
\end{aligned}$$

The first line is due to the Courant-Fisher minimax theorem and $U_0 = \alpha U$. The last line follows again from the Courant-Fisher minimax theorem. Together with inequality (78) this implies the third claim.

**Proof of inequality** (19): First, we note that

$$Z_t U_0 + E_t = Z_t V_L V_L^T U_0 + \underbrace{Z_t V_{L^\perp} V_{L^\perp}^T U_0 + E_t}_{=:H}.$$

Note that since $V_L^T V_U$ has rank $r_\star$, the matrix $Z_t V_L V_L^T U$ must have rank $r_\star$ as well. In particular, since $Z_t V_L V_L^T U = V_L V_L^T Z_t V_L V_L^T U$ this means that $L$ is the subspace spanned by the left-singular vectors of $Z_t V_L V_L^T U$ corresponding to the largest $r_\star$ singular values. Due to Wedin's $\sin\theta$ theorem [68] we obtain that

$$\begin{aligned}
\|V_{L^\perp}^T V_{L_t}\| &\leq \frac{\|H\|}{\sigma_{r_\star}(Z_t V_L V_L^T U_0) - \sigma_{r_\star+1}(Z_t U_0 + E_t)} \\
&\leq \frac{\|H\|}{\alpha\sigma_{r_\star}(Z_t)\,\sigma_{\min}(V_L^T U_0) - \sigma_{r_\star+1}(Z_t U_0 + E_t)} \\
&\leq \frac{\|H\|}{\alpha\sigma_{r_\star}(Z_t)\,\sigma_{\min}(V_L^T U) - \alpha\sigma_{r_\star+1}(Z_t)\|U\| - \|E_t\|},
\end{aligned}$$

where in the last line we also used (18). (Note that the assumption (16) guarantees that the denominator is positive, which is a necessary condition for an application of Wedin's $\sin \theta$ theorem.) Now we observe that

$$\|H\| \le \|Z_t V_{L^\perp} V_{L^\perp}^T U\| + \|E_t\| \le \alpha \|Z_t V_{L^\perp}\| \|U\| + \|E_t\| = \alpha \sigma_{r_\star+1}(Z_t) \|U\| + \|E_t\|.$$

Together with the previous inequality chain, this shows inequality (19). $\qquad\square$

## A.4 Proof of Lemma 8.4

Before proving Lemma 8.4, we are going to introduce some notation. Let $U_t = \sum_{i=1}^r \sigma_i u_i v_i^T$ be the singular value decomposition of $U_t$. Define $L_t := \sum_{i=1}^{r_\star} \sigma_i u_i v_i^T$ and $N_t := \sum_{i=r_\star+1}^r \sigma_i u_i v_i^T$. Denote by $L_t = V_{L_t} \Sigma_{L_t} W_{L_t}^T$ and $N_t = V_{N_t} \Sigma_{N_t} W_{N_t}^T$ the singular value decomposition of those two matrices.

We start by proving the following technical lemma. It says that if the subpace spanned by the columns of $X$ and $L_t$ are aligned, then also the subspaces given by $W_t$ and $W_{L_t^\perp}$ will be closely aligned.

**Lemma A.1.** *Assume that $\|V_{X^\perp}^T V_{L_t}\| \le 1/2$. Then it holds that*

$$\|W_{L_t^\perp}^T W_t\| \le \frac{2\sigma_{r_\star+1}(U_t) \|V_{X^\perp}^T V_{L_t}\|}{\sigma_{r_\star}(U_t)}.$$

*Proof.* We note that

$$
\begin{aligned}
\|W_{L_t^\perp}^T W_t\| &= \sqrt{\|W_{L_t^\perp}^T W_t W_t^T W_{L_t^\perp}\|} \\
&= \sqrt{\|W_{L_t^\perp}^T U_t^T V_X \left(V_X^T U_t U_t^T V_X\right)^{-1} V_X^T U_t W_{L_t^\perp}\|} \\
&= \sqrt{\|W_{L_t^\perp}^T U_t^T V_X \left(V_X^T U_t U_t^T V_X\right)^{-1} V_X^T U_t W_{L_t^\perp}\|} \\
&= \sqrt{\|W_{L_t^\perp}^T N_t^T V_X \left(V_X^T U_t U_t^T V_X\right)^{-1} V_X^T N_t W_{L_t^\perp}\|} \\
&= \sqrt{\|W_{L_t^\perp}^T W_{N_t} \Sigma_{N_t} V_{N_t}^T V_X \left(V_X^T U_t U_t^T V_X\right)^{-1} V_X^T V_{N_t} \Sigma_{N_t} W_{N_t}^T W_{L_t^\perp}\|} \\
&= \sqrt{\|\Sigma_{N_t} V_{N_t}^T V_X \left(V_X^T U_t U_t^T V_X\right)^{-1} V_X^T V_{N_t} \Sigma_{N_t}\|} \\
&\le \frac{\|\Sigma_{N_t}\| \|V_{N_t}^T V_X\|}{\sigma_{\min}(V_X^T U_t)}.
\end{aligned}
$$

In order to control the denominator we note that

$$
\begin{aligned}
\sigma_{\min}(V_X^T U_t) &= \sqrt{\sigma_{\min}(V_X^T U_t U_t^T V_X)} \\
&= \sqrt{\sigma_{\min}(V_X^T (L_t L_t^T + N_t N_t^T) V_X)} \\
&\ge \sqrt{\sigma_{\min}(V_X^T L_t L_t^T V_X)} \\
&= \sigma_{\min}(V_X^T L_t) \\
&\ge \sigma_{\min}(V_X^T V_{L_t}) \sigma_{\min}(L_t) \\
&\ge \frac{\sigma_{\min}(L_t)}{2}.
\end{aligned}
$$

In the last line we have used the assumption $\|V_{X^\perp}^T V_{L_t}\| \le 1/2$. Hence, we have shown that

$$
\begin{aligned}
\|W_{L_t^\perp}^T W_t\| &\le \frac{2\|\Sigma_{N_t}\|\|V_{N_t}^T V_X\|}{\sigma_{\min}(\tilde{L})} \\
&= \frac{2\sigma_{r_\star+1}(U_t)\|V_{N_t}^T V_X\|}{\sigma_{r_\star}(U_t)} \\
&\le \frac{2\sigma_{r_\star+1}(U_t)\|V_{L_t^\perp}^T V_X\|}{\sigma_{r_\star}(U_t)} \\
&= \frac{2\sigma_{r_\star+1}(U_t)\|V_{X^\perp}^T V_{L_t}\|}{\sigma_{r_\star}(U_t)},
\end{aligned}
$$

which finishes the proof. $\qquad\qquad\square$

Now we are in a position to prove Lemma 8.4.

*Proof of Lemma 8.4.* **Proof of inequality** (20)**:** First, we observe that due to Lemma A.1 and the assumption $\|V_{X^\perp}^T V_{L_t}\| \le \frac{1}{8}$ we have that

$$
\|W_{L_t^\perp}^T W_t\| \le \frac{2\sigma_{r_\star+1}(U_t)\|V_{X^\perp}^T V_{L_t}\|}{\sigma_{r_\star}(U_t)} \le 1/4. \tag{79}
$$

Then, we note that

$$
\begin{aligned}
\sigma_{r_\star}(U_t W_t)^2 &= \sigma_{r_\star}\left(W_t^T U_t^T U_t W_t\right) \\
&= \sigma_{r_\star}\left(W_t^T\left(L_t^T L_t + N_t^T N_t\right)W_t\right) \\
&\ge \sigma_{r_\star}\left(W_t^T L_t^T L_t W_t\right) \\
&\ge \sigma_{r_\star}\left(W_t^T W_{L_t}\right)^2 \sigma_{r_\star}(L_t)^2 \\
&= \left(1 - \|W_{L_t^\perp}^T W_t\|^2\right)\sigma_{r_\star}(U_t)^2.
\end{aligned}
$$

Using inequality (79) we obtain inequality (20).

**Proof of inequality** (21)**:** Note that

$$
\begin{aligned}
V_{X^\perp}^T V_{U_t W_t} &= V_{X^\perp}^T V_{U_t W_t} V_{U_t W_t}^T U_t W_t \left(V_{U_t W_t}^T U_t W_t\right)^{-1} \\
&= V_{X^\perp}^T U_t W_t \left(V_{U_t W_t}^T U_t W_t\right)^{-1}.
\end{aligned}
$$

By the triangle inequality it follows that

$$
\|V_{X^\perp}^T V_{U_t W_t}\| \le \|V_{X^\perp}^T L_t W_t \left(V_{U_t W_t}^T U_t W_t\right)^{-1}\| + \|V_{X^\perp}^T N_t W_t \left(V_{U_t W_t}^T U_t W_t\right)^{-1}\|.
$$

The second term can be bounded as follows.

$$
\begin{aligned}
\|V_{X^\perp}^T N_t W_t \left(V_{U_t W_t}^T U_t W_t\right)^{-1}\| &\le \frac{\|N_t W_t\|}{\sigma_{r_\star}(U_t W_t)} \\
&\le \frac{\|N_t W_{N_t}\|\|W_{N_t}^T W_t\|}{\sigma_{r_\star}(U_t W_t)} \\
&= \frac{\sigma_{r_\star+1}(U_t)\|W_{N_t}^T W_t\|}{\sigma_{r_\star}(U_t W_t)} \\
&\le \frac{\sigma_{r_\star+1}(U_t)\|W_{L_t^\perp}^T W_t\|}{\sigma_{r_\star}(U_t W_t)} \\
&\le 2\frac{\sigma_{r_\star+1}(U_t)\|W_{L_t^\perp}^T W_t\|}{\sigma_{r_\star}(U_t)}.
\end{aligned} \tag{80}
$$

In order to bound the first term, we note that

$$\|V_{X^\perp}^T L_t W_t \left(V_{U_t W_t}^T U_t W_t\right)^{-1}\| \le \|V_{X^\perp}^T V_{L_t}\| \|L_t W_t \left(V_{U_t W_t}^T U_t W_t\right)^{-1}\|$$

$$\overset{(a)}{\le} \|V_{X^\perp}^T V_{L_t}\| \left(\|U_t W_t \left(V_{U_t W_t}^T U_t W_t\right)^{-1}\| + \|N_t W_t \left(V_{U_t W_t}^T U_t W_t\right)^{-1}\|\right)$$

$$= \|V_{X^\perp}^T V_{L_t}\| \left(1 + \|N_t W_t \left(V_{U_t W_t} U_t W_t\right)^{-1}\|\right)$$

$$\overset{(c)}{\le} \|V_{X^\perp}^T V_{L_t}\| \left(1 + \frac{\|N_t W_t\|}{\sigma_{r_\star}(U_t W_t)}\right)$$

$$\overset{(c)}{\le} \|V_{X^\perp}^T V_{L_t}\| \left(1 + \frac{\sigma_{r_\star+1}(U_t) \|W_{L_t^\perp}^T W_t\|}{\sigma_{r_\star}(U_t W_t)}\right)$$

$$\le \|V_{X^\perp}^T V_{L_t}\| \left(1 + 2\frac{\sigma_{r_\star+1}(U_t) \|W_{L_t^\perp}^T W_t\|}{\sigma_{r_\star}(U_t)}\right)$$

$$\overset{(d)}{\le} 3\|V_{X^\perp}^T L_t\|.$$

In $(a)$ we have used the triangle inequality and inequality $(b)$ follows from inspecting the inequality chain $(80)$. In $(c)$ we used inequality $(20)$ and $(d)$ follows from $(79)$. Combining our results we obtain that

$$\|V_{X^\perp}^T V_{U_t W_t}\| \le 3\|V_{X^\perp}^T V_{L_t}\| + 2\frac{\sigma_{r_\star+1}(U_t) \|W_{L_t^\perp}^T W_t\|}{\sigma_{r_\star}(U_t)}$$

$$\le 3\|V_{X^\perp}^T V_{L_t}\| + 4\frac{\sigma_{r_\star+1}^2(U_t) \|V_{X^\perp}^T V_{L_t}\|}{\sigma_{r_\star}^2(U_t)}$$

$$\le 7\|V_{X^\perp}^T V_{L_t}\|,$$

where in the second line we used Lemma A.1. This shows $(21)$.

**Proof of inequality** $(22)$**:** We note that

$$\|U_t W_{t,\perp}\| \le \|L_t W_{t,\perp}\| + \|N_t W_{t,\perp}\|$$
$$\le \|L_t W_{t,\perp}\| + \|N_t\| \tag{81}$$
$$= \|L_t W_{t,\perp}\| + \sigma_{r_\star+1}(U_t).$$

Observe that $\|L_t W_{t,\perp}\| = \|L_t W_{t,\perp} W_{t,\perp}^T\|$. Then we compute that

$$L_t W_{t,\perp} W_{t,\perp}^T = L_t \left(\mathrm{Id} - U^T V_X \left(V_X^T U_t U_t^T V_X\right)^{-1} V_X^T U\right)$$

$$= L_t \left(\mathrm{Id} - L_t^T V_X \left(V_X^T U_t U_t^T V_X\right)^{-1} V_X^T U\right)$$

$$= L_t \left(W_{L_t} W_{L_t}^T - L_t^T V_X \left(V_X^T U_t U_t^T V_X\right)^{-1} V_X^T U_t\right).$$

Next, we note that

$$V_X^T U U^T V_X = V_X^T L_t L_t^T V_X + V_X^T N_t N_t^T V_X$$

$$= V_X^T L_t W_{L_t} W_{L_t}^T L_t^T V_X + V_X^T N_t N_t^T V_X$$

$$= V_X^T L_t W_{L_t} \left(\mathrm{Id} + \underbrace{\left(V_X^T L_t W_{L_t}\right)^{-1} V_X^T N_t N_t^T V_X \left(W_{L_t}^T L_t^T V_X\right)^{-1}}_{=:A}\right) W_{L_t}^T L_t^T V_X.$$

Now observe that

$$\|A\| \le \frac{\|V_X^T N_t N_t^T V_X\|}{\sigma_{\min}\left(V_X^T L_t W_{L_t}\right)^2}$$

$$\le \frac{\|V_X^T V_{N_t}\|^2 \|N_t\|^2}{\sigma_{\min}\left(V_X^T V_{L_t}\right)^2 \sigma_{\min}\left(L_t\right)^2}$$

$$\le \frac{\|V_X^T V_{L_t^\perp}\|^2 \sigma_{r_\star+1}\left(U_t\right)^2}{\sigma_{\min}\left(V_X^T V_{L_t}\right)^2 \sigma_{r_\star}\left(U_t\right)^2}$$

$$= \frac{\|V_{X^\perp}^T V_{L_t}\|^2 \sigma_{r_\star+1}\left(U_t\right)^2}{\sigma_{\min}\left(V_X^T V_{L_t}\right)^2 \sigma_{r_\star}\left(U_t\right)^2}$$

$$\le \frac{\|V_{X^\perp}^T V_{L_t}\|^2 \sigma_{r_\star+1}\left(U_t\right)^2}{\left(1 - \|V_{X^\perp}^T V_{L_t}\|^2\right) \sigma_{r_\star}\left(U_t\right)^2}$$

$$\le 1/2.$$

In the last line we have used the assumption $\|V_{X^\perp}^T V_{L_t}\| \le \frac{1}{8}$. To continue note that we have

$$L_t^T V_X \left(V_X^T U_t U_t^T V_X\right)^{-1} V_X^T U_t$$

$$= L_t^T V_X \left(W_{L_t}^T L_t^T V_X\right)^{-1} (\mathrm{Id} + A)^{-1} \left(V_X^T L_t W_{L_t}\right)^{-1} V_X^T U_t$$

$$= W_{L_t} (\mathrm{Id} + A)^{-1} \left(V_X^T L_t W_{L_t}\right)^{-1} V_X^T U_t$$

$$= W_{L_t} (\mathrm{Id} + A)^{-1} W_{L_t}^T + W_{L_t} (\mathrm{Id} + A)^{-1} \left(V_X^T L_t W_{L_t}\right)^{-1} V_X^T N_t$$

$$= W_{L_t} W_{L_t}^T - W_{L_t} A (\mathrm{Id} + A)^{-1} W_{L_t}^T + W_{L_t} (\mathrm{Id} + A)^{-1} \left(V_X^T L_t W_{L_t}\right)^{-1} V_X^T N_t.$$

Note that in the last line we used that $\|A\| \le 1/2$, which we have shown above. It follows that

$$L_t W_{t,\perp} W_{t,\perp}^T = L_t W_{L_t} A (\mathrm{Id} + A)^{-1} W_{L_t}^T - L_t W_{L_t} (\mathrm{Id} + A)^{-1} \left(V_X^T L_t W_{L_t}\right)^{-1} V_X^T N_t.$$

In particular, by the triangle inequality it follows that

$$\|L_t W_{t,\perp}\| \le \underbrace{\|L_t W_{L_t} A (\mathrm{Id} + A)^{-1} W_{L_t}^T\|}_{=:(I)} + \underbrace{\|L_t W_{L_t} (\mathrm{Id} + A)^{-1} \left(V_X^T L_t W_{L_t}\right)^{-1} V_X^T N_t\|}_{=:(II)}. \qquad (82)$$

**Bounding $(I)$:** In order to bound the first term, we note that

$$L_t W_{L_t} A (\mathrm{Id} + A)^{-1} W_{L_t}^T$$

$$= L_t W_{L_t} \left(V_X^T L_t W_{L_t}\right)^{-1} V_X^T N_t N_t^T V_X \left(W_{L_t}^T L_t^T V_X\right)^{-1} (\mathrm{Id} + A)^{-1} W_{L_t}^T$$

$$= L_t W_{L_t} \left(V_{L_t}^T L_t W_{L_t}\right)^{-1} \left(V_X^T V_{L_t}\right)^{-1} V_X^T N_t N_t^T V_X \left(W_{L_t}^T L_t^T V_X\right)^{-1} (\mathrm{Id} + A)^{-1} W_{L_t}^T$$

$$= V_{L_t} \left(V_X^T V_{L_t}\right)^{-1} V_X^T N_t N_t^T V_X \left(V_{L_t}^T V_X\right)^{-1} \left(W_{L_t}^T L_t^T V_{L_t}\right)^{-1} (\mathrm{Id} + A)^{-1} W_{L_t}^T.$$

It follows that

$$\|L_t W_{L_t} A (\mathrm{Id} + A)^{-1} W_{L_t}^T\| \le \frac{\|V_X^T N_t N_t^T V_X\|}{\sigma_{\min}\left(V_X^T V_{L_t}\right)^2 \sigma_{\min}(\mathrm{Id} + A) \sigma_{\min}\left(W_{\tilde{L}}^T L_t^T V_{L_t}\right)}$$

$$\le \frac{\|V_X^T V_{N_t}\|^2 \|N_t\|^2}{\sigma_{\min}\left(V_X^T V_{L_t}\right)^2 (1 - \|A\|) \sigma_{r_\star}\left(U_t\right)}$$

$$\le \frac{\|V_X^T V_{L_t^\perp}\|^2 \sigma_{r_\star+1}\left(U_t\right)^2}{\sigma_{\min}\left(V_X^T V_{L_t}\right)^2 (1 - \|A\|) \sigma_{r_\star}\left(U_t\right)}$$

$$= \frac{\|V_{X^\perp}^T V_{L_t}\|^2 \sigma_{r_\star+1}\left(U_t\right)^2}{\sigma_{\min}\left(V_X^T V_{L_t}\right)^2 (1 - \|A\|) \sigma_{r_\star}\left(U_t\right)}$$

$$\le \frac{\sigma_{r_\star+1}\left(U_t\right)}{2}$$

In the last line we have used the assumption $\|V_{X^\perp}^T V_{L_t}\| \le \frac{1}{8}$ as well as $\|A\| \le 1/2$.

**Bounding** $(II)$**:** We observe that

$$L_t W_{L_t} \left(\mathrm{Id} + A\right)^{-1} \left(V_X^T L_t W_{L_t}\right)^{-1} V_X^T N_t$$

$$= L_t W_{L_t} \left(V_X^T L_t W_{\tilde{L}}\right)^{-1} V_X^T N_t - L_t W_{L_t} A \left(\mathrm{Id} + A\right)^{-1} \left(V_X^T L_t W_{L_t}\right)^{-1} V_X^T N_t$$

$$= V_{L_t} \left(V_X^T V_{L_t}\right)^{-1} V_X^T N_t$$

$$\quad - L_t W_{L_t} \left(V_X^T L_t W_{L_t}\right)^{-1} V_X^T N_t N_t^T V_X \left(W_{L_t}^T L_t^T V_X\right)^{-1} \left(\mathrm{Id} + A\right)^{-1} \left(V_X^T L_t W_{L_t}\right)^{-1} V_X^T N_t$$

$$= V_{L_t} \left(V_X^T V_{L_t}\right)^{-1} V_X^T N_t$$

$$\quad - V_{L_t} \left(V_X^T V_{L_t}\right)^{-1} V_X^T N_t N_t^T V_X \left(W_{L_t}^T L_t^T V_X\right)^{-1} \left(\mathrm{Id} + A\right)^{-1} \left(V_X^T L_t W_{L_t}\right)^{-1} V_X^T N_t$$

$$= V_{L_t} \left(V_X^T V_{L_t}\right)^{-1} V_X^T N_t \left(\mathrm{Id} - N_t^T V_X \left(W_{L_t}^T L_t^T V_X\right)^{-1} \left(\mathrm{Id} + A\right)^{-1} \left(V_X^T L_t W_{L_t}\right)^{-1} V_X^T N_t\right).$$

It follows that

$$\|L_t W_{L_t} \left(\mathrm{Id} + A\right)^{-1} \left(V_X^T L_t W_{L_t}\right)^{-1} V_X^T N_t\|$$

$$\le \frac{\|V_X^T V_{N_t}\| \sigma_{r_\star + 1}\left(U_t\right)}{\sigma_{\min}\left(V_X^T V_{L_t}\right)} \left(1 + \frac{\sigma_{r_\star + 1}\left(U_t\right)^2 \|V_X^T V_{N_t}\|^2}{\left(1 - \|A\|\right) \sigma_{r_\star}\left(U_t\right)^2 \sigma_{\min}\left(V_X^T V_{L_t}\right)^2}\right)$$

$$\le \frac{\|V_X^T V_{L_t^\perp}\| \sigma_{r_\star + 1}\left(U_t\right)}{\sigma_{\min}\left(V_X^T V_{L_t}\right)} \left(1 + \frac{\sigma_{r_\star + 1}\left(U_t\right)^2 \|V_X^T V_{L_t^\perp}\|^2}{\left(1 - \|A\|\right) \sigma_{r_\star}\left(U_t\right)^2 \sigma_{\min}\left(V_X^T V_{L_t}\right)^2}\right)$$

$$\le \frac{\sigma_{r_\star + 1}\left(U_t\right)}{2}.$$

In the last line we have used the assumption $\|V_{X^\perp}^T V_{L_t}\| \le \frac{1}{8}$ as well as $\|A\| \le 1/2$. Hence, from inequality (82) it follows that $\|L_t W_{t,\perp}\| \le \sigma_{r_\star + 1}\left(U_t\right)$. Inserting this result into inequality (81) we obtain inequality (22), which finishes the proof. $\qquad \square$

## A.5 Proof of Lemma 8.5

Before we can prove Lemma 8.5 we will need a technical lemma. In order to state it, recall that $L$ denotes the subspace spanned by the eigenvectors corresponding to the $r_\star$ largest eigenvalues of the matrix $M := \mathcal{A}^* \mathcal{A} \left(X X^T\right)$ and that $V_L \in \mathbb{R}^{n \times r_\star}$ is an orthogonal matrix, whose column span is the subpace $L$. The following lemma, which follows from standard matrix perturbation theory arguments, shows that for if $\mathcal{A}^* \mathcal{A} \left(X X^T\right)$ is sufficiently close to $X X^T$ in spectral norm, then $L$ is aligned with the column space of $X$. Moreover, it says that the eigenvalues of $X X^T$ are close to the ones of $M$.

**Lemma A.2.** *Suppose that* $M := \mathcal{A}^* \mathcal{A} \left(X X^T\right) = X X^T + \tilde{E}$ *with* $\|\tilde{E}\| \le \delta \lambda_{r_\star}\left(X X^T\right)$ *and* $\delta < 1/2$. *Then it holds that*

$$\left(1 - \delta\right) \lambda_1 \left(X X^T\right) \le \lambda_1\left(M\right) \le \left(1 + \delta\right) \lambda_1\left(X X^T\right),$$

$$\lambda_{r_\star + 1}\left(M\right) \le \delta \lambda_{r_\star}\left(X X^T\right),$$

$$\lambda_{r_\star}\left(M\right) \ge \left(1 - \delta\right) \lambda_{r_\star}\left(X X^T\right),$$

$$\|V_{X^\perp}^T V_L\| \le 2\delta.$$

*Proof.* The first three inequalities are a direct consequence of Weyl's inequality. In order to prove the fourth inequality, we denote by $L$ the subspace spanned by the eigenvectors corresponding to the $r_\star$ largest eigenvalues of $M$. From the Davis-Kahan $\sin \Theta$ theorem [69] it follows that

$$\|V_{X^\perp}^T V_L\| \le \frac{\|\tilde{E}\|}{\lambda_{r_\star}\left(X X^T\right) - \|\tilde{E}\|} \overset{(a)}{\le} \frac{\delta}{1 - \delta} \overset{(b)}{\le} 2\delta.$$

Inequality $(a)$ follows from the assumption $\|\tilde{E}\| \le \delta \lambda_{r_\star}\left(X X^T\right)$. In $(b)$ we used that $\delta \le \frac{1}{2}$.

$\qquad \square$

This allows us to prove Lemma 8.5.

*Proof of Lemma 8.5.* Due to the assumption (23) we have that $\gamma < 1/2$, if $\widetilde{c}_2$ is chosen small enough, and hence we can apply Lemma 8.3. Hence, we obtain that

$$\sigma_{r_\star}(U_t) \geq \alpha\sigma_{r_\star}(Z_t)\sigma_{\min}(V_L^T U) - \|E_t\| \overset{(a)}{\geq} \frac{\alpha}{2}\sigma_{r_\star}(Z_t)\sigma_{\min}(V_L^T U), \tag{83}$$

$$\sigma_{r_\star+1}(U_t) \leq \gamma\alpha\sigma_{r_\star}(Z_t)\sigma_{\min}(V_L^T U), \tag{84}$$

where in $(a)$ we used that $\gamma \leq 1/2$. Moreover, we also have that

$$\|V_{L^\perp}^T V_{L_t}\| \leq \frac{\alpha\sigma_{r_\star+1}(Z_t)\|U\| + \|E_t\|}{\alpha\sigma_{r_\star}(Z_t)\sigma_{\min}(V_L^T U) - \alpha\sigma_{r_\star+1}(Z_t)\|U\| - \|E_t\|} \leq \frac{\gamma}{1-\gamma}.$$

Now note that

$$\begin{aligned}
\|V_{X^\perp}^T V_{L_t}\| &= \|V_X^T V_X^T - V_{L_t} V_{L_t}^T\| \\
&\leq \|V_X^T V_X^T - V_L V_L^T\| + \|V_L V_L^T - V_{L_t} V_{L_t}^T\| \\
&= \|V_{X^\perp}^T V_L\| + \|V_{L^\perp}^T V_{L_t}\| \\
&\leq 2\delta + \frac{\gamma}{1-\gamma},
\end{aligned}$$

where in the last inequality we applied Lemma 8.3 and Lemma A.2. Hence, by our assumptions on $\delta$ and $\gamma$ we can apply Lemma 8.4. Together with the inequality (83) we obtain

$$\sigma_{\min}(U_t W_t) \geq \frac{1}{2}\sigma_{r_\star}(U_t) \geq \frac{\alpha}{4}\sigma_{r_\star}(Z_t)\sigma_{\min}(V_L^T U)$$

as well as

$$\begin{aligned}
\|V_{X^\perp}^T V_{U_t W_t}\| &\leq 7\|V_{X^\perp}^T V_{L_t}\| \\
&\leq 7\left(8\delta + \frac{\gamma}{1-\gamma}\right) \\
&\leq 56(\delta + \gamma).
\end{aligned}$$

Moreover, it also follows from Lemma 8.4, inequality (84) and our assumption on $\gamma$ that

$$\begin{aligned}
\|U_t W_{t,\perp}\| &\leq 2\sigma_{r_\star+1}(U_t) \\
&\leq 2\gamma\alpha\sigma_{r_\star}(Z_t)\sigma_{\min}(V_L^T U) \\
&\leq \frac{\kappa^{-2}}{8}\alpha\sigma_{r_\star}(Z_t)\sigma_{\min}(V_L^T U).
\end{aligned}$$

This finishes the proof. $\qquad\qquad\square$

### A.6   Proof of Lemma 8.6

*Proof of Lemma 8.6.* In order to apply Lemma 8.5, we need to show that $\gamma \leq \widetilde{c}_2\kappa^{-2}$ for an appropriately chosen $t_\star = t$. We are going to show the stronger statement $\gamma \leq c_3\kappa^{-2}$, where $c_3$ is a sufficiently small constant depending only on $c$, which will be specified later. Note that by the definition of $\gamma$ it suffices to check the following two conditions.

$$\sigma_{r_\star+1}(Z_t)\|U\| \leq \frac{c_3}{2}\sigma_{r_\star}(Z_t)\sigma_{\min}(V_L^T U)\kappa^{-2}, \tag{85}$$

$$\|E_t\| \leq \frac{c_3}{2}\alpha\sigma_{r_\star}(Z_t)\sigma_{\min}(V_L^T U)\kappa^{-2}. \tag{86}$$

By using the identity $Z_t = (\text{Id} + \mu M)^t$ and by rearranging terms we see that the first inequality is equivalent to the inequality

$$\frac{2\kappa^2\|U\|}{c_3\sigma_{\min}(V_L^T U)} \leq \left(\frac{1 + \mu\lambda_{r_\star}(M)}{1 + \mu\lambda_{r_\star+1}(M)}\right)^t.$$

Hence, if we set

$$t_\star = \left\lceil \underbrace{\ln\left( \frac{2\kappa^2 \|U\|}{c_3 \sigma_{\min}\left(V_L^T U\right)} \right)}_{=:\sigma} \left( \ln\left( \frac{1 + \mu\lambda_{r_\star}\left(M\right)}{1 + \mu\lambda_{r_\star + 1}\left(M\right)} \right) \right)^{-1} \right\rceil,$$

we see that condition (85) is satisfied. Let us check that this choice is feasible, i.e. $t_\star \le t^\star$. By Lemma 8.2 and the definition of $t_\star$ it suffices to show that

$$\ln\left( \frac{2\kappa^2 \|U\|}{c_3 \sigma_{\min}\left(V_L^T U\right)} \right)\left( \ln\left( \frac{1 + \mu\lambda_{r_\star}\left(M\right)}{1 + \mu\lambda_{r_\star + 1}\left(M\right)} \right) \right)^{-1} \le \frac{\ln\left( \frac{\lambda_1(M)}{4\alpha^2(1+\delta_1)\|U\|^3}\left( \frac{\|U_0^T v_1\|_{\ell_2}}{\alpha r} \right) \right)}{8\ln\left(1 + \mu\lambda_1\left(M\right)\right)}.$$

Next, we note that

$$\frac{\ln\left(1 + \mu\lambda_1\left(M\right)\right)}{\ln\left( \frac{1+\mu\lambda_{r_\star}(M)}{1+\mu\lambda_{r_\star+1}(M)} \right)} = \frac{\ln\left(1 + \mu\lambda_1\left(M\right)\right)}{\ln\left( 1 + \frac{\mu\left(\lambda_{r_\star}(M) - \lambda_{r_\star+1}(M)\right)}{1+\mu\lambda_{r_\star+1}(M)} \right)}$$

$$\le \frac{\lambda_1\left(M\right) \cdot \frac{1+\mu\lambda_{r_\star}(M)}{1+\mu\lambda_{r_\star+1}(M)}}{\frac{\lambda_{r_\star}(M) - \lambda_{r_\star+1}(M)}{1+\mu\lambda_{r_\star+1}(M)}} \tag{87}$$

$$= \frac{\lambda_1\left(M\right)\left(1 + \mu\lambda_{r_\star}\left(M\right)\right)}{\lambda_{r_\star}\left(M\right) - \lambda_{r_\star+1}\left(M\right)} \le 2\kappa^2,$$

where in the first inequality we have used the elementary inequality $\frac{x}{1+x} \le \ln\left(1 + x\right) \le x$. in the last inequality we used our assumption on the step size $\mu$, Lemma A.2 as well as our assumption on $\delta > 0$ with a sufficiently small constant $c_1$. Hence, $t_\star \le t^\star$ is implied by

$$\ln\left( \frac{2\kappa^2 \|U\|}{c_3 \sigma_{\min}\left(V_L^T U\right)} \right) \le \frac{1}{9\kappa^2} \ln\left( \frac{\lambda_1\left(M\right)}{4\alpha^2\left(1+\delta_1\right)\|U\|^3}\left( \frac{\|U_0^T v_1\|_{\ell_2}}{\alpha \min\left\{r; n\right\}} \right) \right).$$

By rearranging terms we see that this inequality is equivalent to

$$\alpha^2 \le \frac{\lambda_1\left(M\right)}{4\left(1+\delta_1\right)\|U\|^3}\left( \frac{\|U_0^T v_1\|_{\ell_2}}{\alpha \min\left\{r; n\right\}} \right)\left( \frac{2\kappa^2 \|U\|}{c_3 \sigma_{\min}\left(V_L^T U\right)} \right)^{-9\kappa^2}.$$

Since by assumption $\delta_1 < 1$ and since by Lemma A.2 we have $\lambda_1\left(M\right) \ge \frac{1}{2}\|X\|^2$, we observe that this inequality is implied by

$$\alpha^2 \le \frac{\|X\|^2}{16\|U\|^3}\left( \frac{\|U_0^T v_1\|_{\ell_2}}{\alpha \min\left\{r; n\right\}} \right)\left( \frac{2\kappa^2 \|U\|}{c_3 \sigma_{\min}\left(V_L^T U\right)} \right)^{-9\kappa^2} = \frac{\|X\|^2}{16\|U\|^3}\left( \frac{\|U^T v_1\|_{\ell_2}}{\min\left\{r; n\right\}} \right)\left( \frac{2\kappa^2 \|U\|}{c_3 \sigma_{\min}\left(V_L^T U\right)} \right)^{-9\kappa^2},$$

which follows from assumption (27), which shows $t_\star \le t^\star$.

In order to show condition (86), we recall that by Lemma 8.1 (which we can apply since we just showed $t_\star \le t^\star$)

$$\|E_{t_\star}\| \le \frac{4}{\lambda_1\left(M\right)}\alpha^3 \min\left\{r; n\right\}\left(1+\delta_1\right)\left(1 + \mu\lambda_1\left(M\right)\right)^{3t_\star}\|U\|^3.$$

Hence, inequality (86) is implied by the inequality

$$\frac{8}{\lambda_1\left(M\right)}\alpha^2 \min\left\{r; n\right\}\left(1+\delta_1\right)\left(1 + \mu\lambda_1\left(M\right)\right)^{3t_\star}\|U\|^3 \le c_3\left(1 + \mu\lambda_{r_\star}\left(M\right)\right)^t \sigma_{\min}\left(V_L^T U\right)\kappa^{-2}.$$

This, in turn, is equivalent to

$$\alpha^2 \le \frac{c_3\lambda_1\left(M\right)\sigma_{\min}\left(V_L^T U\right)}{8\min\left\{r; n\right\}\left(1+\delta_1\right)\kappa^2\|U\|^3}\left[ \frac{1 + \mu\lambda_{r_\star}\left(M\right)}{\left(1 + \mu\lambda_1\left(M\right)\right)^3} \right]^{t_\star}. \tag{88}$$

In order to proceed, we note that

$$\left[\frac{1+\mu\lambda_{r_\star}(M)}{(1+\mu\lambda_1(M))^3}\right]^{t_\star} \geq \exp\left(-3t_\star\ln\left(1+\mu\lambda_1(M)\right)\right)$$

$$\geq \exp\left(-\sigma\frac{6\ln\left(1+\mu\lambda_1(M)\right)}{\ln\left(\frac{1+\mu\lambda_{r_\star}(M)}{1+\mu\lambda_{r_\star+1}(M)}\right)}\right).$$

Hence, using (87), we have shown that

$$\left[\frac{1+\mu\lambda_{r_\star}(M)}{(1+\mu\lambda_1(M))^3}\right]^{t_\star} \geq \exp\left(-12\sigma\kappa^2\right).$$

Inserting this into (88) and using the definition of $t_\star$, we have shown that inequality (86) holds, if

$$\alpha^2 \leq \frac{c_2\|X\|^2\sigma_{\min}\left(V_L^TU\right)}{32r\kappa\|U\|^3}\left(\frac{2\kappa^2\|U\|}{c_3\sigma_{\min}\left(V_L^TU\right)}\right)^{-12\kappa^2}$$

holds, which is precisely our assumption on $\alpha$. In particular, we have shown that $\gamma \leq c_3\kappa^{-2}$, which allows us to apply Lemma 8.5. We obtain that

$$\sigma_{\min}\left(U_{t_\star}W_{t_\star}\right) \geq \frac{\alpha}{4}\sigma_{r_\star}\left(Z_{t_\star}\right)\sigma_{\min}\left(V_L^TU\right),$$

$$\|U_tW_{t_\star,\perp}\| \leq \frac{\kappa^{-2}}{8}\alpha\sigma_{r_\star}\left(Z_{t_\star}\right)\sigma_{\min}\left(V_L^TU\right),$$

$$\|V_{X^\perp}^TV_{U_{t_\star}W_{t_\star}}\| \leq 56\left(\delta+\gamma\right) \overset{(a)}{\leq} c\kappa^{-2},$$

where inequality $(a)$ follows from setting $c_1$ and $c_3$ small enough. Setting $\beta := \sigma_{r_\star}\left(Z_{t_\star}\right)\sigma_{\min}\left(V_L^TU\right)$ shows inequalities (29), (30), and (31). It remains to verify that $\|U_{t_\star}\|$, $t_\star$, and $\beta$ have the desired properties. We start with $t_\star$. Note that

$$\ln\left(\frac{1+\mu\lambda_{r_\star}(M)}{1+\mu\lambda_{r_\star+1}(M)}\right) \leq \ln\left(1+\mu\lambda_{r_\star}(M)\right) \leq \mu\lambda_{r_\star}(M) \leq \mu\left(1+\delta\right)\sigma_{r_\star}(X)^2$$

as well as

$$\ln\left(\frac{1+\mu\lambda_{r_\star}(M)}{1+\mu\lambda_{r_\star+1}(M)}\right) \geq \frac{\mu\lambda_{r_\star}(M)}{1+\mu\lambda_{r_\star}(M)} - \mu\lambda_{r_\star+1}(M) \geq \frac{1}{2}\mu\sigma_{r_\star}(X)^2.$$

Here we have used the inequalities $\frac{x}{1+x} \leq \ln\left(1+x\right) \leq x$, $\lambda_{r_\star}(M) \leq \delta\sigma_{\min}(X)^2$, and $(1-\delta)\sigma_{\min}(X)^2 \leq \lambda_{r_\star}(M) \leq (1+\delta)\sigma_{r_\star}(X)^2$ from Lemma A.2. Hence, these estimates show that $t_\star$ has the desired property.

Next, we are going to prove the desired bound for $\|U_{t_\star}\|$. We obtain that

$$\|U_{t_\star}\| \leq \alpha\|Z_{t_\star}\|\|U\| + \|E_{t_\star}\|$$

$$= \alpha\left(1+\mu\lambda_1(M)\right)^{t_\star}\|U\| + \|E_{t_\star}\|$$

$$\overset{(a)}{\leq} 2\alpha\left(1+\mu\lambda_1(M)\right)^{t_\star}\|U\|$$

$$\leq 2\alpha\exp\left(2\sigma\frac{\ln\left(1+\mu\lambda_1(M)\right)}{\ln\left(\frac{1+\mu\lambda_{r_\star}(M)}{1+\mu\lambda_{r_\star+1}(M)}\right)}\right)\|U\|$$

$$\leq 2\alpha\exp\left(4\sigma\kappa^2\right)\|U\|,$$

where $(a)$ follows from (85) and in the last line we used inequality (87). Hence, by inserting the definition of $\sigma$ we have shown that

$$\|U_{t_\star}\| \leq 2\alpha\left(\frac{2\kappa^2\|U\|}{c_3\sigma_{\min}\left(V_L^TU\right)}\right)^{4\kappa^2}\|U\|$$

$$\overset{(a)}{\leq} 2\sqrt{\frac{c_2\|X\|^2\sigma_{\min}\left(V_L^TU\right)}{32\min\{r;n\}\kappa\|U\|}}\left(\frac{2\kappa^2\|U\|}{c_3\sigma_{\min}\left(V_L^TU\right)}\right)^{-2\kappa^2}$$

$$\leq 3\|X\|$$

where in inequality $(a)$ we have used the assumption on $\alpha$. This shows inequality (28). Now let us check that $\beta$ has the desired property. For that, note that

$$\beta = \left(1 + \mu\lambda_{r_\star}(M)\right)^{t_\star}\sigma_{\min}\left(V_L^T U\right) = \sigma_{\min}\left(V_L^T U\right)\exp\left(t_\star\ln\left(1 + \mu\lambda_{r_\star}(M)\right)\right).$$

By inserting the definition of $t_\star$ and using inequality (87) we can show the upper bound for $\beta$ in inequality (32). The lower bound follows immediately from the definition of $\beta$. This finishes the proof. $\qquad\square$

# B  Proofs for the saddle avoidance phase and the refinement phase

## B.1  Proof of Lemma 9.1

*Proof of Lemma 9.1.* Let $W_t$ and $W_{t,\perp}$ be defined as before. We note that

$$
\begin{aligned}
V_X^T U_{t+1}W_t ={}& V_X^T\left(\mathrm{Id} + \mu\mathcal{A}^*\mathcal{A}\left(XX^T - U_t U_t^T\right)\right)U_t W_t\\
={}& V_X^T\left(\mathrm{Id} + \mu\left(XX^T - U_t U_t^T\right) + \mu\left[\left(\mathcal{A}^*\mathcal{A} - \mathrm{Id}\right)\left(XX^T - U_t U_t^T\right)\right]\right)U_t W_t\\
={}& V_X^T U_t W_t + \mu\Sigma_X^2 V_X^T U_t W_t - \mu V_X^T U_t U_t^T U_t W_t + \mu V_X^T\left[\left(\mathcal{A}^*\mathcal{A} - \mathrm{Id}\right)\left(XX^T - U_t U_t^T\right)\right]U_t W_t\\
={}& \left(\mathrm{Id} + \mu\Sigma_X^2\right)V_X^T U_t W_t - \mu V_X^T U_t W_t W_t^T U_t^T U_t W_t + \mu V_X^T\left[\left(\mathcal{A}^*\mathcal{A} - \mathrm{Id}\right)\left(XX^T - U_t U_t^T\right)\right]U_t W_t\\
={}& \left(\mathrm{Id} + \mu\Sigma_X^2\right)V_X^T U_t W_t - \mu V_X^T U_t W_t W_t^T U_t^T V_X V_X^T U_t W_t - \mu V_X^T U_t W_t W_t^T U_t^T V_{X^\perp}V_{X^\perp}^T U_t W_t\\
&+ \mu V_X^T\left[\left(\mathcal{A}^*\mathcal{A} - \mathrm{Id}\right)\left(XX^T - U_t U_t^T\right)\right]U_t W_t\\
={}& \left(\mathrm{Id} + \mu\Sigma_X^2\right)V_X^T U_t W_t\left(\mathrm{Id} - \mu W_t^T U_t^T V_X V_X^T U_t W_t\right) - \mu V_X^T U_t W_t W_t^T U_t^T V_{X^\perp}V_{X^\perp}^T U_t W_t\\
&+ \mu V_X^T\left[\left(\mathcal{A}^*\mathcal{A} - \mathrm{Id}\right)\left(XX^T - U_t U_t\right)\right]U_t W_t + \mu^2\Sigma_X^2 V_X^T U_t W_t W_t^T U_t^T V_X V_X^T U_t W_t\\
={}& \left(\mathrm{Id} + \mu\Sigma_X^2\right)V_X^T U_t W_t\left(\mathrm{Id} - \mu W_t^T U_t^T V_X V_X^T U_t W_t\right) - \mu\underbrace{V_X^T U_t U_t^T V_{X^\perp}V_{X^\perp}^T U_t W_t}_{=:A_1}\\
&+ \mu\underbrace{V_X^T\left[\left(\mathcal{A}^*\mathcal{A} - \mathrm{Id}\right)\left(XX^T - U_t U_t^T\right)\right]U_t W_t}_{=:A_2} + \mu^2\underbrace{\Sigma_X^2 V_X^T U_t U_t^T V_X V_X^T U_t W_t}_{=:A_3}.
\end{aligned}
$$

First, we want to bring all $A_i$ into the form $P_i V_X^T U_t W_t\left(\mathrm{Id} - \mu W_t^T U_t^T V_X V_X^T U_t W_t\right)$ for $i \in \{1; 2; 3\}$.

**Rewriting $A_1$:** Now let the singular value decomposition of $U_{t+1}W_t \in \mathbb{R}^{n\times r_\star}$ be given by $V_{U_{t+1}W_t}\Sigma_{U_{t+1}W_t}W_{U_{t+1}W_t}^T$ with $V_{U_{t+1}W_t} \in \mathbb{R}^{n\times r_\star}$. This allows us to compute

$$
\begin{aligned}
V_{X^\perp}^T U_t W_t &= V_{X^\perp}^T U_t W_t\left(V_X^T U_t W_t\right)^{-1}V_X^T U_t W_t\\
&= V_{X^\perp}^T V_{U_t W_t}V_{U_t W_t}^T U_t W_t\left(V_X^T V_{U_t W_t}V_{U_t W_t}^T U_t W_t\right)^{-1}V_X^T U W_t\\
&= V_{X^\perp}^T V_{U_t W_t}\left(V_X^T V_{U_t W_t}\right)^{-1}V_X^T U_t W_t.
\end{aligned}
$$

We compute that

$$
\begin{aligned}
&V_X^T U_t U_t^T V_{X^\perp}V_{X^\perp}^T U_t W_t\\
={}& V_X^T U_t U_t^T V_{X^\perp}V_{X^\perp}^T V_{U_t W_t}\left(V_X^T V_{U_t W_t}\right)^{-1}V_X^T U_t W_t\\
={}& V_X^T U_t U_t^T V_{X^\perp}V_{X^\perp}^T V_{U_t W_t}\left(V_X^T V_{U_t W_t}\right)^{-1}V_X^T U_t W_t\left(\mathrm{Id} - \mu W_t^T U_t^T V_X V_X^T U_t W_t\right)^{-1}\left(\mathrm{Id} - \mu W_t^T U_t^T V_X V_X^T U_t W_t\right)\\
={}& V_X^T U_t U_t^T V_{X^\perp}V_{X^\perp}^T V_{U_t W_t}\left(V_X^T V_{U_t W_t}\right)^{-1}\left(\mathrm{Id} - \mu V_X^T U W_t W_t^T U_t V_X\right)^{-1}V_X^T U_t W_t\left(\mathrm{Id} - \mu W_t^T U_t^T V_X V_X^T U_t W_t\right)\\
={}& \underbrace{V_X^T U_t U_t^T V_{X^\perp}V_{X^\perp}^T V_{U_t W_t}\left(V_X^T V_{U_t W_t}\right)^{-1}\left(\mathrm{Id} - \mu V_X^T U_t U_t^T V_X\right)^{-1}}_{=:P_1}V_X^T U_t W_t\left(\mathrm{Id} - \mu W_t^T U_t^T V_X V_X^T U_t W_t\right).
\end{aligned}
$$

(89)

**Rewriting $A_2$:** We observe that

$$
\begin{aligned}
U_t W_t &= V_{U_t W_t}V_{U_t W_t}^T U_t W_t\left(V_X^T V_{U_t W_t}V_{U_t W_t}^T U_t W_t\right)^{-1}V_X^T U_t W_t\\
&= V_{U_t W_t}\left(V_X^T V_{U_t W_t}\right)^{-1}V_X^T U_t W_t.
\end{aligned}
$$

Hence, we can write

$$V_X^T \left[ (\mathcal{A}^* \mathcal{A} - \mathrm{Id}) \left( X X^T - U_t U_t^T \right) \right] U_t W_t$$

$$= V_X^T \left[ (\mathcal{A}^* \mathcal{A} - \mathrm{Id}) \left( X X^T - U_t U_t^T \right) \right] V_{U_t W_t} \left( V_X^T V_{U_t W_t} \right)^{-1} V_X^T U_t W_t$$

$$= V_X^T \left[ (\mathcal{A}^* \mathcal{A} - \mathrm{Id}) \left( X X^T - U_t U_t^T \right) \right] V_{U_t W_t} \left( V_X^T V_{U_t W_t} \right)^{-1} V_X^T U_t W_t \left( \mathrm{Id} - \mu W_t^T U_t^T V_X V_X^T U_t W_t \right)^{-1}$$
$$\cdot \left( \mathrm{Id} - \mu W_t^T U_t^T V_X V_X^T U_t W_t \right)$$

$$= \underbrace{V_X^T \left[ (\mathcal{A}^* \mathcal{A} - \mathrm{Id}) \left( X X^T - U_t U_t^T \right) \right] V_{U_t W_t} \left( V_X^T V_{U_t W_t} \right)^{-1} \left( \mathrm{Id} - \mu V_X^T U_t W_t W_t^T U_t^T V_X \right)^{-1} V_X^T U_t W_t}_{=:P_2}$$
$$\cdot \left( \mathrm{Id} - \mu W_t^T U_t^T V_X V_X^T U_t W_t \right).$$

**Rewriting $A_3$:** Note that

$$\Sigma_X^2 V_X^T U_t U_t^T V_X V_X^T U_t W_t$$

$$= \Sigma_X^2 V_X^T U_t W_t W_t^T U_t^T V_X V_X^T U_t W_t$$

$$= \Sigma_X^2 V_X^T U_t W_t W_t^T U_t^T V_X V_X^T U_t W_t \left( \mathrm{Id} - \mu W_t^T U_t^T V_X V_X^T U_t W_t \right)^{-1} \left( \mathrm{Id} - \mu W_t^T U_t^T V_X V_X^T U_t W_t \right)$$

$$= \underbrace{\Sigma_X^2 V_X^T U_t W_t \left( \mathrm{Id} - \mu W_t^T U_t^T V_X V_X^T U_t W_t \right)^{-1} W_t^T U_t^T V_X V_X^T U_t W_t}_{=:P_3} \left( \mathrm{Id} - \mu W_t^T U_t V_X V_X^T U_t W_t \right).$$

Hence, we have computed that

$$V_X^T U_{t+1} W_t = \left( \mathrm{Id} + \mu \Sigma_X^2 - \mu P_1 + \mu P_2 + \mu^2 P_3 \right) V_X^T U_t W_t \left( \mathrm{Id} - \mu W_t^T U_t^T V_X V_X^T U W_t \right). \quad (90)$$

It follows that

$$\sigma_{\min} \left( V_X^T U_{t+1} W_t \right)$$

$$\geq \sigma_{\min} \left( \mathrm{Id} + \mu \Sigma_X^2 - \mu P_1 + \mu P_2 + \mu^2 P_3 \right) \sigma_{\min} \left( V_X^T U_t W_t \left( \mathrm{Id} - \mu W_t^T U_t^T V_X V_X^T U_t W_t \right) \right)$$

$$\overset{(a)}{=} \sigma_{\min} \left( \mathrm{Id} + \mu \Sigma_X^2 - \mu P_1 + \mu P_2 + \mu^2 P_3 \right) \sigma_{\min} \left( V_X^T U_t W_t \right) \left( 1 - \mu \sigma_{\min}^2 \left( V_X^T U_t W_t \right) \right)$$

$$= \sigma_{\min} \left( \mathrm{Id} + \mu \Sigma_X^2 - \mu P_1 + \mu P_2 + \mu^2 P_3 \right) \sigma_{\min} \left( V_X^T U_t \right) \left( 1 - \mu \sigma_{\min}^2 \left( V_X^T U_t \right) \right) \quad (91)$$

$$\overset{(b)}{\geq} \left( \sigma_{\min} \left( \mathrm{Id} + \mu \Sigma_X^2 \right) - \mu \| P_1 \| - \mu \| P_2 \| - \mu^2 \| P_3 \| \right) \sigma_{\min} \left( V_X^T U_t \right) \left( 1 - \mu \sigma_{\min}^2 \left( V_X^T U_t \right) \right)$$

$$= \left( 1 + \mu \sigma_{\min}^2 (X) - \mu \| P_1 \| - \mu \| P_2 \| - \mu^2 \| P_3 \| \right) \sigma_{\min} \left( V_X^T U_t \right) \left( 1 - \mu \sigma_{\min}^2 \left( V_X^T U_t \right) \right).$$

Equality $(a)$ can be obtained by using the singular value decomposition of $V_X^T U_t W_t$ and the fact that $\mu \leq 1 / \left( \sqrt{3} \| V_X^T U_t \|^2 \right)$, which follows from our assumption on $\mu$. For inequality $(b)$ we used Weyl's inequality. In order to proceed, we are going to estimate $\| P_1 \|$, $\| P_2 \|$, and $\| P_3 \|$. First, we note that

$$\| P_1 \| \overset{(a)}{\leq} \| V_X^T U_t W_t W_t^T U_t^T V_{X^\perp} V_{X^\perp}^T V_{U_t W} \| \| \left( V_X^T V_{U_t W_t} \right)^{-1} \| \| \left( \mathrm{Id} - \mu V_X^T U_t U_t^T V_X \right)^{-1} \|$$

$$\leq \| U_t W_t \| \| V_{X^\perp}^T U_t W_t \| \| V_{X^\perp}^T V_{U_t W_t} \| \| \left( V_X^T V_{U_t W_t} \right)^{-1} \| \| \left( \mathrm{Id} - \mu V_X^T U_t U_t^T V_X \right)^{-1} \|$$

$$\leq \| U_t W_t \|^2 \| V_{X^\perp}^T V_{U_t W_t} \|^2 \left( V_X^T V_{U_t W_t} \right)^{-1} \| \| \left( \mathrm{Id} - \mu V_X^T U_t U_t^T V_X \right)^{-1} \|$$

$$= \frac{\| U_t W_t \|^2 \| V_{X^\perp}^T V_{U_t W_t} \|^2}{\sigma_{\min} \left( V_X^T V_{U_t W_t} \right) \sigma_{\min} \left( \mathrm{Id} - \mu V_X^T U_t U_t^T V_X \right)}$$

$$= \frac{\| U_t W_t \|^2 \| V_{X^\perp}^T V_{U_t W_t} \|^2}{\sigma_{\min} \left( V_X^T V_{U_t W_t} \right) \left( 1 - \mu \| V_X^T U_t \|^2 \right)}$$

$$\overset{(b)}{\leq} 4 \| U_t W_t \|^2 \| V_{X^\perp}^T V_{U_t W_t} \|^2$$

$$\overset{(c)}{\leq} 36 \| X \|^2 \| V_{X^\perp}^T V_{U_t W_t} \|^2$$

$$\overset{(d)}{\leq} \frac{1}{4} \sigma_{\min} (X)^2.$$

For inequality $(a)$ we used the submultiplicativity of the spectral norm and the fact that $V_X^T U_t = V_X^T U_t W_t W_t^T$. In $(b)$ we used the assumption $\|V_{X^\perp}^T V_{U_t W_t}\| \leq c\kappa^{-1}$ and $\mu \leq c_1 \|X\|^{-2}\kappa^{-2} \leq c\|V_X^T U_t\|^{-2}/9$. In inequality $(c)$ we used the assumption $\|U_t\| \leq 3\|X\|$. Inequality $(d)$ follows from the assumption $\|V_{X^\perp}^T V_{U_t W_t}\| \leq c\kappa^{-1}$, where the constant $c$ is chosen sufficiently small.

In order to estimate $\|P_2\|$ we note that

$$
\begin{aligned}
\|P_2\| &\overset{(a)}{\leq} \left\| \left[ \left( \mathcal{A}^* \mathcal{A} - \mathrm{Id} \right) \left( XX^T - U_t U_t^T \right) \right] \right\| \left\| \left( V_X^T V_{U_t W_t} \right)^{-1} \right\| \left\| \left( \mathrm{Id} - \mu V_X^T U_t U_t^T V_X \right)^{-1} \right\| \\
&= \frac{\left\| \left[ \left( \mathcal{A}^* \mathcal{A} - \mathrm{Id} \right) \left( XX^T - U_t U_t^T \right) \right] \right\|}{\sigma_{\min} \left( V_X^T V_{U_t W_t} \right) \left( 1 - \mu \|V_X^T U_t\|^2 \right)} \\
&\overset{(b)}{\leq} 4\left\| \left[ \left( \mathcal{A}^* \mathcal{A} - \mathrm{Id} \right) \left( XX^T - U_t U_t^T \right) \right] \right\|.
\end{aligned}
$$

In $(a)$ we used the submultiplicativity of the spectral norm. In inequality $(b)$ we used the assumption $\|V_{X^\perp}^T V_{U_t W_t}\| \leq c\kappa^{-1}$ and $\mu \leq c\|X\|^{-2}\kappa^{-2} \leq c\|V_X^T U_t\|^{-2}/9$. Next, we are going to estimate $\|P_3\|$ by

$$
\begin{aligned}
\|P_3\| &\leq \|\Sigma_X^2\| \|V_X^T U_t W_t\| \left\| \left( \mathrm{Id} - \mu W_t^T U_t^T V_X V_X^T U_t W_t \right)^{-1} \right\| \|W_t^T U_t^T V_X\| \\
&= \frac{\|X\|^2 \|V_X^T U_t W_t\|^2}{1 - \mu \|V_X^T U_t W_t\|^2} \\
&\leq 2\|X\|^2 \|V_X^T U_t W_t\|^2 \\
&\leq 2\|X\|^2 \|U_t W_t\|^2 \\
&\leq 18\|X\|^4.
\end{aligned}
$$

In the last line we used the assumption $\|U_t\| \leq 3\|X\|$. Inserting our estimates for $\|P_1\|$, $\|P_2\|$, and $\|P_3\|$ into (91) we obtain that

$$
\begin{aligned}
\sigma_{\min} \left( V_X^T U_{t+1} W_t \right) &\geq \left( 1 + \frac{3}{4}\mu\sigma_{\min}(X)^2 - 4\mu \left\| \left[ \left( \mathcal{A}^* \mathcal{A} - \mathrm{Id} \right) \left( XX^T - U_t U_t^T \right) \right] \right\| - 18\mu^2 \|X\|^4 \right) \\
&\qquad \sigma_{\min} \left( V_X^T U_t \right) \left( 1 - \mu\sigma_{\min}^2 \left( V_X^T U_t \right) \right) \\
&\overset{(a)}{\geq} \left( 1 + \frac{1}{2}\mu\sigma_{\min}^2(X) \right) \sigma_{\min} \left( V_X^T U_t \right) \left( 1 - \mu\sigma_{\min}^2 \left( V_X^T U_t \right) \right) \\
&= \sigma_{\min} \left( V_X^T U_t \right) \left( 1 + \frac{1}{2}\mu\sigma_{\min}(X)^2 \left( 1 - \mu\sigma_{\min}^2 \left( V_X^T U_t \right) \right) - \mu\sigma_{\min}^2 \left( V_X^T U_t \right) \right) \\
&\overset{(b)}{\geq} \sigma_{\min} \left( V_X^T U_t \right) \left( 1 + \frac{1}{4}\mu\sigma_{\min}^2(X) - \mu\sigma_{\min}^2 \left( V_X^T U_t \right) \right).
\end{aligned}
$$

Inequality $(a)$ follows from assumption (43) and the assumption $\mu \leq c\kappa^{-2}\|X\|^{-2}$. Inequality $(b)$ is a consequence of our assumption on the step size $\mu$ and the assumption $\|U_t\| \leq 3\|X\|$. The final claim follows from the observation that $\sigma_{\min} \left( V_X^T U_{t+1} \right) \geq \sigma_{\min} \left( V_X^T U_{t+1} W_t \right)$. $\qquad \square$

## B.2   Proof of Lemma 9.2

Before we can prove Lemma 9.2, we first need the following technical lemma.

**Lemma B.1.** *Suppose that the assumptions of Lemma 9.2 are fulfilled with a small enough constant $c > 0$. Then we have that*

$$
\|V_{X^\perp}^T V_{U_{t+1} W_t}\| \leq 2\|V_{X^\perp}^T V_{U_t W_t}\| + 2\mu\| \left( \mathcal{A}^* \mathcal{A} \left( XX^T - U_t U_t^T \right) \right)\|. \tag{92}
$$

*In particular, it holds that $\|V_{X^\perp}^T V_{U_{t+1} W_t}\| \leq 1/50$.*

*Proof.* We note that

$$
U_{t+1} W_t = \left( \mathrm{Id} + \mu\mathcal{A}^* \mathcal{A} \left( XX^T - U_t U_t^T \right) \right) U_t W_t.
$$

Let $V_{U_t W_t} \Sigma_{U_t W_t} W_{U_t W}^T = U_t W_t$ be the singular value decomposition of $U_t W_t$. Set

$$
Z := \left( \mathrm{Id} + \mu\mathcal{A}^* \mathcal{A} \left( XX^T - U_t U_t^T \right) \right) V_{U_t W_t}.
$$

Since $\Sigma_{U_t W_t} W_{U_t W}^T$ has full rank by assumption, the matrix $Z = V_Z \Sigma_Z W_Z^T$ has the same column space as the matrix $U_{t+1} W_t$. In particular, it follows that

$$\|V_{X^\perp}^T V_{U_{t+1} W_t}\| = \|V_{X^\perp}^T V_Z\|$$
$$\leq \|V_{X^\perp}^T V_Z \Sigma_Z W_Z^T\| \| \left(\Sigma_Z W_Z^T\right)^{-1}\|$$
$$= \|V_{X^\perp}^T Z\| \|Z^{-1}\|$$
$$= \frac{\|V_{X^\perp}^T Z\|}{\sigma_{\min}(Z)}.$$

By Weyl's inequality it holds that

$$\sigma_{\min}(Z) \geq \sigma_{\min}(V_{U_t W_t}) - \mu \| \left(\mathcal{A}^* \mathcal{A}\left(XX^T - U_t U_t^T\right)\right) V_{U_t W_t}\|$$
$$= 1 - \mu \| \left(\mathcal{A}^* \mathcal{A}\left(XX^T - U_t U_t^T\right)\right) V_{U_t W_t}\|$$
$$= 1 - \mu \| \left(\mathcal{A}^* \mathcal{A}\left(XX^T - U_t U_t^T\right)\right)\|$$
$$\geq 1/2,$$

where in the last inequality we used the assumption on the step size $\mu$. Moreover, note that

$$\|V_{X^\perp}^T Z\| \leq \|V_{X^\perp}^T V_{U_t W_t}\| + \mu\| \left(\mathcal{A}^* \mathcal{A}\left(XX^T - U_t U_t^T\right)\right)\|.$$

This implies inequality (92). Using the assumptions on $\|V_{X^\perp}^T V_{U_t W_t}\|$ and $\mu$, where the constant $c$ is chosen small enough, it follows that $\|V_{X^\perp}^T V_{U_{t+1} W_t}\| \leq 1/50$, which finishes the proof. $\qquad\square$

With all ingredients in place, we can give a proof of Lemma 9.2.

*Proof of Lemma 9.2.* As a first step we are going to establish a formula for $W_t^T W_{t+1,\perp}$. Recall that $V_X^T U_{t+1} W_{t+1,\perp} = 0$ due to the definition of $W_{t+1,\perp}$. Since $W_t W_t^T + W_{t,\perp} W_{t,\perp}^T = \mathrm{Id}$ we obtain that

$$V_X^T U_{t+1} W_t W_t^T W_{t+1,\perp} = -V_X^T U_{t+1} W_{t,\perp} W_{t,\perp}^T W_{t+1,\perp},$$

or, equivalently,

$$W_t^T W_{t+1,\perp} = -\left(V_X^T U_{t+1} W_t\right)^{-1} V_X^T U_{t+1} W_{t,\perp} W_{t,\perp}^T W_{t+1,\perp}. \tag{93}$$

Now recall that we want to bound $\|U_{t+1} W_{t+1,\perp}\|$ from above. Note that using $V_X^T U_{t+1} W_{t+1,\perp} = 0$ we have

$$U_{t+1} W_{t+1,\perp} = V_X V_X^T U_{t+1} W_{t+1,\perp} + V_{X^\perp} V_{X^\perp}^T U_{t+1} W_{t+1,\perp} = V_{X^\perp} V_{X^\perp}^T U_{t+1} W_{t+1,\perp},$$

which implies that $\|U_{t+1} W_{t+1,\perp}\| = \|V_{X^\perp}^T U_{t+1} W_{t+1,\perp}\|$. Due to $W_t W_t^T + W_{t,\perp} W_{t,\perp}^T = \mathrm{Id}$ we have that

$$V_{X^\perp}^T U_{t+1} W_{t+1,\perp} = \underbrace{V_{X^\perp}^T U_{t+1} W_t W_t^T W_{t+1,\perp}}_{=(a)} + \underbrace{V_{X^\perp}^T U_{t+1} W_{t,\perp} W_{t,\perp}^T W_{t+1,\perp}}_{=(b)}. \tag{94}$$

We are going to consider the two summands individually.

**Summand** $(a)$**:** We note that from (93) it follows that

$$V_{X^\perp}^T U_{t+1} W_t W_t^T W_{t+1,\perp} = -V_{X^\perp}^T U_{t+1} W_t \left(V_X^T U_{t+1} W_t\right)^{-1} V_X^T U_{t+1} W_{t,\perp} W_{t,\perp}^T W_{t+1,\perp}.$$

Let the singular value decomposition of $U_{t+1} W_t \in \mathbb{R}^{n \times r_\star}$ be given by $V_{U_{t+1} W_t} \Sigma_{U_{t+1} W_t} W_{U_{t+1} W_t}^T$ with $V_{U_{t+1} W_t} \in \mathbb{R}^{n \times r_\star}$. By assumption we have that $V_X^T U_{t+1} W_t$ is invertible, which also implies that $U_{t+1} W_t$ has full-rank. Hence, we can compute that

$$V_{X^\perp}^T U_{t+1} W_t \left(V_X^T U_{t+1} W_t\right)^{-1} = V_{X^\perp}^T V_{U_{t+1} W_t} V_{U_{t+1} W_t}^T U_{t+1} W_t \left(V_X^T V_{U_{t+1} W_t} V_{U_{t+1} W_t}^T U_{t+1} W_t\right)^{-1}$$
$$= V_{X^\perp}^T V_{U_{t+1} W_t} V_{U_{t+1} W_t}^T U_{t+1} W_t \left(V_{U_{t+1} W_t}^T U_{t+1} W_t\right)^{-1} \left(V_X^T V_{U_{t+1} W_t}\right)^{-1}$$
$$= V_{X^\perp}^T V_{U_{t+1} W_t} \left(V_X^T V_{U_{t+1} W_t}\right)^{-1},$$

which shows that

$$V_{X^\perp}^T U_{t+1} W_t W_t^T W_{t+1,\perp} = -V_{X^\perp}^T V_{U_{t+1}W_t} \left(V_X^T V_{U_{t+1}W_t}\right)^{-1} V_X^T U_{t+1} W_{t,\perp} W_{t,\perp}^T W_{t+1,\perp}.$$

Moreover, we note that

$$
\begin{aligned}
V_X^T U_{t+1} W_{t,\perp} &= V_X^T U_t W_{t,\perp} + \mu V_X^T \left[\mathcal{A}^* \mathcal{A}\left(XX^T - U_t U_t^T\right)\right] U_t W_{t,\perp} \\
&= V_X^T U_t W_{t,\perp} + \mu V_X^T \left(XX^T - U_t U_t^T\right) U_t W_{t,\perp} + \mu V_X^T \left[\left(\mathcal{A}^* \mathcal{A} - \mathrm{Id}\right)\left(XX^T - U_t U_t^T\right)\right] U_t W_{t,\perp} \\
&\overset{(a)}{=} -\mu V_X^T U_t U_t^T U_t W_{t,\perp} + \mu V_X^T \left[\left(\mathcal{A}^* \mathcal{A} - \mathrm{Id}\right)\left(XX^T - U_t U_t^T\right)\right] U_t W_{t,\perp} \\
&= \mu V_X^T \left[-U_t U_t^T + \left[\left(\mathcal{A}^* \mathcal{A} - \mathrm{Id}\right)\left(XX^T - U_t U_t^T\right)\right]\right] U_t W_{t,\perp}.
\end{aligned}
$$

In equality $(a)$ we used that $V_X^T U_t W_{t,\perp} = 0$ and $X^T U W_{t,\perp} = 0$, which follows from the definition of $W_{t,\perp}$. Hence, we have shown that

$$
\begin{aligned}
&V_{X^\perp}^T U_{t+1} W_t W_t^T W_{t+1,\perp} \\
=&\mu V_{X^\perp}^T V_{U_{t+1}W_t} \left(V_X^T V_{U_{t+1}W_t}\right)^{-1} V_X^T \left[U_t U_t^T - \left[\left(\mathcal{A}^* \mathcal{A} - \mathrm{Id}\right)\left(XX^T - U_t U_t^T\right)\right]\right] U_t W_{t,\perp} W_{t,\perp}^T W_{t+1,\perp} \\
\overset{(b)}{=}&\mu V_{X^\perp}^T V_{U_{t+1}W_t} \left(V_X^T V_{U_{t+1}W_t}\right)^{-1} \underbrace{V_X^T \left[U_t U_t^T V_{X^\perp} - \left[\left(\mathcal{A}^* \mathcal{A} - \mathrm{Id}\right)\left(XX^T - U_t U_t^T\right)\right] V_{X^\perp}\right]}_{=:M_1} V_{X^\perp}^T U_t W_{t,\perp} W_{t,\perp}^T W_{t+1,\perp} \\
=&\mu V_{X^\perp}^T V_{U_{t+1}W_t} \left(V_X^T V_{U_{t+1}W_t}\right)^{-1} M_1 V_{X^\perp}^T U_t W_{t,\perp} W_{t,\perp}^T W_{t+1,\perp}
\end{aligned}
$$

In equality $(b)$ above we used that $V_X V_X^T U_t W_{t,\perp} = 0$, which is a consequence of $V_X^T U_t W_{t,\perp} = 0$. It follows that

$$
\begin{aligned}
\|V_{X^\perp}^T U_{t+1} W_t W_t^T W_{t+1,\perp}\| &\le \mu \|V_{X^\perp}^T V_{U_{t+1}W_t}\| \|\left(V_X^T V_{U_{t+1}W_t}\right)^{-1}\| \|M_1\| \|V_{X^\perp}^T U_t W_{t,\perp}\| \|W_{t,\perp}^T W_{t+1,\perp}\| \\
&\le \mu \|V_{X^\perp}^T V_{U_{t+1}W_t}\| \|\left(V_X^T V_{U_{t+1}W_t}\right)^{-1}\| \|M_1\| \|V_{X^\perp}^T U_t W_{t,\perp}\| \\
&= \mu \frac{\|V_{X^\perp}^T V_{U_{t+1}W_t}\| \|M_1\| \|V_{X^\perp}^T U_t W_{t,\perp}\|}{\sigma_{\min}\left(V_X^T V_{U_{t+1}W_t}\right)}.
\end{aligned}
$$

In order to proceed we note that by Lemma B.1 it holds that $\|V_{X^\perp}^T V_{U_{t+1}W}\| \le 1/50$. This implies that

$$
\begin{aligned}
\sigma_{\min}\left(V_X^T V_{U_{t+1}W_t}\right) &= \sqrt{\sigma_{\min}\left(V_{U_{t+1}W_t}^T V_X V_X^T V_{U_{t+1}W_t}\right)} \\
&= \sqrt{\sigma_{\min}\left(V_{U_{t+1}W_t}^T \left(\mathrm{Id} - V_{X^\perp} V_{X^\perp}^T\right) V_{U_{t+1}W_t}\right)} \\
&= \sqrt{1 - \|V_{U_{t+1}W_t}^T V_{X^\perp} V_{X^\perp}^T V_{U_{t+1}W_t}\|} \\
&= \sqrt{1 - \|V_{X^\perp}^T V_{U_{t+1}W_t}\|^2} \\
&\ge 1/2.
\end{aligned}
$$

Hence, we have shown that

$$\|V_{X^\perp}^T U_{t+1} W_t W_t^T W_{t+1,\perp}\| \le 2\mu \|V_{X^\perp}^T V_{U_{t+1}W_t}\| \|M_1\| \|V_{X^\perp}^T U_t W_{t,\perp}\|.$$

We can estimate $\|M_1\|$ by

$$
\begin{aligned}
\|M_1\| &\overset{(a)}{\le} \|V_X^T U_t U_t^T V_{X^\perp}\| + \|\left[\left(\mathcal{A}^* \mathcal{A} - \mathrm{Id}\right)\left(XX^T - U_t U_t^T\right)\right] V_{X^\perp}\| \\
&\overset{(b)}{=} \|V_X^T U_t W_t W_t^T U_t^T V_{X^\perp}\| + \|\left[\left(\mathcal{A}^* \mathcal{A} - \mathrm{Id}\right)\left(XX^T - U_t U_t^T\right)\right] V_{X^\perp}\| \\
&\le \|V_X^T U_t W_t\| \|V_{X^\perp}^T U_t W_t\| + \|\left[\left(\mathcal{A}^* \mathcal{A} - \mathrm{Id}\right)\left(XX^T - U_t U_t^T\right)\right] V_{X^\perp}\| \\
&\le \|V_X^T U_t W_t\| \|V_{X^\perp}^T U_t W_t\| + \|\left(\mathcal{A}^* \mathcal{A} - \mathrm{Id}\right)\left(XX^T - U_t U_t^T\right)\| \\
&\le \|V_X^T U_t W_t\| \|V_{X^\perp}^T V_{U_t W_t}\| \|U_t W_t\| + \|\left(\mathcal{A}^* \mathcal{A} - \mathrm{Id}\right)\left(XX^T - U_t U_t^T\right)\| \\
&\le \|V_{X^\perp}^T V_{U_t W_t}\| \|U_t W_t\|^2 + \|\left(\mathcal{A}^* \mathcal{A} - \mathrm{Id}\right)\left(XX^T - U_t U_t^T\right)\|.
\end{aligned}
$$

In inequality $(a)$ we used the triangle inequality and in equality $(b)$ we used that $V_X^T U_t = V_X^T U_t W_t W_t^T$. Hence, we have shown that

$$\|V_{X^\perp}^T U_{t+1} W_t W_t^T W_{t+1,\perp}\|$$

$$\leq 2\mu \left( \|V_{X^\perp}^T V_{U_t W_t}\| \|U_t W_t\|^2 + \|(\mathcal{A}^*\mathcal{A} - \mathrm{Id})(XX^T - U_t U_t^T)\| \right) \|V_{X^\perp}^T V_{U_{t+1} W_t}\| \|V_{X^\perp}^T U_t W_{t,\perp}\|$$

$$\leq 2\mu \left( \|V_{X^\perp}^T V_{U_t W_t}\| \|U_t W_t\|^2 + \|(\mathcal{A}^*\mathcal{A} - \mathrm{Id})(XX^T - U_t U_t^T)\| \right) \|V_{X^\perp}^T V_{U_{t+1} W_t}\| \|U_t W_{t,\perp}\|$$

$$\leq 2\mu \left( 9 \|V_{X^\perp}^T V_{U_t W_t}\| \|X\|^2 + \|(\mathcal{A}^*\mathcal{A} - \mathrm{Id})(XX^T - U_t U_t^T)\| \right) \|V_{X^\perp}^T V_{U_{t+1} W_t}\| \|U_t W_{t,\perp}\|$$

$$\overset{(a)}{\leq} 4\mu \left( 9 \|V_{X^\perp}^T V_{U_t W_t}\| \|X\|^2 + \|(\mathcal{A}^*\mathcal{A} - \mathrm{Id})(XX^T - U_t U_t^T)\| \right)$$
$$\cdot \left( \|V_{X^\perp}^T V_{U_t W_t}\| + \mu \|(\mathcal{A}^*\mathcal{A})(XX^T - U_t U_t^T)\| \right) \|U_t W_{t,\perp}\|$$

$$\leq 4\mu \left( 9 \|V_{X^\perp}^T V_{U_t W_t}\| \|X\|^2 + \|(\mathcal{A}^*\mathcal{A} - \mathrm{Id})(XX^T - U_t U_t^T)\| \right)$$
$$\cdot \left( \|V_{X^\perp}^T V_{U_t W_t}\| + \mu \|(\mathcal{A}^*\mathcal{A} - \mathrm{Id})(XX^T - U_t U_t^T)\| + \mu \|XX^T - U_t U_t^T\| \right) \|U_t W_{t,\perp}\|$$

$$\overset{(b)}{\leq} 4\mu \left( 9 \|V_{X^\perp}^T V_{U_t W_t}\| \|X\|^2 + \|(\mathcal{A}^*\mathcal{A} - \mathrm{Id})(XX^T - U_t U_t^T)\| \right)$$
$$\cdot \left( \|V_{X^\perp}^T V_{U_t W_t}\| + \mu \|(\mathcal{A}^*\mathcal{A} - \mathrm{Id})(XX^T - U_t U_t^T)\| + 10\mu \|X\|^2 \right) \|U_t W_{t,\perp}\|$$

$$\overset{(c)}{\leq} \mu \left( 9 \|V_{X^\perp}^T V_{U_t W_t}\| \|X\|^2 + \|(\mathcal{A}^*\mathcal{A} - \mathrm{Id})(XX^T - U_t U_t^T)\| \right) \|U_t W_{t,\perp}\|,$$

where in inequality $(a)$ we used Lemma B.1. For inequality $(b)$ we used the assumption $\|U_t\| \leq 3\|X\|$ and for inequality $(c)$ we used assumption $\|V_{X^\perp}^T V_{U_t W_t}\| \leq c\kappa^{-1}$ and the assumption on the step size $\mu$ with a small enough constant $c > 0$.

**Summand $(b)$:** First, we compute that

$$V_{X^\perp}^T U_{t+1} W_{t,\perp}$$

$$= V_{X^\perp}^T U_t W_{t,\perp} + \mu V_{X^\perp}^T (XX^T - U_t U_t^T) U_t W_{t,\perp} + \mu V_{X^\perp}^T \left[ (\mathcal{A}^*\mathcal{A} - \mathrm{Id})(XX^T - U_t U_t^T) \right] U_t W_{t,\perp}$$

$$= V_{X^\perp}^T U_t W_{t,\perp} - \mu V_{X^\perp}^T U_t U_t^T U_t W_{t,\perp} + \mu V_{X^\perp}^T \left[ (\mathcal{A}^*\mathcal{A} - \mathrm{Id})(XX^T - U_t U_t^T) \right] U_t W_{t,\perp}$$

$$= V_{X^\perp}^T U_t W_{t,\perp} - \mu V_{X^\perp}^T U_t U_t^T V_{X^\perp} V_{X^\perp}^T U_t W_{t,\perp} + \mu V_{X^\perp}^T \left[ (\mathcal{A}^*\mathcal{A} - \mathrm{Id})(XX^T - U_t U_t^T) \right] U_t W_{t,\perp}$$

$$= \left( \mathrm{Id} - \mu V_{X^\perp}^T U_t U_t^T V_{X^\perp} - \mu V_{X^\perp}^T \left[ (\mathcal{A}^*\mathcal{A} - \mathrm{Id})(XX^T - U_t U_t^T) \right] V_{X^\perp} \right) V_{X^\perp}^T U_t W_{t,\perp}$$

$$= V_{X^\perp}^T U_t W_{t,\perp} - \mu V_{X^\perp}^T U_t W_{t,\perp} W_{t,\perp}^T U_t^T V_{X^\perp} V_{X^\perp}^T U_t W_{t,\perp} - \mu V_{X^\perp}^T U_t W_t W_t^T U_t^T V_{X^\perp} V_{X^\perp}^T U_t W_{t,\perp}$$
$$+ \mu V_{X^\perp}^T \left[ (\mathcal{A}^*\mathcal{A} - \mathrm{Id})(XX^T - U_t U_t^T) \right] V_{X^\perp} V_{X^\perp}^T U_t W_{t,\perp}$$

$$= \left( \mathrm{Id} - \mu V_{X^\perp}^T U_t W_t W_t^T U_t^T V_{X^\perp} + \mu V_{X^\perp}^T \left[ (\mathcal{A}^*\mathcal{A} - \mathrm{Id})(XX^T - U_t U_t^T) V_{X^\perp} \right] \right) V_{X^\perp}^T U_t W_{t,\perp} \left( \mathrm{Id} - \mu W_{t,\perp}^T U_t^T U_t W_{t,\perp} \right)$$
$$- \mu^2 \left( V_{X^\perp}^T U_t W_t W_t^T U_t^T V_{X^\perp} - V_{X^\perp}^T \left[ (\mathcal{A}^*\mathcal{A} - \mathrm{Id})(XX^T - U_t U_t^T) V_{X^\perp} \right] \right) V_{X^\perp}^T U_t W_{t,\perp} W_{t,\perp}^T U_t^T U_t W_{t,\perp}.$$

Set for brevity of notation $M_2 := V_{X^\perp}^T U_t W_t W_t^T U_t^T V_{X^\perp}$ and $M_3 := V_{X^\perp}^T (\mathcal{A}^*\mathcal{A} - \mathrm{Id})(XX^T - U_t U_t^T) V_{X^\perp}$. Hence, we have computed that

$$V_{X^\perp}^T U_{t+1} W_{t,\perp}$$

$$= (\mathrm{Id} - \mu M_2 + \mu M_3) V_{X^\perp}^T U_t W_{t,\perp} \left( \mathrm{Id} - \mu W_{t,\perp}^T U_t^T U_t W_{t,\perp} \right) - \mu^2 (M_2 - M_3) V_{X^\perp}^T U_t W_{t,\perp} W_{t,\perp}^T U_t^T U_t W_{t,\perp}$$

$$= (\mathrm{Id} - \mu M_2 + \mu M_3) V_{X^\perp}^T U_t W_{t,\perp} \left( \mathrm{Id} - \mu W_{t,\perp}^T U_t^T V_{X^\perp} V_{X^\perp}^T U_t W_{t,\perp} \right) - \mu^2 (M_2 - M_3) V_{X^\perp}^T U_t W_{t,\perp} W_{t,\perp}^T U_t^T U_t W_{t,\perp}.$$

Hence, we obtain that

$$\|V_{X^\perp}^T U_{t+1} W_{t,\perp} W_{t,\perp}^T W_{t+1,\perp}\|$$

$$\leq \| (\mathrm{Id} - \mu M_2 + \mu M_3) V_{X^\perp}^T U_t W_{t,\perp} \left( \mathrm{Id} - \mu W_{t,\perp}^T U_t^T V_{X^\perp} V_{X^\perp}^T U_t W_{t,\perp} \right)$$
$$- \mu^2 (M_2 - M_3) V_{X^\perp}^T U_t W_{t,\perp} W_{t,\perp}^T U_t^T U_t W_{t,\perp} \| \|W_{t,\perp}^T W_{t+1,\perp}\|$$

$$\leq \| (\mathrm{Id} - \mu M_2 + \mu M_3) V_{X^\perp}^T U_t W_{t,\perp} \left( \mathrm{Id} - \mu W_{t,\perp}^T U_t^T V_{X^\perp} V_{X^\perp}^T U_t W_{t,\perp} \right)$$
$$- \mu^2 (M_2 - M_3) V_{X^\perp}^T U_t W_{t,\perp} W_{t,\perp}^T U_t^T U_t W_{t,\perp} \|$$

$$\leq \| (\mathrm{Id} - \mu M_2 + \mu M_3) V_{X^\perp}^T U_t W_{t,\perp} \left( \mathrm{Id} - \mu W_{t,\perp}^T U_t^T V_{X^\perp} V_{X^\perp}^T U_t W_{t,\perp} \right) \|$$
$$+ \mu^2 \| (M_2 - M_3) V_{X^\perp}^T U_t W_{t,\perp} W_{t,\perp}^T U_t^T U_t W_{t,\perp} \|.$$

In order to proceed, we compute that

$$\| \left( \mathrm{Id} - \mu M_2 + \mu M_3 \right) V_{X^\perp}^T U_t W_{t,\perp} \left( \mathrm{Id} - \mu W_{t,\perp}^T U_t^T V_{X^\perp} V_{X^\perp}^T U_t W_{t,\perp} \right) \|$$

$$\overset{(a)}{\leq} \| \mathrm{Id} - \mu M_2 + \mu M_3 \| \| V_{X^\perp}^T U_t W_{t,\perp} \left( \mathrm{Id} - \mu W_{t,\perp}^T U_t^T V_{X^\perp} V_{X^\perp}^T U_t W_{t,\perp} \right) \|$$

$$\overset{(b)}{=} \| \mathrm{Id} - \mu M_2 + \mu M_3 \| \| V_{X^\perp}^T U_t W_{t,\perp} \| \left( 1 - \mu \| V_{X^\perp}^T U_t W_{t,\perp} \|^2 \right)$$

$$\overset{(c)}{\leq} \left( \| \mathrm{Id} - \mu M_2 \| + \mu \| M_3 \| \right) \| V_{X^\perp}^T U_t W_{t,\perp} \| \left( 1 - \mu \| V_{X^\perp}^T U_t W_{t,\perp} \|^2 \right)$$

$$\overset{(d)}{\leq} \left( 1 + \mu \| M_3 \| \right) \| V_{X^\perp}^T U_t W_{t,\perp} \| \left( 1 - \mu \| V_{X^\perp}^T U_t W_{t,\perp} \|^2 \right)$$

$$= \left( 1 + \mu \| M_3 \| \right) \| U_t W_{t,\perp} \| \left( 1 - \mu \| U_t W_{t,\perp} \|^2 \right)$$

$$\leq \| U_t W_{t,\perp} \| \left( 1 - \mu \| U W_{t,\perp} \|^2 + \mu \| \left( \mathcal{A}^* \mathcal{A} - \mathrm{Id} \right) \left( X X^T - U_t U_t^T \right) V_{X^\perp} \| - \mu^2 \| M_3 \| \| U_t W_{t,\perp} \|^2 \right)$$

$$\leq \| U_t W_{t,\perp} \| \left( 1 - \mu \| U_t W_{t,\perp} \|^2 + \mu \| \left( \mathcal{A}^* \mathcal{A} - \mathrm{Id} \right) \left( X X^T - U_t U_t^T \right) V_{X^\perp} \| \right)$$

$$\leq \| U_t W_{t,\perp} \| \left( 1 - \mu \| U_t W_{t,\perp} \|^2 + \mu \| \left( \mathcal{A}^* \mathcal{A} - \mathrm{Id} \right) \left( X X^T - U_t U_t^T \right) \| \right).$$

The inequality $(a)$ follows from the submultiplicativity of the spectral norm. Equality $(b)$ can be seen be using the singular value decomposition of $V_{X^\perp}^T U_t W_{t,\perp}$ and the assumption $\mu \leq c_1 \| X \|^{-2} \leq 1/(\sqrt{3} \| V_{X^\perp}^T U_t W_{t,\perp} \|^2)$. Inequality $(c)$ follows from the triangle inequality. For inequality $(d)$ we used that $0 \leq \mathrm{Id} - \mu V_{X^\perp}^T U_t W_t W_t^T U_t^T V_{X^\perp} \leq \mathrm{Id}$, which again is a consequence of our assumptions on $\mu$ and $\| U_t \| \leq 3 \| X \|$. For the $O\left( \mu^2 \right)$-term we note that

$$\| \left( M_2 - M_3 \right) V_{X^\perp}^T U_t W_{t,\perp} W_{t,\perp}^T U_t^T U_t W_{t,\perp} \|$$

$$= \| \left( M_2 - M_3 \right) V_{X^\perp}^T U_t W_{t,\perp} W_{t,\perp}^T U_t^T V_{X^\perp} V_{X^\perp}^T U_t W_{t,\perp} \|$$

$$\leq \| M_2 - M_3 \| \| V_{X^\perp}^T U_t W_{t,\perp} \|^3$$

$$\leq \left( \| M_2 \| + \| M_3 \| \right) \| V_{X^\perp}^T U_t W_{t,\perp} \|^3$$

$$\leq \left( \| V_{X^\perp}^T U_t W_t \|^2 + \| M_3 \| \right) \| V_{X^\perp}^T U_t W_{t,\perp} \|^3$$

$$\leq \left( \| V_{X^\perp}^T V_{U_t W_t} \|^2 \| U_t W_t \|^2 + \| M_3 \| \right) \| V_{X^\perp}^T U_t W_{t,\perp} \|^3$$

$$= \left( \| V_{X^\perp}^T V_{U_t W_t} \|^2 \| U_t W_t \|^2 + \| M_3 \| \right) \| U_t W_{t,\perp} \|^3$$

$$= \left( \| V_{X^\perp}^T V_{U_t W_t} \|^2 \| U_t W_t \|^2 + \| \left( \mathcal{A}^* \mathcal{A} - \mathrm{Id} \right) \left( X X^T - U_t U_t^T \right) V_{X^\perp} \| \right) \| U_t W_{t,\perp} \|^3$$

$$\leq \left( \| V_{X^\perp}^T V_{U_t W_t} \|^2 \| U_t W_t \|^2 + \| \left( \mathcal{A}^* \mathcal{A} - \mathrm{Id} \right) \left( X X^T - U_t U_t^T \right) \| \right) \| U_t W_{t,\perp} \|^3.$$

It follows that

$$\mu^2 \| \left( M_2 - M_3 \right) V_{X^\perp}^T U_t W_{t,\perp} W_{t,\perp}^T U_t^T U_t W_{t,\perp} \|$$

$$\leq \mu^2 \left( \| V_{X^\perp}^T V_{U_t W_t} \|^2 \| U_t W_t \|^2 + \| \left( \mathcal{A}^* \mathcal{A} - \mathrm{Id} \right) \left( X X^T - U_t U_t^T \right) \| \right) \| U_t W_{t,\perp} \|^3$$

$$\overset{(a)}{\leq} \mu^2 \left( 9 \| X \|^2 + \| \left( \mathcal{A}^* \mathcal{A} - \mathrm{Id} \right) \left( X X^T - U_t U_t^T \right) \| \right) \| U_t W_{t,\perp} \|^3$$

$$\overset{(b)}{\leq} \frac{\mu}{2} \| U_t W_{t,\perp} \|^3,$$

In $(a)$ we used that $\| U_t \| \leq 3 \| X \|$. In $(b)$ we used our assumption on the step size $\mu$. This implies that

$$\| (b) \| \leq \| U_t W_{t,\perp} \| \left( 1 - \frac{\mu}{2} \| U_t W_{t,\perp} \|^2 + \mu \| \left( \mathcal{A}^* \mathcal{A} - \mathrm{Id} \right) \left( X X^T - U_t U_t^T \right) \| \right).$$

**Conclusion:** Putting things together it follows

$$\| V_{X^\perp}^T U_{t+1} W_{t+1,\perp} \|$$

$$\leq \| (a) \| + \| (b) \|$$

$$\leq \left( 1 - \frac{\mu}{2} \| U_t W_{t,\perp} \|^2 + 9 \mu \| V_{X^\perp}^T V_{U_t W_t} \| \| X \|^2 + 2 \mu \| \left( \mathcal{A}^* \mathcal{A} - \mathrm{Id} \right) \left( X X^T - U_t U_t^T \right) \| \right) \| U_t W_{t,\perp} \|.$$

This finishes the proof. □

## B.3 Proof of Lemma 9.3

We define the inverse of the square root of a symmetric, positiv definite matrix $A = V_A \Sigma_A V_A^T$ by $A^{-1/2} := V_A \Sigma_A^{-1/2} V_A^T$, where $\left(\Sigma_A^{-1/2}\right)_{ii} := \frac{1}{\sqrt{A_{ii}}}$. We will need the following technical lemma, which gives a bound on the first order Taylor-approximation of the matrix inverse square root.

**Lemma B.2.** *Let $A$ be a symmetric, positiv definite matrix such that $\|A\| \leq 1/2$. Then there holds*

$$\left\| (Id + A)^{-1/2} - Id + \frac{1}{2}A \right\| \leq 3\|A\|^2.$$

*Proof.* Since $A$ is symmetric, this can be readily deduced from the (one-dimensional) Taylor's theorem. Indeed, we have that for $|x| \leq 1/2$ that

$$\left| \frac{1}{\sqrt{1+x}} - 1 + \frac{x}{2} \right| \leq \sup_{|z| \leq 1/2} \left| \frac{3}{8} \cdot (1+z)^{-5/2} \cdot x^2 \right|$$
$$\leq 3x^2.$$

$\square$

The next technical lemma shows that the orthogonal matrices $W_t$ and $W_{t+1}$ span approximately the same column space.

**Lemma B.3.** *Assume that the assumptions of Lemma 9.3 are fulfilled. Then it holds that*

$$\|W_{t,\perp}^T W_{t+1}\| \leq \mu \left( \frac{1}{6400} \sigma_{\min}(X)^2 + \|U_t W_t\| \|U_t W_{t,\perp}\| \right) \|V_{X^\perp}^T V_{U_t W_t}\| + 4\mu \left\| \left[ (Id - \mathcal{A}^* \mathcal{A}) \left( XX^T - U_t U_t^T \right) \right] \right\| \tag{95}$$

*and*

$$\sigma_{\min} \left( W_t^T W_{t+1} \right) \geq 1/2.$$

*Proof.* Due to $V_X^T U_{t+1} = V_X^T U_{t+1} W_{t+1} W_{t+1}^T$ we observe that

$$\|W_{t,\perp}^T W_{t+1}\| = \|W_{t,\perp}^T U_{t+1}^T V_X \left( V_X^T U_{t+1} U_{t+1}^T V_X \right)^{-1/2} \|.$$

We note that

$V_X^T U_{t+1} W_{t,\perp}$
$= V_X^T \left( Id + \mu \left( XX^T - U_t U_t^T \right) \right) U_t W_{t,\perp} - \mu V_X^T \left[ (Id - \mathcal{A}^* \mathcal{A}) \left( XX^T - U_t U_t^T \right) \right] U_t W_{t,\perp}$
$= -\mu V_X^T U_t U_t^T U_t W_{t,\perp} - \mu V_X^T \left[ (Id - \mathcal{A}^* \mathcal{A}) \left( XX^T - U_t U_t^T \right) \right] U_t W_{t,\perp}$
$= -\mu V_X^T U_t W_t W_t^T U_t^T U_t W_{t,\perp} - \mu V_X^T \left[ (Id - \mathcal{A}^* \mathcal{A}) \left( XX^T - U_t U_t^T \right) \right] U_t W_{t,\perp}$
$= -\mu V_X^T U_t W_t W_t^T U_t^T V_{X^\perp} V_{X^\perp}^T U_t W_{t,\perp} - \mu V_X^T \left[ (Id - \mathcal{A}^* \mathcal{A}) \left( XX^T - U_t U_t^T \right) \right] U_t W_{t,\perp}$
$= -\mu V_X^T U_t W_t W_t^T U_t^T V_{U_t W_t} V_{U_t W_t}^T V_{X^\perp} V_{X^\perp}^T U_t W_{t,\perp} - \mu V_X^T \left[ (Id - \mathcal{A}^* \mathcal{A}) \left( XX^T - U_t U_t^T \right) \right] U_t W_{t,\perp}.$

It follows that

$$\|W_{t,\perp}^T W_{t+1}\| \leq \mu \| \left( V_X^T U_{t+1} U_{t+1}^T V_X \right)^{-1/2} V_X^T U_t W_t W_t^T U_t^T V_{U_t W_t} V_{U_t W_t}^T V_{X^\perp} V_{X^\perp}^T U_t W_{t,\perp} \|$$
$$+ \mu \| \left[ (Id - \mathcal{A}^* \mathcal{A}) \left( XX^T - U_t U_t^T \right) \right] \| \|U_t W_{t,\perp}\| \| \left( V_X^T U_{t+1} U_{t+1}^T V_X \right)^{-1/2} \|$$
$$= \mu \| \left( V_X^T U_{t+1} U_{t+1}^T V_X \right)^{-1/2} V_X^T U_t W_t W_t^T U_t^T V_{U_t W_t} V_{U_t W_t}^T V_{X^\perp} V_{X^\perp}^T U_t W_{t,\perp} \|$$
$$+ \mu \frac{\| \left[ (Id - \mathcal{A}^* \mathcal{A}) \left( XX^T - U_t U_t^T \right) \right] \| \|U_t W_{t,\perp}\|}{\sigma_{\min} \left( V_X^T U_{t+1} \right)}$$
$$\leq \mu \| \left( V_X^T U_{t+1} U_{t+1}^T V_X \right)^{-1/2} V_X^T U_t W_t \| \|U_t W_t\| \|U_t W_{t,\perp}\| \|V_{X^\perp}^T V_{U_t W_t}\|$$
$$+ \mu \frac{\| \left[ (Id - \mathcal{A}^* \mathcal{A}) \left( XX^T - U_t U_t^T \right) \right] \| \|U_t W_{t,\perp}\|}{\sigma_{\min} \left( V_X^T U_{t+1} \right)}.$$

We note that

$$\left(V_X^T U_{t+1} U_{t+1}^T V_X\right)^{-1/2} V_X^T U_t W_t = \left(V_X^T U_{t+1} U_{t+1}^T V_X\right)^{-1/2} V_X^T U_{t+1} W_t$$
$$- \mu \left(V_X^T U_{t+1} U_{t+1}^T V_X\right)^{-1/2} V_X^T \mathcal{A}^* \mathcal{A} \left(X X^T - U_t U_t^T\right) U_t W_t.$$

It follows that

$$\left\| \left(V_X^T U_{t+1} U_{t+1}^T V_X\right)^{-1/2} V_X^T U_t W_t \right\|$$
$$\leq \left\| \left(V_X^T U_{t+1} U_{t+1}^T V_X\right)^{-1/2} V_X^T U_{t+1} W_t \right\| + \mu \left\| \left(V_X^T U_{t+1} U_{t+1}^T V_X\right)^{-1/2} V_X^T \mathcal{A}^* \mathcal{A} \left(X X^T - U_t U_t^T\right) U_t W_t \right\|$$
$$\leq \left\| \left(V_X^T U_{t+1} U_{t+1}^T V_X\right)^{-1/2} V_X^T U_{t+1} \right\| + \mu \left\| \left(V_X^T U_{t+1} U_{t+1}^T V_X\right)^{-1/2} V_X^T \mathcal{A}^* \mathcal{A} \left(X X^T - U_t U_t^T\right) U_t W_t \right\|$$
$$= 1 + \mu \left\| \left(V_X^T U_{t+1} U_{t+1}^T V_X\right)^{-1/2} V_X^T \mathcal{A}^* \mathcal{A} \left(X X^T - U_t U_t^T\right) U_t W_t \right\|$$
$$\leq 1 + \mu \frac{\left\| \mathcal{A}^* \mathcal{A} \left(X X^T - U_t U_t^T\right) \right\| \|U_t W_t\|}{\sigma_{\min}\left(V_X^T U_{t+1}\right)}.$$

Hence, we obtain that

$$\|W_{t,\perp}^T W_{t+1}\| \leq \mu \left(1 + \mu \frac{\left\| \mathcal{A}^* \mathcal{A} \left(X X^T - U_t U_t^T\right) \right\| \|U_t W_t\|}{\sigma_{\min}\left(V_X^T U_{t+1}\right)}\right) \|U_t W_t\| \|U_t W_{t,\perp}\| \|V_{X^\perp}^T V_{U_t W_t}\|$$
$$+ \mu \frac{\left\| \left(\mathrm{Id} - \mathcal{A}^* \mathcal{A}\right) \left(X X^T - U_t U_t^T\right) \right\| \|U_t W_{t,\perp}\|}{\sigma_{\min}\left(V_X^T U_{t+1}\right)}. \tag{96}$$

Next, we are going to show $\sigma_{\min}\left(V_X^T U_{t+1}\right) \geq \frac{\sigma_{\min}(U_t W_t)}{2}$. We note that

$$\sigma_{\min}\left(V_X^T U_{t+1}\right) \geq \sigma_{\min}\left(V_X^T U_{t+1} W_t\right)$$
$$= \sigma_{\min}\left(V_X^T \left(\mathrm{Id} + \mu \left[(\mathcal{A}^* \mathcal{A})\left(X X^T - U_t U_t^T\right)\right]\right) U_t W_t\right)$$
$$= \sigma_{\min}\left(V_X^T \left(\mathrm{Id} + \mu \left[(\mathcal{A}^* \mathcal{A})\left(X X^T - U_t U_t^T\right)\right]\right) V_{U_t W_t} V_{U_t W_t}^T U_t W_t\right)$$
$$\geq \sigma_{\min}\left(V_X^T \left(\mathrm{Id} + \mu \left[(\mathcal{A}^* \mathcal{A})\left(X X^T - U_t U_t^T\right)\right]\right) V_{U_t W_t}\right) \sigma_{\min}\left(V_{U_t W_t}^T U_t W_t\right)$$
$$= \sigma_{\min}\left(V_X^T V_{U_t W_t} + \mu V_X^T \left[(\mathcal{A}^* \mathcal{A})\left(X X^T - U_t U_t^T\right)\right] V_{U_t W_t}\right) \sigma_{\min}\left(U_t W_t\right)$$
$$\geq \left(\sigma_{\min}\left(V_X^T V_{U_t W_t}\right) - \mu \|V_X^T \left[(\mathcal{A}^* \mathcal{A})\left(X X^T - U_t U_t^T\right)\right] V_{U_t W_t}\|\right) \sigma_{\min}\left(U_t W_t\right).$$

We observe that due to our assumption on $\|V_{X^\perp}^T V_{U_t W_t}\|$ we have that

$$\sigma_{\min}\left(V_X^T V_{U_t W_t}\right) = \sqrt{1 - \|V_{X^\perp}^T V_{U_t W_t}\|^2} \geq \frac{3}{4}.$$

Next, we note that due to assumptions (44), (46) and $\|U_t\| \leq 3\|X\|$ we have that

$$\mu \|V_X^T \left[(\mathcal{A}^* \mathcal{A})\left(X X^T - U_t U_t^T\right)\right] V_{U_t W_t}\| \leq \mu \|(\mathcal{A}^* \mathcal{A})\left(X X^T - U_t U_t^T\right)\|$$
$$\leq 10\mu \|X\|^2 + \mu \|\left(\mathrm{Id} - \mathcal{A}^* \mathcal{A}\right)\left(X X^T - U_t U_t^T\right)\| \leq \frac{1}{4}.$$

Hence, we have shown that $\sigma_{\min}\left(V_X^T U_{t+1}\right) \geq \frac{\sigma_{\min}(U_t W_t)}{2}$. This implies due to inequality (96) that

$$\|W_{t,\perp}^T W_{t+1}\| \leq \mu \left(1 + 2\mu \frac{\left\| \mathcal{A}^* \mathcal{A} \left(X X^T - U_t U_t^T\right) \right\| \|U_t W_t\|}{\sigma_{\min}\left(U_t W_t\right)}\right) \|U_t W_t\| \|U_t W_{t,\perp}\| \|V_{X^\perp}^T V_{U_t W_t}\|$$
$$+ 2\mu \frac{\left\| \left(\mathrm{Id} - \mathcal{A}^* \mathcal{A}\right)\left(X X^T - U_t U_t^T\right) \right\| \|U_t W_{t,\perp}\|}{\sigma_{\min}\left(U_t W_t\right)}$$
$$\overset{(a)}{\leq} \mu \|V_{X^\perp}^T V_{U_t W_t}\| \|U_t W_t\| \|U_t W_{t,\perp}\| + 4\mu \|\left(\mathrm{Id} - \mathcal{A}^* \mathcal{A}\right)\left(X X^T - U_t U_t^T\right)\|$$
$$+ 4\mu^2 \|(\mathcal{A}^* \mathcal{A})\left(X X^T - U_t U_t^T\right)\| \|U_t W_t\|^2 \|V_{X^\perp}^T V_{U_t W_t}\|.$$

In inequality $(a)$ we have used the assumption that $\|U_t W_{t,\perp}\| \le 2\sigma_{\min}(U_t W_t)$. In order to proceed, we note that

$$\|(\mathcal{A}^*\mathcal{A})(XX^T - U_t U_t^T)\| \le \|XX^T - U_t U_t^T\| + \|(\mathrm{Id} - \mathcal{A}^*\mathcal{A})(XX^T - U_t U_t^T)\|$$
$$\le 11\|X\|^2,$$

where we used the assumption $\|U_t W_t\| \le 3\|X\|$ and $\|[(\mathrm{Id} - \mathcal{A}^*\mathcal{A})(XX^T - U_t U_t^T)]\| \le c\sigma_{\min}(X)^2$. Hence, we obtain that

$$\|W_{t,\perp}^T W_{t+1}\| \le \mu\left(\frac{1}{6400}\sigma_{\min}(X)^2 + \|U_t W_t\|\|U_t W_{t,\perp}\|\right)\|V_{X^\perp}^T V_{U_t W_t}\| + 4\mu\|(\mathrm{Id} - \mathcal{A}^*\mathcal{A})(XX^T - U_t U_t^T)\|,$$

where we also used $\mu \le c\kappa^{-2}\|X\|^{-2}$. Hence, we have shown inequality (95).

In order to finish the proof we note that

$$
\|W_{t,\perp}^T W_{t+1}\| \overset{(a)}{\le} \mu\left(\frac{1}{6400}\sigma_{\min}(X)^2 + \|U_t W_t\|\|U_t W_{t,\perp}\|\right)\|V_{X^\perp}^T V_{U_t W_t}\| + 4\mu c\sigma_{\min}(X)^2
$$
$$
\overset{(b)}{\le} \mu\left(\frac{1}{6400}\sigma_{\min}(X)^2 + 9\|X\|^2\right)\|V_{X^\perp}^T V_{U_t W_t}\| + 4c\mu\sigma_{\min}(X)^2
$$
$$
\overset{(c)}{\lesssim} c.
$$

In inequality $(a)$ we used the assumption $\|(\mathrm{Id} - \mathcal{A}^*\mathcal{A})(XX^T - U_t U_t^T)\| \le c\sigma_{\min}(X)^2$. In inequality $(b)$ we used $\|U_t W_{t,\perp}\| \le 3\|X\|$ and $\|U_t W_t\| \le 3\|X\|$. To obtain inequality $(c)$ we used the assumption $\mu \le c\|X\|^{-2}$. By choosing $c > 0$ small enough we obtain that $\|W_{t,\perp}^T W_{t+1}\| \le 1/2$. Note that this implies that

$$\sigma_{\min}(W_t^T W_{t+1}) = \sqrt{1 - \|W_{t,\perp}^T W_{t+1}\|^2} \ge 1/2,$$

which finishes the proof. $\qquad\square$

Now we have provided all the technical preliminaries to prove Lemma 9.3.

*Proof of Lemma 9.3.* In order to simplify notation, we define $M := \mathcal{A}^*\mathcal{A}(XX^T - U_t U_t^T)$. Hence, we may write

$$U_{t+1} = (\mathrm{Id} + \mu M)U_t.$$

Now we note that

$$
\begin{aligned}
U_{t+1}W_{t+1} &= (\mathrm{Id} + \mu M)U_t W_{t+1}\\
&= (\mathrm{Id} + \mu M)U_t W_t W_t^T W_{t+1} + (\mathrm{Id} + \mu M)U_t W_{t,\perp}W_{t,\perp}^T W_{t+1}\\
&= (\mathrm{Id} + \mu M)V_{U_t W}V_{U_t W}^T U_t W_t W_t^T W_{t+1} + (\mathrm{Id} + \mu M)U_t W_{t,\perp}W_{t,\perp}^T W_{t+1}.
\end{aligned}
\tag{97}
$$

Note that $V_{U_t W_t}^T U_t W_t W_t^T W_{t+1}$ is invertible, since $V_{U_t W_t}^T U_t W_t$ is invertible by assumption (45) and $W_t^T W_{t+1}$ is invertible by Lemma B.3. Hence, we see that

$$
\begin{aligned}
&(\mathrm{Id} + \mu M)U_t W_{t,\perp}W_{t,\perp}^T W_{t+1}\\
&= (\mathrm{Id} + \mu M)U_t W_{t,\perp}W_{t,\perp}^T W_{t+1}\left(V_{U_t W_t}^T U_t W_t W_t^T W_{t+1}\right)^{-1}V_{U_t W_t}^T U_t W_t W_t^T W_{t+1}\\
&= (\mathrm{Id} + \mu M)\underbrace{U_t W_{t,\perp}W_{t,\perp}^T W_{t+1}\left(V_{U_t W_t}^T U_t W_t W_t^T W_{t+1}\right)^{-1}V_{U_t W_t}^T}_{=P}V_{U_t W_t}V_{U_t W_t}^T U_t W_t W_t^T W_{t+1}\\
&= (\mathrm{Id} + \mu M)PV_{U_t W_t}V_{U_t W_t}^T U_t W_t W_t^T W_{t+1}.
\end{aligned}
$$

Hence, by inserting the last equation into equation (97) we obtain that

$$U_{t+1}W_{t+1} = (\mathrm{Id} + \mu M)(\mathrm{Id} + P)V_{U_t W_t}V_{U_t W_t}^T U_t W_t W_t^T W_{t+1}$$

Recall that $V_{U_t W_t}^T U_t W_t W_t^T W_{t+1}$ is an invertible matrix. This implies that the span of the left-singular vectors of

$$Z := (\mathrm{Id} + \mu M) (\mathrm{Id} + P) V_{U_t W_t}$$

is the same as the span of the left-singular vectors of $U_{t+1} W_{t+1}$. Let $V_Z \Sigma_Z W_Z^T$ be the singular value decomposition of $Z$. From these considerations it follows that

$$\| V_{X^\perp}^T V_{U_{t+1} W_{t+1}} \| = \| V_{X^\perp}^T V_Z \| = \| V_{X^\perp}^T V_Z W_Z^T \|.$$

Next, we note that

$$V_Z W_Z^T = Z \left( Z^T Z \right)^{-1/2}$$
$$= (\mathrm{Id} + \mu M)(\mathrm{Id} + P) V_{U_t W_t} \left( V_{U_t W_t}^T \left( \mathrm{Id} + P^T \right) (\mathrm{Id} + \mu M)^2 (\mathrm{Id} + P) V_{U_t W_t} \right)^{-1/2}.$$

We note that

$$(\mathrm{Id} + \mu M)(\mathrm{Id} + P) = \mathrm{Id} + \underbrace{\mu M + P + \mu M P}_{=:B}$$
$$= \mathrm{Id} + \underbrace{\mu \left( X X^T - U_t U_t^T \right)}_{=:B_1} + \underbrace{\mu \left( \mathcal{A}^* \mathcal{A} - \mathrm{Id} \right) \left( X X^T - U_t U_t^T \right)}_{=:B_2}$$
$$+ \underbrace{U_t W_{t,\perp} W_{t,\perp}^T W_{t+1} \left( V_{U_t W_t}^T U_t W_t W_t^T W_{t+1} \right)^{-1} V_{U_t W_t}^T}_{=:B_3} + \underbrace{\mu M P}_{=:B_4}.$$

Hence, we have that

$$Z \left( Z^T Z \right)^{-1/2}$$
$$= (\mathrm{Id} + B) V_{U_t W_t} \left( V_{U W_t}^T \left( \mathrm{Id} + B + B^T + B^T B \right) V_{U_t W_t} \right)^{-1/2}$$
$$= (\mathrm{Id} + B) V_{U_t W_t} \left( \mathrm{Id} + V_{U_t W_t}^T B V_{U_t W_t} + V_{U_t W_t}^T B^T V_{U_t W_t} + V_{U_t W_t}^T B^T B V_{U_t W_t} \right)^{-1/2}.$$

It follows from Lemma B.2 that

$$\left( \mathrm{Id} + V_{U_t W_t}^T B V_{U_t W_t} + V_{U_t W_t}^T B^T V_{U_t W_t} + V_{U_t W_t}^T B^T B V_{U_t W_t} \right)^{-1/2}$$
$$= \mathrm{Id} - \frac{1}{2} \left( V_{U_t W_t}^T B V_{U_t W_t} + V_{U_t W_t}^T B^T V_{U_t W_t} + V_{U_t W_t}^T B^T B V_{U_t W_t} \right) + C,$$

where $C$ is matrix, which satisfies

$$\| C \| \leq 3 \| V_{U_t W_t}^T B V_{U_t W_t} + V_{U_t W_t}^T B^T V_{U_t W_t} + V_{U_t W_t}^T B^T B V_{U_t W_t} \|^2. \tag{98}$$

It follows that

$$Z \left( Z^T Z \right)^{-1/2}$$
$$= (\mathrm{Id} + B) V_{U_t W_t} \left( \mathrm{Id} - \frac{1}{2} \left( V_{U_t W_t}^T B V_{U_t W_t} + V_{U_t W_t}^T B^T V_{U_t W_t} + V_{U_t W_t}^T B^T B V_{U_t W_t} \right) + C \right)$$
$$= V_{U_t W_t} + B V_{U_t W_t} - \frac{1}{2} (\mathrm{Id} + B) V_{U_t W_t} V_{U_t W_t}^T \left( B + B^T \right) V_{U_t W_t} - D,$$

where we have set

$$D = (\mathrm{Id} + B) V_{U_t W_t} \left( \frac{1}{2} V_{U_t W_t}^T B^T B V_{U_t W_t} - C \right). \tag{99}$$

Hence,

$$V_{X^\perp}^T Z \left(Z^T Z\right)^{-1/2}$$

$$=V_{X^\perp}^T \left(\mathrm{Id} + B - \frac{1}{2}V_{U_t W_t} V_{U_t W_t}^T \left(B + B^T\right)\right) V_{U_t W_t} - \frac{1}{2}V_{X^\perp}^T B V_{U_t W_t} V_{U_t W_t}^T \left(B + B^T\right) V_{U_t W_t} - V_{X^\perp}^T D$$

$$=\underbrace{V_{X^\perp}^T \left(\mathrm{Id} + B_1 - \frac{1}{2}V_{U_t W_t} V_{U_t W_t}^T \left(B_1 + B_1^T\right)\right) V_{U_t W_t}}_{=:(I)}$$

$$+\underbrace{V_{X^\perp}^T \left(B_2 - \frac{1}{2}V_{U_t W_t} V_{U_t W_t}^T \left(B_2 + B_2^T\right)\right) V_{U_t W_t}}_{=:(II)}$$

$$+\underbrace{V_{X^\perp}^T \left(B_3 - \frac{1}{2}V_{U_t W_t} V_{U_t W_t}^T \left(B_3 + B_3^T\right)\right) V_{U_t W_t}}_{=:(III)}$$

$$+\underbrace{V_{X^\perp}^T \left(B_4 - \frac{1}{2}V_{U_t W_t} V_{U_t W_t}^T \left(B_4 + B_4^T\right)\right) V_{U_t W_t}}_{=:(IV)} \underbrace{- \frac{1}{2}V_{X^\perp}^T B V_{U_t W_t} V_{U_t W_t}^T \left(B + B^T\right) V_{U_t W_t}}_{=:(V)} \underbrace{- V_{X^\perp}^T D}_{=:(VI)}.$$

**Estimating $(I)$:** We observe that

$$V_{X^\perp}^T \left(\mathrm{Id} + B_1 - \frac{1}{2}V_{U_t W_t} V_{U_t W_t}^T \left(B_1 + B_1^T\right)\right) V_{U_t W_t}$$

$$=V_{X^\perp}^T \left(\mathrm{Id} + \mu \left(\mathrm{Id} - V_{U_t W_t} V_{U_t W_t}^T\right)\left(X X^T - U_t U_t^T\right)\right) V_{U_t W_t}$$

$$=V_{X^\perp}^T V_{U_t W_t} + \mu V_{X^\perp}^T \left(\mathrm{Id} - V_{U_t W_t} V_{U_t W_t}^T\right)\left(X X^T - U_t U_t^T\right) V_{U_t W_t}$$

$$=V_{X^\perp}^T V_{U_t W_t} + \mu V_{X^\perp}^T \left(\mathrm{Id} - V_{U_t W_t} V_{U_t W_t}^T\right) X X^T V_{U_t W_t} - \mu V_{X^\perp}^T \left(\mathrm{Id} - V_{U_t W_t} V_{U_t W_t}^T\right) U U_t^T V_{U_t W_t}$$

$$=V_{X^\perp}^T V_{U_t W_t} - \mu V_{X^\perp}^T V_{U_t W_t} V_{U_t W_t}^T X X^T V_{U_t W_t} - \mu V_{X^\perp}^T \left(\mathrm{Id} - V_{U_t W_t} V_{U_t W_t}^T\right) U_t U_t^T V_{U_t W_t}$$

$$=V_{X^\perp}^T V_{U_t W_t} - \mu V_{X^\perp}^T V_{U_t W_t} V_{U_t W_t}^T X X^T V_{U_t W_t} - \mu V_{X^\perp}^T \left(\mathrm{Id} - V_{U_t W_t} V_{U_t W_t}^T\right) U_t W_{t,\perp} W_{t,\perp}^T U_t^T V_{U_t W_t}$$

$$=V_{X^\perp}^T V_{U_t W_t} \left(\mathrm{Id} - \mu V_{U_t W_t}^T X X^T V_{U_t W_t}\right) - \mu V_{X^\perp}^T \left(\mathrm{Id} - V_{U_t W_t} V_{U_t W_t}^T\right) U_t W_{t,\perp} W_{t,\perp}^T U_t^T V_{U_t W_t}$$

$$=V_{X^\perp}^T V_{U_t W_t} \left(\mathrm{Id} - \mu V_{U_t W_t}^T X X^T V_{U_t W_t}\right) - \mu V_{X^\perp}^T \left(\mathrm{Id} - V_{U_t W_t} V_{U_t W_t}^T\right) U_t W_{t,\perp} W_{t,\perp}^T U_t^T V_{X^\perp} V_{X^\perp}^T V_{U_t W_t}.$$

It follows that

$$\|(I)\| \le \|V_{X^\perp}^T V_{U_t W_t}\| \left(1 - \mu \sigma_{\min}\left(V_{U_t W_t}^T X X^T V_{U_t W_t}\right)\right) + \mu\|V_{X^\perp}^T V_{U_t W_t}\|\|U_t W_{t,\perp}\|^2$$

$$\le \|V_{X^\perp}^T V_{U_t W_t}\| \left(1 - \frac{\mu}{2}\sigma_{\min}\left(X\right)^2\right) + \mu\|V_{X^\perp}^T V_{U_t W_t}\|\|U_t W_{t,\perp}\|^2$$

$$= \|V_{X^\perp}^T V_{U_t W_t}\| \left(1 - \frac{\mu}{2}\sigma_{\min}\left(X\right)^2 + \mu\|U_t W_{t,\perp}\|^2\right)$$

$$\le \|V_{X^\perp}^T V_{U_t W_t}\| \left(1 - \frac{\mu}{3}\sigma_{\min}\left(X\right)^2\right).$$

**Bounding $(II)$:** We observe that

$$V_{X^\perp}^T \left(B_2 - \frac{1}{2}V_{U_t W_t} V_{U_t W_t}^T \left(B_2 + B_2^T\right)\right) V_{U_t W_t}$$

$$= \mu V_{X^\perp}^T \left(\left(\mathrm{Id} - V_{U_t W_t} V_{U_t W_t}^T\right)\left(\mathcal{A}^* \mathcal{A} - \mathrm{Id}\right)\left(X X^T - U_t U_t^T\right)\right) V_{U_t W_t}.$$

It follows that

$$\left\|V_{X^\perp}^T \left(B_2 - \frac{1}{2}V_{U_t W_t} V_{U_t W_t}^T \left(B_2 + B_2^T\right)\right) V_{U_t W_t}\right\| \le \mu\|\left(\mathrm{Id} - \mathcal{A}^* \mathcal{A}\right)\left(X X^T - U_t U_t^T\right)\|.$$

**Estimating $(III)$:** First, we recall that

$$B_3 = U_t W_{t,\perp} W_{t,\perp}^T W_{t+1} \left(V_{U_t W_t}^T U_t W_t W_t^T W_{t+1}\right)^{-1} V_{U_t W_t}^T$$

$$= U_t W_{t,\perp} W_{t,\perp}^T W_{t+1} \left(W_t^T W_{t+1}\right)^{-1} \left(V_{U_t W_t}^T U_t W_t\right)^{-1} V_{U_t W_t}^T.$$

Before we proceed further, we need to understand $W_{t,\perp}^T W_{t+1}$ and $W_t^T W_{t+1}$. By Lemma B.3 it holds that

$$\|W_{t,\perp}^T W_{t+1}\| \leq \mu\left(\frac{1}{800}\sigma_{\min}^2(X) + \|U_t W_t\|\|U_t W_{t,\perp}\|\right)\|V_{X^\perp}^T V_{U_t W_t}\| + 4\mu\|\left(\mathrm{Id} - \mathcal{A}^*\mathcal{A}\right)\left(XX^T - U_t U_t^T\right)\|$$

and

$$\sigma_{\min}\left(W_t^T W_{t+1}\right) \geq 1/2.$$

It follows that

$$
\begin{aligned}
\|B_3\| &\leq \|W_{t,\perp}^T W_{t+1}\|\|U_t W_{t,\perp}\|\|\left(W_t^T W_{t+1}\right)^{-1}\|\|\left(V_{U_t W_t}^T U_t W_t\right)^{-1}\| \\
&= \frac{\|W_{t,\perp}^T W_{t+1}\|\|U_t W_{t,\perp}\|}{\sigma_{\min}\left(W_t^T W_{t+1}\right)\sigma_{\min}\left(V_{U_t W_t}^T U_t W_t\right)} \\
&= \frac{\|W_{t,\perp}^T W_{t+1}\|\|U_t W_{t,\perp}\|}{\sigma_{\min}\left(W_t^T W_{t+1}\right)\sigma_{\min}\left(U_t W_t\right)} \\
&\leq 4\|W_{t,\perp}^T W_{t+1}\|.
\end{aligned}
$$

Hence, we can conclude that

$$
\begin{aligned}
&\left\|V_{X^\perp}^T\left(B_3 - \frac{1}{2}V_{U_t W_t}V_{U_t W_t}^T\left(B_3 + B_3^T\right)\right)V_{U_t W_t}\right\| \\
&\leq 2\|B_3\| \\
&\leq 8\|W_{t,\perp}^T W_{t+1}\| \\
&\overset{(a)}{\leq} \mu\left(\frac{1}{800}\sigma_{\min}^2(X) + 8\|U_t W_t\|\|U_t W_{t,\perp}\|\right)\|V_{X^\perp}^T V_{U_t W_t}\| + 32\mu\|\left(\mathrm{Id} - \mathcal{A}^*\mathcal{A}\right)\left(XX^T - U_t U_t^T\right)\| \\
&\overset{(b)}{\leq} \frac{1}{400}\mu \cdot \sigma_{\min}^2(X)\|V_{X^\perp}^T V_{U_t W_t}\| + 32\mu\|\left(\mathrm{Id} - \mathcal{A}^*\mathcal{A}\right)\left(XX^T - U_t U_t^T\right)\|.
\end{aligned}
$$

Inequality $(a)$ follows from Lemma B.3. In $(b)$ we used the assumption $\|U_t W_{t,\perp}\| \leq c\kappa^{-2}\sigma_{\min}(X)$ and $\|U_t W_t\| \leq 3\|X\|$.

**Bounding $(IV)$:** We start by noticing that

$$
\begin{aligned}
\mu\|\mathcal{A}^*\mathcal{A}\left(XX^T - U_t U_t^T\right)\| &\leq \mu\left(\|XX^T - U_t U_t^T\| + \|\left(\mathrm{Id} - \mathcal{A}^*\mathcal{A}\right)\left(XX^T - U_t U_t^T\right)\|\right) \\
&\leq 11\mu\|X\|^2 \\
&\leq 11c\kappa^{-2},
\end{aligned}
$$

where we have used the assumption $\|U\| \leq 3\|X\|$, (44) and (46). Hence, we obtain that

$$
\begin{aligned}
&\left\|V_{X^\perp}^T\left(B_4 - \frac{1}{2}V_{U_t W_t}V_{U_t W_t}^T\left(B_4 + B_4^T\right)\right)V_{U_t W_t}\right\| \\
&\leq 2\|B_4\| \\
&= 2\mu\|MP\| \\
&\leq 2\mu\|\mathcal{A}^*\mathcal{A}\left(XX^T - U_t U_t^T\right)\|\|B_3\| \\
&\leq 22c\kappa^{-2}\|B_3\| \\
&\overset{(a)}{\leq} \frac{\mu}{50} \cdot \kappa^{-2}\sigma_{\min}^2(X)\|V_{X^\perp}^T V_{U_t W_t}\| + 352c\mu\|\left[\left(\mathrm{Id} - \mathcal{A}^*\mathcal{A}\right)\left(XX^T - U_t U_t^T\right)\right]\|.
\end{aligned}
$$

Inequality $(a)$ follows from similar arguments as when we were bounding $(III)$ and by choosing the constant $c > 0$ small enough.

**Bounding** $(V)$: First, we want to estimate $\|B\|$. We note that

$$
\begin{aligned}
\|B\| &\overset{(a)}{\leq} \mu\|M\| + \|P\| + \mu\|MP\| \\
&\overset{(b)}{\leq} \mu\|M\| + \|B_3\| + \mu\|M\|\|B_3\| \\
&\overset{(c)}{=} \mu\|\mathcal{A}^*\mathcal{A}\left(XX^T - U_tU_t^T\right)\| + \left(1 + \mu\|\mathcal{A}^*\mathcal{A}\left(XX^T - U_tU_t^T\right)\|\right)\|B_3\| \\
&\overset{(d)}{\leq} \mu\|\mathcal{A}^*\mathcal{A}\left(XX^T - U_tU_t^T\right)\| + 2\|B_3\| \\
&\overset{(e)}{\leq} \mu\|\mathcal{A}^*\mathcal{A}\left(XX^T - U_tU_t^T\right)\| + \frac{1}{400}\mu \cdot \sigma_{\min}^2(X)\|V_{X^\perp}^T V_{U_t W_t}\| + 32\mu\|\left[(\mathrm{Id} - \mathcal{A}^*\mathcal{A})\left(XX^T - U_tU_t^T\right)\right]\| \\
&\leq \mu\|XX^T - U_tU_t^T\| + \frac{1}{400}\mu \cdot \sigma_{\min}^2(X)\|V_{X^\perp}^T V_{U_t W_t}\| + 33\mu\|\left(\mathrm{Id} - \mathcal{A}^*\mathcal{A}\right)\left(XX^T - U_tU_t^T\right)\|.
\end{aligned}
$$

In $(a)$ we used the triangle inequality and in $(b)$ we used $B_3 = P$ and the submultiplicativity of the spectral norm. To obtain equality $(c)$ we inserted the definition of $M$. For $(d)$ we used that $\mu\|\mathcal{A}^*\mathcal{A}\left(XX^T - U_tU_t^T\right)\| \leq 2$, which follows from assumption (44) and (46). For inequality $(e)$ we used our bound for $\|B_3\|$, which we have derived when bounding $(III)$. Hence, we have shown that

$$
\|B\| \leq \mu\|XX^T - U_tU_t^T\| + \frac{1}{400}\mu \cdot \sigma_{\min}^2(X)\|V_{X^\perp}^T V_{U_t W_t}\| + 33\mu\|\left(\mathrm{Id} - \mathcal{A}^*\mathcal{A}\right)\left(XX^T - U_tU_t^T\right)\|. \tag{100}
$$

We obtain that

$$
\begin{aligned}
&\frac{1}{2}\|V_{X^\perp}^T B V_{U_t W_t} V_{U_t W_t}^T\left(B + B^T\right)V_{U_t W_t}\| \\
&\leq \frac{1}{2}\|B\|\|B + B^T\| \\
&\leq \|B\|^2 \\
&\overset{(a)}{\leq} 3\mu^2\left(\|XX^T - U_tU_t^T\|^2 + \frac{1}{400^2}\sigma_{\min}^4(X)\|V_{X^\perp}^T V_{U_t W_t}\|^2 + 33^2\|\left(\mathrm{Id} - \mathcal{A}^*\mathcal{A}\right)\left(XX^T - U_tU_t^T\right)\|^2\right) \\
&\overset{(b)}{\leq} 3\mu^2\|XX^T - U_tU_t^T\|^2 + \frac{3}{2}c\frac{\mu\kappa^{-4}}{400^2}\sigma_{\min}^2(X)\|V_{X^\perp}^T V_{U_t W_t}\|^2 + \frac{3}{2}\mu^2\sigma_{\min}^2(X)\|\left(\mathrm{Id} - \mathcal{A}^*\mathcal{A}\right)\left(XX^T - U_tU_t^T\right)\| \\
&\overset{(c)}{\leq} 3\mu^2\|XX^T - U_tU_t^T\|^2 + 3c\frac{\mu\kappa^{-4}}{2\cdot 400^2}\sigma_{\min}^2(X)\|V_{X^\perp}^T V_{U_t W_t}\|^2 + \frac{3}{2}c\mu\kappa^{-4}\|\left(\mathrm{Id} - \mathcal{A}^*\mathcal{A}\right)\left(XX^T - U_tU_t^T\right)\|,
\end{aligned} \tag{101}
$$

where in $(a)$ we used inequality (100) combined with Jensen's inequality. For inequality $(b)$ we used (44) and (46). Inequality $(c)$ follows again from (46).

**Bounding** $(VI)$: We are first going to show that $\|B\| \leq 1$. Indeed, we have that

$$
\begin{aligned}
\|B\| &\overset{(a)}{\leq} 10\mu\|X\|^2 + \frac{1}{400}\mu \cdot \sigma_{\min}^2(X)\|V_{X^\perp}^T V_{U_t W_t}\| + 33\mu\|\left(\mathrm{Id} - \mathcal{A}^*\mathcal{A}\right)\left(XX^T - U_tU_t^T\right)\| \\
&\overset{(b)}{\leq} 10\mu\|X\|^2 + \frac{1}{400}\mu \cdot \sigma_{\min}^2(X) + 33c\mu\sigma_{\min}^2(X) \\
&\overset{(c)}{\leq} 1,
\end{aligned}
$$

where in $(a)$ we used inequality (100) and the assumption $\|U\| \leq 3\|X\|$. For inequality $(b)$ we used (44) and for inequality $(c)$ we used assumption (46). We note that from inequality (99) it follows that

$$
\|D\| \leq (1 + \|B\|)\left(\frac{1}{2}\|B\|^2 + \|C\|\right) \leq 2\left(\|B\|^2 + \|C\|\right). \tag{102}
$$

In the first inequality we used the triangle inequality and in the second inequality we used $\|B\| \leq 1$. Note that from (98) and again $\|B\| \leq 1$ it follows that

$$
\|C\| \leq 3\left(2\|B\| + \|B\|^2\right)^2 \leq 27\|B\|^2. \tag{103}
$$

Hence, we obtain that

$$\|D\|$$

$$\overset{(a)}{\leq} 56\|B\|^2$$

$$\overset{(b)}{\leq} 56\left(3\mu^2\|XX^T - U_tU_t^T\|^2 + 3c\frac{\mu\kappa^{-4}}{2\cdot 400^2}\sigma_{\min}^2(X)\|V_{X^\perp}^T V_{U_tW_t}\|^2 + \frac{3}{2}c\mu\kappa^{-4}\|\left(\mathrm{Id} - \mathcal{A}^*\mathcal{A}\right)\left(XX^T - U_tU_t^T\right)\|\right).$$

Inequality $(a)$ is due to the inequalities (102) and (103) and inequality $(b)$ follows from (101).

**Combining the estimates:** By combining our results we obtain that for small enough $c > 0$ we have that

$$\|V_{X^\perp}^T V_{U_{t+1}W_{t+1}}\|$$

$$\leq \|(I)\| + \|(II)\| + \|(III)\| + \|(IV)\| + \|(V)\| + \|(VI)\|$$

$$\leq \left(1 - \frac{\mu}{4}\sigma_{\min}^2(X)\right)\|V_{X^\perp}^T V_{U_tW_t}\| + 100\mu\|\left(\mathrm{Id} - \mathcal{A}^*\mathcal{A}\right)\left(XX^T - U_tU_t^T\right)\| + 500\mu^2\|XX^T - U_tU_t^T\|^2.$$

This finishes the proof. $\qquad\square$

## B.4  Proof of Lemma 9.4

*Proof of Lemma 9.4.* We observe that

$$U_{t+1} = U_t + \mu\left(XX^T - U_tU_t^T\right)U_t + \mu\left[\left(\mathcal{A}^*\mathcal{A} - \mathrm{Id}\right)\left(XX^T - U_tU_t^T\right)\right]U_t$$

$$= \left(\mathrm{Id} - \mu U_tU_t^T\right)U_t + \mu XX^T U_t + \mu\left[\left(\mathcal{A}^*\mathcal{A} - \mathrm{Id}\right)\left(XX^T - U_tU_t^T\right)\right]U_t.$$

Note that $\|\left(\mathrm{Id} - \mu U_tU_t^T\right)U_t\| = \left(1 - \mu\|U_t\|^2\right)\|U_t\|$ due to $\mu \leq \frac{1}{27}\|X\|^{-2} \leq \frac{1}{3}\|U_t\|^2$. Hence, by the triangle inequality and submultiplicativity of the spectral norm we obtain that

$$\|U_{t+1}\| \leq \left(1 - \mu\|U_t\|^2 + \mu\|X\|^2 + \mu\|\left(\mathcal{A}^*\mathcal{A} - \mathrm{Id}\right)\left(XX^T - U_tU_t^T\right)\|\right)\|U_t\|.$$

Hence, by our assumption on $\|\left(\mathrm{Id} - \mathcal{A}^*\mathcal{A}\right)\left(XX^T - U_tU_t^T\right)\|$ we obtain that

$$\|U_{t+1}\| \leq \left(1 - \mu\|U_t\|^2 + 2\mu\|X\|^2\right)\|U_t\|. \tag{104}$$

Now assume that $2\|X\| \leq \|U_t\| \leq 3\|X\|$. Then it follows from the last inequality that $\|U_{t+1}\| \leq \|U_t\|$, which due to the assumption $\|U_t\| \leq 3\|X\|$ implies the claim $\|U_{t+1}\| \leq 3\|X\|$. However, if $\|U_t\| \leq 2\|X\|$ holds, then by combining inequality (104) with the assumption $\mu \leq \frac{\|X\|^{-2}}{27}$ we obtain that $\|U_{t+1}\| \leq 3\|X\|$ as well, which finishes the proof. $\qquad\square$

## B.5  Proof of Lemma 9.5

**Lemma B.4.** *Under the assumptions of Lemma 9.5 it holds that*

$$\left\|V_{X^\perp}^T U_tU_t^T\right\| \leq 3\left\|V_X^T\left(XX^T - U_tU_t^T\right)\right\| + \left\|U_tW_{t,\perp}W_{t,\perp}^T U_t^T\right\|$$

*as well as*

$$\left\|XX^T - U_tU_t^T\right\| \leq 4\left\|V_X^T\left(XX^T - U_tU_t^T\right)\right\| + \left\|U_tW_{t,\perp}W_{t,\perp}^T U_t^T\right\|.$$

*Proof.* We notice that by the triangle inequality and submultiplicativity it holds that

$$\left\|V_{X^\perp}^T U_tU_t^T\right\| \leq \left\|V_{X^\perp}^T U_tU_t^T V_X\right\| + \left\|V_{X^\perp}^T U_tU_t^T V_{X^\perp}\right\|$$

$$= \left\|V_{X^\perp}^T\left(XX^T - U_tU_t^T\right)V_X\right\| + \left\|V_{X^\perp}^T U_tU_t^T V_{X^\perp}\right\|$$

$$= \left\|V_X^T\left(XX^T - U_tU_t^T\right)\right\| + \left\|V_{X^\perp}^T U_tU_t^T V_{X^\perp}\right\|.$$

In order to bound the second term we compute that

$$\left\|V_{X^\perp}^T U_tU_t^T V_{X^\perp}\right\| \leq \left\|V_{X^\perp}^T U_tW_tW_t^T U_t^T V_{X^\perp}\right\| + \left\|V_{X^\perp}^T U_tW_{t,\perp}W_{t,\perp}^T U_t^T V_{X^\perp}\right\|$$

$$= \left\|V_{X^\perp}^T U_tW_tW_t^T U_t^T V_{X^\perp}\right\| + \left\|U_tW_{t,\perp}W_{t,\perp}^T U_t^T\right\|.$$

In order to bound the first term we note that

$$
\begin{aligned}
\left\|V_{X^\perp}^T U_t W_t W_t^T U_t^T V_{X^\perp}\right\| &= \left\|V_{X^\perp}^T V_{U_t W_t} V_{U_t W_t}^T U_t W_t W_t^T U_t^T V_{X^\perp}\right\| \\
&= \left\|V_{X^\perp}^T V_{U_t W_t} \left(V_X^T V_{U_t W_t}\right)^{-1} V_X^T V_{U_t W_t} V_{U_t W_t}^T U_t W_t W_t^T U_t^T V_{X^\perp}\right\| \\
&\le \left\|V_{X^\perp}^T V_{U_t W_t}\right\| \left\|\left(V_X^T V_{U_t W_t}\right)^{-1}\right\| \left\|V_X^T V_{U_t W_t} V_{U_t W_t}^T U_t W_t W_t^T U_t^T\right\| \\
&= \frac{\left\|V_{X^\perp}^T V_{U_t W_t}\right\|}{\sigma_{\min}\left(V_X^T V_{U_t W_t}\right)} \left\|V_X^T U_t U_t^T V_{X^\perp}\right\| \\
&= \frac{\left\|V_{X^\perp}^T V_{U_t W_t}\right\|}{\sigma_{\min}\left(V_X^T V_{U_t W_t}\right)} \left\|V_X^T \left(XX^T - U_t U_t^T\right) V_{X^\perp}\right\| \\
&\le \frac{\left\|V_{X^\perp}^T V_{U_t W_t}\right\|}{\sigma_{\min}\left(V_X^T V_{U_t W_t}\right)} \left\|V_X^T \left(XX^T - U_t U_t^T\right)\right\| \\
&\le 2\left\|V_X^T \left(XX^T - U_t U_t^T\right)\right\|.
\end{aligned}
$$

Hence we can conclude that

$$
\left\|V_{X^\perp}^T U_t U_t^T\right\| \le 3\left\|V_X^T \left(XX^T - U_t U_t^T\right)\right\| + \left\|U_t W_{t,\perp} W_{t,\perp}^T U_t^T\right\|,
$$

which shows the first inequality in the lemma. In order to prove the second inequality, we note that by the triangle inequality and submultiplicativity it holds that

$$
\begin{aligned}
\left\|XX^T - U_t U_t^T\right\| &\le \left\|V_X^T \left(XX^T - U_t U_t^T\right)\right\| + \left\|V_{X^\perp}^T U_t U_t^T\right\| \\
&\le 4\left\|V_X^T \left(XX^T - U_t U_t^T\right)\right\| + \left\|U_t W_{t,\perp} W_{t,\perp}^T U_t^T\right\|,
\end{aligned}
$$

where in the last line we used the previous inequality. This finishes the proof. $\qquad\square$

After having provided the necessary ingredients, we are in a position to prove Lemma 9.5.

*Proof of Lemma 9.5.* Recall that

$$
U_{t+1} = U_t + \mu\left[(\mathcal{A}^*\mathcal{A})\left(XX^T - U_t U_t^T\right)\right] U_t.
$$

Next, we compute that

$$
\begin{aligned}
XX^T - U_{t+1} U_{t+1}^T =\ & XX^T - U_t U_t^T - \mu\left[(\mathcal{A}^*\mathcal{A})\left(XX^T - U_t U_t^T\right)\right] U_t U_t^T - \mu U_t U_t^T\left[(\mathcal{A}^*\mathcal{A})\left(XX^T - U_t U_t^T\right)\right] \\
& - \mu^2\left[(\mathcal{A}^*\mathcal{A})\left(XX^T - U_t U_t^T\right)\right] U_t U_t^T\left[(\mathcal{A}^*\mathcal{A})\left(XX^T - U_t U_t^T\right)\right] \\
=\ & XX^T - U_t U_t^T - \mu\left(XX^T - U_t U_t^T\right) U_t U_t^T - \mu U_t U_t^T\left(XX^T - U_t U_t\right) \\
& + \mu\left[(\mathrm{Id} - \mathcal{A}^*\mathcal{A})\left(XX^T - U_t U_t^T\right)\right] U_t U_t^T + \mu U_t U_t^T\left[(\mathrm{Id} - \mathcal{A}^*\mathcal{A})\left(XX^T - U_t U_t^T\right)\right] \\
& - \mu^2\left[(\mathcal{A}^*\mathcal{A})\left(XX^T - UU_t^T\right)\right] U_t U_t^T\left[(\mathcal{A}^*\mathcal{A})\left(XX^T - U_t U_t^T\right)\right] \\
=\ & \underbrace{\left(\mathrm{Id} - \mu U_t U_t^T\right)\left(XX^T - U_t U_t^T\right)\left(\mathrm{Id} - \mu U_t U_t^T\right)}_{=(I)} + \underbrace{\mu\left[(\mathrm{Id} - \mathcal{A}^*\mathcal{A})\left(XX^T - U_t U_t\right)\right] U_t U_t^T}_{=(II)} \\
& + \underbrace{\mu U_t U_t^T\left[(\mathrm{Id} - \mathcal{A}^*\mathcal{A})\left(XX^T - U_t U_t^T\right)\right]}_{=(III)} - \underbrace{\mu^2 U_t U_t\left(XX^T - U_t U_t^T\right) U_t U_t^T}_{=(IV)} \\
& - \underbrace{\mu^2\left[(\mathcal{A}^*\mathcal{A})\left(XX^T - U_t U_t^T\right)\right] U_t U_t^T\left[(\mathcal{A}^*\mathcal{A})\left(XX^T - U_t U_t^T\right)\right]}_{=(V)}.
\end{aligned}
$$

We are going to deal with each summand individually.

**Estimation of** $(I)$**:** We note that

$$V_X^T \left( \mathrm{Id} - \mu U_t U_t^T \right) \left( X X^T - U_t U_t^T \right) \left( \mathrm{Id} - \mu U_t U_t \right)$$
$$= V_X^T \left( \mathrm{Id} - \mu U_t U_t^T \right) V_X V_X^T \left( X X^T - U_t U_t^T \right) \left( \mathrm{Id} - \mu U_t U_t^T \right)$$
$$\quad + V_X^T \left( \mathrm{Id} - \mu U_t U_t^T \right) V_{X^\perp} V_{X^\perp}^T \left( X X^T - U_t U_t^T \right) \left( \mathrm{Id} - \mu U_t U_t^T \right)$$
$$= V_X^T \left( \mathrm{Id} - \mu U_t U_t^T \right) V_X V_X^T \left( X X^T - U_t U_t \right) \left( \mathrm{Id} - \mu U_t U_t^T \right) + \mu V_X^T U_t U_t^T V_{X^\perp} V_{X^\perp}^T U_t U_t^T \left( \mathrm{Id} - \mu U_t U_t \right)$$
$$= \left( \mathrm{Id} - \mu V_X^T U_t U_t V_X \right) V_X^T \left( X X^T - U_t U_t^T \right) \left( \mathrm{Id} - \mu U_t U_t^T \right) + \mu V_X^T U_t U_t^T V_{X^\perp} V_{X^\perp}^T U_t U_t^T \left( \mathrm{Id} - \mu U_t U_t^T \right).$$

Hence, we obtain that

$$\left\| \left( \mathrm{Id} - \mu V_X^T U_t U_t^T V_X \right) V_X^T \left( X X^T - U_t U_t^T \right) \left( \mathrm{Id} - \mu U_t U_t^T \right) \right\|$$
$$\leq \left\| \left( \mathrm{Id} - \mu V_X^T U_t U_t^T V_X \right) \right\| \left\| V_X^T \left( X X^T - U_t U_t^T \right) \right\| \left\| \left( \mathrm{Id} - \mu U_t U_t^T \right) \right\|$$
$$\leq \left\| \left( \mathrm{Id} - \mu V_X^T U_t U_t^T V_X \right) \right\| \left\| V_X^T \left( X X^T - U_t U_t^T \right) \right\|$$
$$= \left( 1 - \mu \sigma_{\min} \left( V_X^T U_t U_t^T V_X \right) \right) \left\| V_X^T \left( X X^T - U_t U_t^T \right) \right\|$$
$$\leq \left( 1 - \mu \sigma_{\min}^2 \left( V_X^T U_t W_t \right) \right) \left\| V_X^T \left( X X^T - U_t U_t^T \right) \right\|.$$

Next, we note that

$$\sigma_{\min}^2 \left( V_X^T U_t W_t \right) = \sigma_{\min}^2 \left( V_X^T V_{U_t W_t} V_{U_t W_t}^T U_t W_t \right)$$
$$\geq \sigma_{\min}^2 \left( V_X^T V_{U_t W_t} \right) \sigma_{\min}^2 \left( U_t W_t \right)$$
$$\geq \frac{1}{2} \sigma_{\min}^2 \left( U_t W_t \right)$$
$$\geq \frac{1}{20} \sigma_{\min}^2 \left( X \right),$$

where in the last line we used the assumption $\sigma_{\min}^2 \left( U_t W_t \right) \geq \frac{1}{10} \sigma_{\min}^2 \left( X \right)$. Hence, we have shown that

$$\left\| \left( \mathrm{Id} - \mu V_X^T U_t U_t^T V_X \right) V_X^T \left( X X^T - U_t U_t^T \right) \left( \mathrm{Id} - \mu U_t U_t^T \right) \right\| \leq \left( 1 - \frac{\mu}{20} \sigma_{\min}^2 \left( X \right) \right) \left\| V_X^T \left( X X^T - U_t U_t^T \right) \right\|.$$

Next, we note that

$$\left\| V_X^T U_t U_t^T V_{X^\perp} V_{X^\perp}^T U_t U_t^T \left( \mathrm{Id} - \mu U_t U_t \right) \right\|$$
$$\leq \left\| V_X^T U_t U_t^T V_{X^\perp} V_{X^\perp}^T U_t U_t \right\| \left\| \mathrm{Id} - \mu U_t U_t^T \right\|$$
$$\leq \left\| V_X^T U_t U_t^T V_{X^\perp} V_{X^\perp}^T U_t U_t^T \right\|$$
$$\leq \left\| V_X^T U_t W_t W_t^T U_t V_{X^\perp} V_{X^\perp}^T U_t U_t^T \right\|$$
$$\leq \left\| V_X^T U_t W_t \right\| \left\| V_{X^\perp}^T U_t W_t \right\| \left\| V_{X^\perp}^T U_t U_t^T \right\|$$
$$\leq \left\| U_t W_t \right\|^2 \left\| V_{X^\perp}^T V_{U_t W_t} \right\| \left\| V_{X^\perp}^T U_t U_t^T \right\|$$
$$\leq 9 \| X \|^2 \left\| V_{X^\perp}^T V_{U_t W_t} \right\| \left\| V_{X^\perp}^T U_t U_t^T \right\|$$
$$\leq 9 \| X \|^2 \left\| V_{X^\perp}^T V_{U_t W_t} \right\| \left( 3 \left\| V_X^T \left( X X^T - U_t U_t^T \right) \right\| + \left\| U_t W_{t,\perp} W_{t,\perp}^T U_t \right\| \right),$$

where in the last line we used Lemma B.4. Then, using the assumption $\left\| V_{X^\perp}^T V_{U_t W_t} \right\| \leq c \kappa^{-2}$ it follows that

$$\left\| V_X^T U_t U_t^T V_{X^\perp} V_{X^\perp}^T U U_t^T \left( \mathrm{Id} - U_t U_t^T \right) \right\|$$
$$\leq \frac{1}{100} \sigma_{\min}^2 \left( X \right) \left\| V_X^T \left( X X^T - U_t U_t^T \right) \right\| + \frac{\sigma_{\min}^2 \left( X \right)}{400} \left\| U_t W_{t,\perp} W_{t,\perp}^T U_t \right\|.$$

Hence, we have shown that

$$\left\| V_X^T \left( \mathrm{Id} - \mu U_t U_t^T \right) \left( X X^T - U_t U_t^T \right) \left( \mathrm{Id} - \mu U U_t^T \right) \right\|$$
$$\leq \left( 1 - \frac{\mu}{40} \sigma_{\min}^2 \left( X \right) \right) \left\| V_X^T \left( X X^T - U_t U_t^T \right) \right\| + \mu \frac{\sigma_{\min}^2 \left( X \right)}{400} \left\| U_t W_{t,\perp} W_{t,\perp}^T U_t \right\|.$$

**Estimation of** $(II)$**:** We note that

$$\left\|V_X^T\left[\left(\mathrm{Id}-\mathcal{A}^*\mathcal{A}\right)\left(XX^T-U_tU_t^T\right)\right]U_tU_t^T\right\| \le \left\|\left(\mathrm{Id}-\mathcal{A}^*\mathcal{A}\right)\left(XX^T-U_tU_t^T\right)\right\|\|U_t\|^2$$
$$\lesssim \left\|\left(\mathrm{Id}-\mathcal{A}^*\mathcal{A}\right)\left(XX^T-U_tU_t^T\right)\right\|\|X\|^2$$
$$\lesssim c\sigma_{\min}^2\left(X\right)\left\|XX^T-U_tU_t^T\right\|$$
$$\lesssim c\sigma_{\min}^2\left(X\right)\left(\left\|V_X^T\left(XX^T-U_tU_t^T\right)\right\|+\left\|U_tW_{t,\perp}W_{t,\perp}^TU_t\right\|\right).$$

In the second inequality we used the assumption $\|U\| \le 3\|X\|$ and in the third inequality we used assumption (48). In the fourth inequality we applied Lemma B.4. Hence, by choosing the constant $c > 0$ small enough, we obtain that

$$\left\|V_X^T\left[\left(\mathrm{Id}-\mathcal{A}^*\mathcal{A}\right)\left(XX^T-UU_t^T\right)\right]U_tU_t^T\right\| \le \frac{1}{1000}\sigma_{\min}^2\left(X\right)\left(\left\|V_X^T\left(XX^T-UU_t^T\right)\right\|+\left\|U_tW_{t,\perp}W_{t,\perp}^TU_t^T\right\|\right).$$

**Estimation of** $(III)$**:** In the analogous way as in the estimation of $(II)$ we derive that

$$\left\|V_X^TU_tU_t^T\left[\left(\mathrm{Id}-\mathcal{A}^*\mathcal{A}\right)\left(XX^T-U_tU_t^T\right)\right]\right\| \le \frac{1}{1000}\sigma_{\min}^2\left(X\right)\left(\left\|V_X^T\left(XX^T-U_tU_t^T\right)\right\|+\left\|U_tW_{t,\perp}W_{t,\perp}^TU_t\right\|\right).$$

**Estimation of** $(IV)$**:** We note that it follows from submultiplicativity of the spectral norm that

$$\left\|V_X^TU_tU_t^T\left(XX^T-U_tU_t^T\right)U_tU_t^T\right\| \le \|U\|^4\|XX^T-U_tU_t\|$$
$$\lesssim \|X\|^4\|XX^T-U_tU_t^T\|$$
$$\lesssim \|X\|^4\left\|V_X^T\left(XX^T-U_tU_t^T\right)\right\|+\|X\|^4\|U_tW_{t,\perp}\|^2,$$

where in the second line we used the assumption $\|U_t\| \le 3\|X\|$. In the third line we used Lemma B.4. Then using the assumption $\mu \le c\kappa^{-2}\|X\|^{-2}$ it follows that

$$\mu^2\left\|V_X^TU_tU_t^T\left(XX^T-U_tU_t^T\right)U_tU_t^T\right\| \le \frac{\mu}{200}\sigma_{\min}^2\left(X\right)\left\|V_X^T\left(XX^T-U_tU_t^T\right)\right\|+\mu\frac{\sigma_{\min}^2\left(X\right)}{1000}\left\|U_tW_{t,\perp}W_{t,\perp}^TU_t^T\right\|.$$

**Estimation of** $(V)$**:** We first note that

$$\left\|\left(\mathcal{A}^*\mathcal{A}\right)\left(XX^T-U_tU_t^T\right)\right\| \le \left\|XX^T-U_tU_t^T\right\|+\left\|\left[\left(\mathrm{Id}-\mathcal{A}^*\mathcal{A}\right)\left(XX^T-U_tU_t^T\right)\right]\right\|$$
$$\le \left(1+c\kappa^{-2}\right)\left\|XX^T-U_tU_t\right\|$$
$$\le 2\left\|XX^T-U_tU_t^T\right\|,$$

where we have used Assumption (48). In a similar manner, again using Assumption (48), we can show that

$$\|\left(\mathcal{A}^*\mathcal{A}\right)\left(XX^T-U_tU_t^T\right)\| \le 2\left\|XX^T-U_tU_t^T\right\| \le 2\left(\|X\|^2+\|U_t\|^2\right).$$

Hence, it follows that

$$\left\|V_X^T\left[\left(\mathcal{A}^*\mathcal{A}\right)\left(XX^T-U_tU_t^T\right)\right]U_tU_t^T\left[\left(\mathcal{A}^*\mathcal{A}\right)\left(XX^T-U_tU_t^T\right)\right]\right\|$$
$$\le\left\|\left[\left(\mathcal{A}^*\mathcal{A}\right)\left(XX^T-U_tU_t^T\right)\right]\right\|\|U_t\|^2\|\left(\mathcal{A}^*\mathcal{A}\right)\left(XX^T-U_tU_t^T\right)\|$$
$$\lesssim\left\|XX^T-U_tU_t^T\right\|\|U_t\|^2\|\left(\mathcal{A}^*\mathcal{A}\right)\left(XX^T-U_tU_t^T\right)\|$$
$$\lesssim\left\|XX^T-U_tU_t^T\right\|\|U_t\|^2\left(\|X\|^2+\|U_t\|^2\right)$$
$$\lesssim\left\|XX^T-U_tU_t^T\right\|\|X\|^4$$
$$\lesssim\left(\left\|V_X^T\left(XX^T-U_tU_t^T\right)\right\|+\left\|U_tW_{t,\perp}W_{t,\perp}^TU_t^T\right\|\right)\|X\|^4,$$

where in the third and fourth line we used the estimates from above. The fifth line is due to assumption $\|U_t\| \le 3\|X\|$. In the last line we used Lemma B.4. Hence, it follows that

$$\mu^2\left\|V_X^T\left[\left(\mathcal{A}^*\mathcal{A}\right)\left(XX^T-U_tU_t^T\right)\right]U_tU_t^T\left[\left(\mathcal{A}^*\mathcal{A}\right)\left(XX^T-U_tU_t^T\right)\right]\right\|$$
$$\le\frac{\mu}{1000}\sigma_{\min}^2\left(X\right)\left\|V_X^T\left(XX^T-U_tU_t^T\right)\right\|+\frac{\mu}{400}\sigma_{\min}^2\left(X\right)\left\|U_tW_{t,\perp}W_{t,\perp}^TU_t^T\right\|,$$

where the last inequality is due to due to the assumption $\mu \leq c\kappa^{-2}\|X\|^{-2}$ for a sufficiently small constant $c > 0$.

**Combining the estimates:** By combining the estimates, it follows that

$$\left\|V_X^T\left(XX^T - U_{t+1}U_{t+1}^T\right)\right\| \leq \left(1 - \frac{\mu}{200}\sigma_{\min}^2\left(X\right)\right)\left\|V_X^T\left(XX^T - U_tU_t^T\right)\right\| + \mu\frac{\sigma_{\min}^2\left(X\right)}{100}\left\|U_tW_{t,\perp}W_{t,\perp}^TU_t^T\right\|,$$

which finishes the proof. □