# OpenReview forum: "Small random initialization is akin to spectral learning: Optimization and generalization guarantees for overparameterized low-rank matrix reconstruction"
_NeurIPS.cc/2021/Conference — NeurIPS 2021 Poster_

### Official Review · Reviewer_TBNY · 2021-07-14

**Rating:** 7
**Confidence:** 3

**Summary:**

This paper studies the role of small random initialization followed by gradient descent in the setting of low-rank matrix sensing. Authors show that the trajectory of the gradient descent from small random initialization behaves similarly as spectral initialization in the initial stage, and then moves from degenerate saddle points and converges to the ground truth. Numerical experiments are provided to validate the theory.

**Limitations And Societal Impact:**

Yes

**Main Review:**

The paper is well-written and easy to read. It considers a basic problem, and shows interesting results about the role of random initialization followed by gradient descent. The literature review is comprehensive. Therefore, I'd like to see it accepted.

Below are some minor suggestions/questions:
1. It would be better to provide more detailed informal high-level sketches for the last two stages in Section 4.
2. For the over-parameterized case, it seems that the iterate need to be singular in order for the matrix estimator to be close to the ground truth. Does this affect the convergence rate of vanilla gradient descent? It might be better to add a few discussions.

Typo:
A transpose seems to be missing in equation (1)

**Time Spent Reviewing:**

4

---

> ### Author Response · Authors · 2021-08-11
> **Response to Reviewer TBNY**
>
> Thank you for your detailed review and your positive feedback. We have addressed your concerns and respond to your questions below.
>
> ---
> _"It would be better to provide more detailed informal high-level sketches for the last two stages in Section 4."_
>
> We will add a more detailed proof sketch of Phase II+III in Section 4 in the final version.
>
> ---
> _"For the over-parameterized case, it seems that the iterate need to be singular in order for the matrix estimator to be close to the ground truth. Does this affect the convergence rate of vanilla gradient descent? It might be better to add a few discussions."_
>
> This is a great suggestion. In Figure 5a) one can see that the test and train error decay very quickly in the first few iterations. After that, the matrix $U_t$ is very close to the ground truth $X$, but nearly singular. As Figure 5a) shows the train and test error decay then much slower in this regime. We will add a discussion in the final version.
>
> ---
> _"Typo: A transpose seems to be missing in equation (1)"_
>
> Thanks for spotting this typo. We will fix it.

---

### Official Review · Reviewer_pNVA · 2021-07-15

**Rating:** 7
**Confidence:** 3

**Summary:**

This paper takes an initial step to explain why small random initialization followed by a few iterations gradient descent on non-convex optimization can often converge to a global minimum point with also good generalization ability. Specifically, this paper focuses on low-rank matrix recovery, a highly non-convex optimization problem with spurious local optimal points, and shows that: (1) This method has similar behavior to the popular spectral methods with three phases—a spectral or alignment phase, a saddle avoidance/refinement phase, and a local refinement phase. (2) Larger parameter space tends to have a larger spectral bias which helps convergence to a global optimal point with also smaller generalization error.

**Ethical Concerns:**

I did not see any ethical concerns arising from this theoretical work.

**Ethics Review Area:**

["I don’t know"]

**Limitations And Societal Impact:**

This work points out its limitation that the sample complexity has sub-optimal dependence on $r_*$. Also, I think this work could be generalized into more non-convex optimization problems in the future. This work simply points out that “As with other technologies such insights may potentially also be used nefariously. ”

**Main Review:**

Originality: This work is original as it explains a phenomenon that is not explained well before.

Quality: The submission is technically sound. The claims are well supported by theoretical analysis and experimental results. Appropriate methods are used. This is a complete work. Both the strengths and weaknesses of their work are explained.

Clarity: The submission is generally clear and well organized, but some points need explanation as pointed out in advice and questions below.

Significance: This paper takes an initial and important step to explain why small random initialization followed by a few iterations gradient descent on non-convex optimization can often converge to a global minimum point with also good generalization ability. This work only focuses on the low-rank matrix reconstruction problem and an important future direction is to extend to other non-convex optimization problems, so I think this work will perhaps be cited.

Advice and questions:
(1) What is the power method? You may give more explanation or citations.

(2) Typo in the last paragraph of Section 1: Use “as long as” in “This holds as long the measurement operator obeys a popular restricted isometry property”.

(3) In Section 2, you might add application examples of (1) and explain the meanings of $y_i$, $A_i$, $XX^{\top}$ in the examples. You may also list some applications in the introduction.

(4) What does $\mathcal{A}^*$ mean in the gradient descent equation below eq. (2)? Also, since $X$ is unknown while $y_i$ is known, why not using $y_i$ instead of $X$ in the update rule?

(5) You put the case of $r\ge 2r_*$ to the main text and $r\ge r_*$ including two special cases into the appendix. Why is the former more important?

(6) In Theorem 3.3, can the probability $1-Ce^{-\widetilde{c}r}$ get arbitrarily close to 1? Does it rely on $\alpha$?

(7) How does eq. (6) relate to the equation between eqs. (4) and (5)? For example, does eq. (6) imply that equation? You might well explain that around eq. (6).

(8) In Phase II, the $r_*$-th eigenvalues will align. How does that relate to saddle avoidance? Also, $\alpha\propto \sigma_{r_*}(U_0) \ll \mathcal{O}[\sigma_{\min}(X)]$ yes? If so, I did not see something like $\alpha\le \mathcal{O}[\sigma_{\min}(X)]$ in eq. (4). Could you explain?

(9) What’s the value of r for Figure 2?

(10) You said “We note that while all experimental depictions are based on a single trial, in line with the NeuRIPS guidelines we have drawn these curves multiple times (not depicted) and the behavior of the plots do not change. ” Have you tried multiple initializations of U? Do they have almost overlapping curves?

(11) In your paragraph beginning with “Evolution of the test error and the refinement phase”, eq. (6) might also be used to explain the result.

(12) In Figure 3, you might also add the plot of computation complexity (iteration number $\times$ number of flops or involved entries per iteration) VS r. This may help us select proper r in practice I think.

(13) In Figure 4, you may also add the plot of the number of iterations/computation complexity VS alpha. Hopefully the iteration number is proportional to $\ln(1/\alpha)$. Also, there is a typo in the caption—“relative”.

(14) You could add a sentence like “We depict the results in Figure 5.” to the final paragraph.

(15) For Figure 5b, is the initial point already close to a local minimum point? (computing gradient norm might be helpful) How about trying more initial points with alpha=1?

**Time Spent Reviewing:**

About 4 hours

---

> ### Author Response · Authors · 2021-08-11
> **Response to Reviewer pNVA**
>
> Thank you for your detailed review and  comments. We have addressed your concerns and respond to your questions below.
>
> ---
> (1) _"What is the power method? You may give more explanation or citations."_
>
>  Thanks for pointing this out. We will add a reference.
>
> ---
>
>  (2)  _"Typo in the last paragraph of Section 1: Use “as long as” in “This holds as long the measurement operator obeys a popular restricted isometry property.”_
>
> Thanks, we will fix this.
>
> ---
> (3) _"In Section 2, you might add application examples of (1) and explain the meanings of $y_i, A_i, XX^T$ in the examples. You may also list some applications in the introduction."_
>
> Thanks for the suggestion. We will discuss several applications in the introduction (matrix completion, quantum state tomography, and covariance sketching).
>
> ---
>  (4) _"What does $\mathcal{A}^*$ mean in the gradient descent equation below eq. (2)?"_
>
> $\mathcal{A}^*$ is the adjoint of the operator $\mathcal{A}$. We will add an explanation to the manuscript.
>
>
>  _"Also, since $X$ is unknown while $y_i$ is known, why not using $y_i$ instead of $X$ in the update rule?"_
>
> That is a great suggestion. We will implement this.
>
> ---
> (5) _"You put the case of_ $r\ge 2 r_*$ _to the main text and_ $r\ge r_{\star}$  _including two special cases into the appendix. Why is the former more important?"_
>
> While we think all three cases are important,  we decided not to include all of them in the main results due to the page limit. However, the case $r\ge 2r_{\star}$, which has higher overparameterization, is maybe best suited to the beyond NTK motivation from our introduction.% motivations more since this includes the overparameterized scenario.
>
> ---
> (6) _"In Theorem 3.3, can the probability $1-Ce^{-\widetilde{c}r}$  get arbitrarily close to 1? Does it rely on $\alpha$?"_
>
> That is a great question. Indeed, we think that by decreasing $\alpha$ we can increase the success probability (and bring it arbitrarily close to $1$). However, to increase the clarity of the presentation, we have not included this in our analysis.
>
> ---
> (7) _"How does eq. (6) relate to the equation between eqs. (4) and (5)? For example, does eq. (6) imply that equation? You might well explain that around eq. (6)."_
>
> Eq. (6) does imply this equation. We will make this more clear in the final version.
>
> ---
>
> (8) _"In Phase II, the $r_{\star}$th eigenvalues will align. How does that relate to saddle avoidance? Also, $\alpha\propto \sigma_{r_*}(U_0) \ll \mathcal{O}[\sigma_{\min}(X)]$ yes? If so, I did not see something like $\alpha\le \mathcal{O}[\sigma_{\min}(X)]$ in eq. (4). Could you explain?"_
>
> The dependence on the condition number $\kappa$ in eq. (4) implies $\alpha\propto \sigma_{r_*}(U_0) \ll \mathcal{O}[\sigma_{\min}(X)]$.
>
> ---
>
> (9) _"What’s the value of $r$ for Figure 2?"_
>
> In Figure 2 we have $r_{\star}=r=1$. We will clarify this in the final version.
>
> ---
> (10) _"You said “We note that while all experimental depictions are based on a single trial, in line with the NeuRIPS guidelines we have drawn these curves multiple times (not depicted) and the behavior of the plots do not change. ” Have you tried multiple initializations of U? Do they have almost overlapping curves?"_
>
> Yes, we have tried several random initializations and the curves are almost overlapping (but of course not exactly the same).
>
> ---
> (11) _"In your paragraph beginning with “Evolution of the test error and the refinement phase”, eq. (6) might also be used to explain the result."_
>
> Thank you for the suggestion. We will add this.
>
> ---
>
> (12) _"In Figure 3, you might also add the plot of computation complexity (iteration number $\times$ number of flops or involved entries per iteration) VS $r$. This may help us select proper r in practice I think."_
>
> Excellent suggestion, we will add this.
>
> ---
>
> (13) _"In Figure 4, you may also add the plot of the number of iterations/computation complexity VS alpha. Hopefully the iteration number is proportional to $\ln(1/\alpha)$ ."_
>
> Great suggestion, we will add this.
>
>  _"Also, there is a typo in the caption—“relative”."_
>
> Thanks, we will fix this.
>
> ---
>
> (14) _"You could add a sentence like “We depict the results in Figure 5.” to the final paragraph."_
>
> Thanks for the suggestion. We will add this to the paragraph.
>
> ---
> (15) _"For Figure 5b, is the initial point already close to a local minimum point? (computing gradient norm might be helpful) How about trying more initial points with $\alpha$=1?"_
>
> Yes, in Figure 5b, the initial point is already close to a local minimum point. In fact, any random initialization will be close to a local minimum point as it has been established in the _lazy training literature_, see [20,29]. We have repeated the experiment with several random initializations and the curves are almost overlapping.

---

### Official Review · Reviewer_SPzT · 2021-07-15

**Rating:** 6
**Confidence:** 4

**Summary:**

This paper studies both the optimization and the generalization of the non-convex overparameterized low-rank matrix reconstruction problem, showing that the small (random) initialization plays an important role. In detail, they divided the dynamic into three phases, in the first phase the eigenspace gradually aligns with the ground truth; the second phase is the saddle avoidance/refinement phase; and the last phase is the local convergence period.

**Limitations And Societal Impact:**

Yes

**Main Review:**

This paper is very solid, extending previous Li et al. [1]'s result in the case that the overparameterized model U is $d*r'$ instead of $d*d$. However, this result is not interesting now. The small initialization has been well studied in different areas (e.g. low rank matrix recovery, tensor decomposition, linear neural network and so on). Also, compared to Li et al. [1]'s result, their result seems to be only a marginal improvement. The detailed comments are provided as following:
1. Why we need to have the first two phases? Is it because some technical reason? For me there is only one phase: the eigenspace aligns to the ground truth and the minimum eigenvalue converges to the ground truth. Also, it is quite confusing for me that why the second phase named "saddle avoidance/refinement phase". I cannot see any intuition why and how GD avoid this from the main body.
2. From Figure 2, the distance measure between the two eigenspaces will decrease first and then increase again. This is very strange to me. My guess is that this comes from the "ground truth" you chosen, $A^*A(XX')$ instead of $XX'$. Will this re-increasing phenomenon effect the proof? Or what if directly measure the distance compared to $XX'$.
3. As you mentioned in line 236-237, the behavior of the plots do not change. Do you running different initialization or choose different Gaussian (symmetric) measurement matrices? For me, it is quite strange that the behaviors with different random seed are exactly the same.
4. It seems this result also needs early stopping. Is there any way to show that the generalization won't blow up again like in [2]?

[1] Yuanzhi Li, Tengyu Ma, and Hongyang Zhang. Algorithmic regularization in over-parameterized matrix sensing and neural networks with quadratic activations. In Conference On Learning Theory, pages 2–47. PMLR, 2018.
[2] Jiacheng Zhuo, Jeongyeol Kwon, Nhat Ho, and Constantine Caramanis. On the computational and statistical complexity of over-parameterized matrix sensing. arXiv preprint arXiv:2102.02756, 2021.

**Time Spent Reviewing:**

24 hours

---

> ### Author Response · Authors · 2021-08-10
> **Response to Reviewer SPzT**
>
> Thank you for your detailed review and comments. We have addressed your concerns and respond to your questions below.
>
> ## Novelty compared to Li et al. [1]
>
> Our main contribution is to establish the intimate connection between small random initialization followed by gradient descent and spectral initialization which does not have a counterpart in [1]. This novel perspective offers a variety of improvements over [1] which we have also discussed in lines 293-301 in the supplementary.
>
> - In our result, the required number of measurements do not depend on the scale of initialization. This is important as it allows to show that the generalization error goes to zero as the scale of initialization goes to zero, i.e. $ \alpha \rightarrow 0 $, with a fixed number of measurements $m$ that does not depend on the scale of initialization $\alpha$. Whereas, in [1] the required number of measurements goes to infinity as the scale of initialization $\alpha$ (and hence the generalization) error goes to zero.
> - Our analysis includes all $r$ satisfying $r_{\star} \le r \le n$ in contrast to $r=n$ required in [1].
> - Perhaps of less import, our results also contain improvements in terms of sample complexity and the step size i.e. requiring fewer samples and allowing for larger learning rates.
>
> We will further highlight these benefits in the final version.
>
> ## Why Phase I and II? Why is Phase II called saddle avoidance phase?
>
> In Phase I the eigenspace does not align with the ground truth. Instead it aligns with the leading eigenvectors of $\mathcal{A}^* \mathcal{A} \left(XX^T\right)$ which is the matrix used in spectral initialization.
>
> In Phase II, the individual eigenvalues may grow with different speeds (especially if the condition number of $X$ is large.) Then the singular value $\sigma_{r_*} \left(U_t\right)$ may still be quite small, whereas $\sigma_1 \left(U_t\right)$ may be already at the order of $ \sigma_1 \left( X \right) $. In this case, we can no longer guarantee that the signal part of $U_t$ is aligned with the eigenspace spanned by the leading eigenvectors of $\mathcal{A}^* \mathcal{A} \left(XX^T\right)$. Hence, we need to differentiate  in the analysis between Phase I and Phase II. Also, the reason Phase II is called saddle avoidance is that for this problem it is known (based on landscape analysis) that saddle points have minimum singular value of zero and thus an increase in the minimum singular value which occurs in this phase corresponds to moving away from such saddles.
>
> ## Why alignment with $\mathcal{A}^*\mathcal{A}(XX^T)$ instead of $XX^T$?
>
> This is a great question. This "re-increasing phenomenon" is exactly what is predicted by our theory. The signal $U_t$ aligns first in the spectral phase with the eigenspace corresponding to the leading eigenvectors of $\mathcal{A}^*\mathcal{A}(XX^T)$. This relates to the spectral initialization, which is used frequently in the non-convex optimization literature. After that, in the refinement phase/local convergence phase, the signal $U_t$ learns the subspace associated with $X$, which is why the angle is increasing.
>
> One can measure the angle to $X$ instead. What one will observe is that the angle goes down, stays constant, and then goes to $0$. We we will include this graph in our figures. In fact, our analysis was in part motivated by this lack of strict monotonic decrease of the angle with $XX^T$, necessitating our analysis which is based on initial alignment with $\mathcal{A}^* \mathcal{A} \left(XX^T\right)$ instead.
>
> ## Re plots do not change with random seeds
>
> We have run the experiments with different random seeds and the behavior of the curves are very similar (almost overlapping but of course not identical). We are not suggesting that they are exactly the same. Instead, we are just pointing out that this behavior is consistent across different random seeds. In the final version of the paper, we will add a section in the supplementary to indeed show that this behavior is consistent across different random seeds.
>
> ## Re early stopping and connection to [2].
>
> We note that our results do not require early stopping when $r=r_*$. We believe it is possible to remove the need for early stopping via a more refined analysis in the local convergence phase and [2] may provide some insights in this regard. However, we note that the results in [2] are not applicable for this local phase as this paper requires on the order of $rn$ samples, i.e. no overparameterization w.r.t. sample complexity, whereas our result allows for $r_{\star}^2 n$ samples, which can be much smaller. We also note that [2] is a local convergence analysis, whereas we focus on a convergence analysis from a random initialization. Due to this difference, these results are not comparable. That said, as mentioned, the results in [2] may provide some insights for our local convergence phase and thus we will add a discussion to this effect. Thanks for the pointer to [2].

---

> > ### Comment · Reviewer_SPzT · 2021-08-31
> > **Thank you for your feedback**
> >
> > Thanks a lot for your feedback! I tend to increase my score from 5 to 6.
> >
> > But still, I suggest you on making a few modifications.
> > 1. small random initialization is similar to spectral initialization, this should be the main point of this paper (you should definitely highlight this.) since other contributions are marginal compared to previous paper, and that's why I intend to increase the score.
> > 2. For the name of second phase. I understand that the singular value of $U_t$ is increasing indicates that it cannot become saddle point. However, it is not necessary the case the distance to a saddle point is becoming larger. Hence, I suggest that choosing another name for this phase.
> > 3. This paper is a long and technically sound paper, it would be great if you can make the motivation much more clear.

---

> > > ### Author Response · Authors · 2021-09-01
> > > **Response**
> > >
> > > Thank you for increasing your score. We briefly address the points you raised.
> > >
> > > 1. We particularly chose our title as we did try to emphasize this. However, we will further highlight it, following your suggestion.
> > > 2. We agree that it is not clear from our proof that the distance increases (although simulations suggest the angular distance does; hence our use of ''move away"). We do not think saddle avoidance implies distance increase. Rather, it implies that in this phase we avoid saddles, in line with the naming. However, we will think about alternative naming to avoid confusion and/or add simulations to verify whether distance also increases.
> > > 3. Thank you for the suggestion. We will work on making the motivation more clear.

---

### Official Review · Reviewer_UAzE · 2021-07-19

**Rating:** 6
**Confidence:** 3

**Summary:**

The authors study the nonconvex approach to the positive semidefinite (PSD)
matrix sensing problem, a much-studied model problem for tractable nonconvex
optimization in the literature. Specifically, they consider the
"overparameterized" version of this problem: there is an unknown ground truth
factor matrix $X \in \mathbb{R}^{n \times r_{\star}}$ which one observes through
linear measurements $\mathcal{A}(X X^T) \in \mathbb{R}^m$; one seeks to find
$X$ by minimizing the nonconvex objective $U \mapsto (1/4) \\| \mathcal{A} (
UU^T - XX^T) \\|^2$ ($\ell^2$ norm) using randomly-initialized gradient descent
from initialization $\alpha U \in \mathbb{R}^{n \times r}$ with $r \geq
r_{\star}$,  where $\alpha > 0$ is a small scale parameter (e.g. inverse
polynomial in $n$) and $U$ is an i.i.d. gaussian matrix with entries of
suitable variance. The authors analyze this problem in the setting where the
measurement operator $\mathcal{A}$ has the RIP (e.g. enough gaussian
measurements) and the general setting of $r \geq r_{\star}$; they prove that
gradient descent recovers the true factor matrix $X$ up to symmetry to any
desired accuracy (measured in frobenius norm), where the precision is improved
by decreasing the magnitude of $\alpha$ (at the cost of a logarithmic gain in
the number of gradient descent iterations required) and with constants that
depend on the condition number of $X$ (Theorem 3.3). The approach to the proof
is claimed as novel, with a connection made between overparameterization in
this model and the common spectral initialization schemes in use, and synthetic
experiments are presented that support the authors' separation of their
approach to the convergence proof into distinct phases, and the predictions of
their main theorem.



**Limitations And Societal Impact:**

Yes.

**Main Review:**

## Summary wrt rating

The paper provides new results on nonconvex gradient descent for PSD matrix
sensing under an RIP property in the setting where the problem is
overparameterized, and the proof techniques seem to reveal novel qualitative
insights into the behavior of the gradient descent algorithm in this
overparameterized setting. Here, I found the simple experiments in Figures 4
and 5 to be quite helpful illustrations. In a similar way, these results also
seem to constitute a valuable contribution to the literature on provable
guarantees for generic optimization methods (randomly initialized gradient
descent) on nonconvex problems. My rating reflects some issues with the paper,
mostly around context wrt the literature and organization. Specifically,
the authors have submitted the "full version" of the paper as the supplementary
material -- I don't see anything wrong with this from a policy perspective, but
they seem to have not prepared the "9 page" version of this work very
carefully, to the extent that there are a number of omissions of highly related
work in both the introduction/motivation and in the presentation of the results
on overparameterized matrix sensing. This has the effect of making it seem like
the authors are unaware of these works, and seems to make it near-impossible to
understand the precise contributions of the authors (wrt generality of their
result; sharpness of the rates; and insights in the proofs) relative to prior
art as someone not actively working on matrix sensing/matrix completion problems. I will discuss these issues in detail below; I would expect to be able to
raise my rating given their satisfactory resolution.

## Context and contributions

It seems to me that the concrete contributions of the authors are: new results
for overparameterized PSD matrix sensing with RIP (global convergence; large
$r$ allowed; possibly new rates); new qualitative insights into the behavior of
randomly-initialized gradient descent on this problem (with small
initialization); a novel contribution to the literature on guarantees for
randomly-initialized gradient methods (i.e. saddle-avoidance-free) on
nonconvex problems. I find it somewhat unclear how most of the motivation in
the introduction connects to these contributions. I also find it challenging to
appreciate these contributions from the current version of the submission due
to omissions of comparisons to relevant prior works. I will give more details
below.

- The paper seems to be motivated from the idea that in *general ML theory
  settings* with overparameterized models, there is a need to understand the
  issues of "small random initialization" and generalization. I find this claim
  incompletely justified here -- for example, there are a number of inverse
  problems in the nonconvex optimization literature in which it is natural to
  formulate the problem with a compact constraint set, where there is no natural
  notion of "small initialization" (e.g. [2-5] below), and one instead uses a
  random initialization or a data-driven initialization; similarly in practical
  deep learning settings it seems like the standard initialization strategies
  (e.g. "He initialization", etc.) lead to matrices with $\Theta(1)$
  operator norms, so it is not clear to me how these are "small". If the
  authors want to motivate their work in this way (e.g. sentence at line 39),
  it would seem appropriate to provide additional justification here.
- The dichotomy outlined in the paragraph from lines 42-61 omits a significant
  body of work on understanding structures in practical problems that enable
  randomly-initialized gradient descent to succeed: for example, [1, 6-8] below
  are relevant here. One could also argue that NTK analyses of deep learning
  problems where generalization is established fall into this category, e.g.
  [9-10] below, although these approaches do not proceed by a geometric
  analysis.
- In a similar vein, I do not understand why the authors have neglected to
  discuss prior art on overparameterized PSD matrix sensing (including [1, 6,
  8] below) in the introduction and related work, given its importance in
  establishing the novelty of the authors' contributions. Footnote 1 seems
  dubious to me in this connection.
- The paragraph from lines 62-71 seems to be only tenuously related to the
  paper's PSD matrix sensing setting. In particular, I had the impression that
  "beyond-NTK" analyses of neural network training in the teacher-student setting were
  more similar to high-order tensor decomposition problems than matrix sensing
  under RIP (e.g. [11]), in particular requiring a specialized initialization
  to work with sharp rates; and in general, these approaches seem to be
  unsuitable for extension to non-shallow neural networks of the kind used in
  practice (relevant to a "guiding principle for practitioners" in line 71).
- In section 3, outside of a discussion of the sample complexity, the authors compare their result only to the cited reference
  [20], which applies to a completely different initialization scheme. The
  authors should compare here to relevant prior works in order to demonstrate
  the novelty of their contributions: i.e. [1] below (this work seems extremely
  relevant, and even offers a precedent for the authors' ideas about "small
  random initialization" with $\alpha$) and [6] below, plus other relevant
  results. I would also think it appropriate, where possible, to compare to the
  proof techniques employed in these works and point out any novelties of the
  authors' approach in this context (e.g. specifically with regards to the
  "small initialization is akin to spectral learning" claim, to demonstrate its
  novelty in the authors' work -- naively, it seems to me that an idea similar to this is employed in [6], and boils down to gradient descent bringing the iterates into the 'basin of attraction' of good solutions without the need for targeted initialization. The same principle is at play in [5, 7] below).

Because of the amount of omitted discussion of relevant works in the PSD matrix
sensing / nonconvex optimization literature, and the lack of convincing
connections between the overparameterized PSD matrix sensing problem and
general ML theory problems (e.g. deep learning theory, "small random
initialization" ubiquity), it would seem to me to be most appropriate to
replace the discussion of these latter issues in the intro with a more detailed
discussion of the former. I also note that there seems to be a lot of
additional space available by reducing figures later in the paper to fit all
panels onto one line, instead of two.


## References mentioned above

[1] https://arxiv.org/abs/1712.09203

[2] http://arxiv.org/abs/1206.5882

[3] https://arxiv.org/abs/1511.03607

[4] https://arxiv.org/abs/1906.02435

[5] https://arxiv.org/abs/1901.00256

[6] https://link.springer.com/article/10.1007/s10107-019-01363-6

[7] https://arxiv.org/abs/1809.10313

[8] https://arxiv.org/abs/2012.15467

[9] https://openreview.net/forum?id=O-6Pm_d_Q-

[10] https://openreview.net/forum?id=fgd7we_uZa6

[11] https://arxiv.org/abs/2102.02410

[12] https://arxiv.org/abs/2006.08857


**Time Spent Reviewing:**

5

---

> ### Author Response · Authors · 2021-08-10
> **Response to Reviewer UAzE**
>
> Thank you for your detailed review and comments. We have addressed your concerns and respond to your questions below.
>
> ## Re summary wrt ratings
>
> Thank you for your positive assessment of our contributions. We are indeed aware of most of the literature you mention and have even discussed some in the supplementary in detail. That said, we agree that is better if more detail on these related work is also included in the 9 pages and we will gladly do so. See further detail below.
>
> ## Re justification for small random initialization
>
> We are not claiming in our introduction that small, random initialization is or should be employed for all non-convex optimization problems.  However, it  has been noted in a number of works (e.g. [29], https://arxiv.org/abs/2002.09277, https://arxiv.org/pdf/2006.13409.pdf) that for many modern machine learning architectures, the scale of initialization is important for the generalization/test behavior. "Large" initialization leads to a linearized/NTK-like regime, whereas stronger generalization performance is typically observed for "smaller" initializations. Therefore developing an analysis for nonconvex problems starting from "small" random initialization is an important contemporary challenge in modern machine learning. This of course does not mean that for every nonconvex problem this is a suitable initialization but reflects the fact that for many contemporary problems this is the most popular training paradigm. We note that "small"/"large" initialization here is used qualitatively and is problem-dependent. Indeed from this perspective, He initialization would be considered "small" compared to NTK like initializations. To further clarify this, we will elaborate these issues further near line 39 to clarify the crucial role of scale of initialization for a large variety of modern machine learning tasks.
>
> ## Re related literature beyond dichotomy and NTK falling into this category
>
> Thank you for pointing out this literature. We have already discussed some of these papers in detail in the related work/supplementary. We do agree that these papers go beyond this dichotomy in some ways and it is good to add a brief discussion of them in the intro which we will gladly do. That said, these papers have certain assumptions (e.g.~[6] requires very particular leave-one-out analysis relying on randomized measurements) or shortcomings (e.g as scale of init goes to zero in error does not go to zero unless an infinite number of samples are used) and therefore in our opinion do not fully go beyond this dichotomy. As mentioned above these issues have been discussed at length in lines 270-279, 293-301 in the supplementary, but we will be adding more detail both in the main body and supplementary per your suggestion. Regarding the NTK analysis of deep learning falling into this category" while we agree that they superficially fall into this category in the sense they also rely on random initialization, we believe that discussion of this literature is more appropriate when discussing the generalization challenge as currently done in lines 62-71. The reason is that in the NTK regime, the model can essentially be treated as linear/convex and hence does not really fall into the nonconvex category. That said, we do agree that [9-10] are quite relevant in the NTK regime and we will add them to our NTK discussion in lines 62-71. We again want to re-emphasize that while NTK analysis does rely on random initialization it only applies when the scale of initialization is small (as measured by a width-dependent quantity). The main focus of this paper is of course to go beyond this "large" initialization regime.
>
> ## Prior art re overparam. matrix sensing and footnote 1
>
> As mentioned, these papers have certain assumptions (e.g. [6] requires a very particular leave-one-out analysis relying on randomized measurements) or shortcomings (e.g as scale of init goes to zero in [1] the error does not go to zero unless an infinite number of samples are used). Since these discussions are more technical in nature we thought it may be more appropriate to include these comparisons after we have presented our own results in detail as we have currently done in the supplementary. That said, we do agree that a brief discussion is merited in the intro and we will be adding more detail both in the main body and supplementary per your suggestion. Specifically, following your suggestion we will add a discussion at the end of Section 1, where will we a) point out improvements over [1] and b) discuss the relationship to [6,8].
>
> **Re Footnote 1**
> To the extent of our knowledge, as worded this footnote is accurate as (1) [6] requires randomized measurements and can not handle measurements obeying RIP and it also focuses on a problem where $r=r^*=1$, (2) the analysis of [1] can not be applied in the non-overparameterized setting, indeed that analysis focuses on the highly overparameterized setting of $r=n$ versus $r=r^*$ which is the point of discussion in this footnote, and (3) [8] does not focus on vanilla gradient descent but a Riemannian gradient descent scheme. That said we agree that this discussion needs to have more detail to avoid confusion. We will remove this footnote and move it Section 3, where it is more appropriate, and elaborate/add further detail.
>
> ## Re tenuous relationship with beyond-NTK analysis and neural network being more related to tensor decomposition problems
>
> We believe the reviewer may have misunderstood the purpose of the discussion in lines 62-71. The purpose of this paragraph is to motivate the important role that "small" random initialization plays in achieving good generalization for a broad family of nonlinear learning problems and to highlight the limitations of analysis techniques (such as NTK) that rely on "large" initialization but achieve subpar generalization. This behavior holds for a broad family of nonlinearities and not just neural networks. Figure 5 clearly shows this behavior holds for the nonlinear learning problem of interest in this paper where the nonlinear function is measurements from a low-rank matrix. Indeed, a very precise analysis of our problem exists in the NTK regime (e.g. see Section 4.2 in arXiv:1812.10004) where the authors show good training behavior but as Figure 5 (b) demonstrates this regime has poor generalization performance. As stated towards the end of the intro the goal of this paper is, as a first step, to address these challenges for a particular nonlinearity which is of interest in low-rank reconstruction. However, as we have also briefly stated, we believe the connection between spectral initialization and small random initialization followed by gradient descent goes far beyond the nonlinearity of interest in low-rank reconstruction. In fact, we completely agree with the reviewer that the teacher-student setting is closely tied to tensor decompositions. However, this does not contradict the view discussed above as we believe in this neural network setting the first few iterations of gradient descent starting from small random initialization behaves akin to spectral learning (however the spectral learning problem in this case is a higher-order tensor decomposition problem instead of the order two tensor (a.k.a.~matrix) decomposition problem that shows up in our paper). From this perspective, [11] mentioned by the reviewer also further supports the relevance of our overall analysis strategy for neural network training as the result of [11] can be viewed as the analysis that can be carried out in the refinement/local convergence phase. Such an analysis when combined with an analysis that shows small random init followed by a few iterates of gradient descent is similar to specialized spectral tensor initialization schemes would yield a full analysis of the generalization behavior of gradient descent starting from small random initialization. We do agree that the current discussion is brief and speculative and we will thus add a more detailed discussion/evidence of this rather intriguing connection.
>
> ## Comparison with [1] in the main text in lieu of supp
>
> We have discussed this relationship in Section 3.3 of the Supplementary. We do agree however that we should also include this discussion in the main text which we originally avoided due to space limitations. The reviewer's suggestion of perhaps making the figures smaller and or remove some whitespace around it may create space to do this.
>
> ## Discuss novelty of proof technique
>
> Great suggestion, we will point to/add a discussion on the novelties of our proof technique. We respectfully disagree that a similar idea to the spectral phase is employed in [6] as this paper does not establish a connection between spectral initialization and the first few iterates of gradient descent which is central to our analysis. Of course, we agree that both papers as well as [5,7] contain analyses that guide the trajectory to the local basin but the way this happens is based on completely different paradigms (e.g. [6] does this by building an auxiliary sequence relying on the randomness of the measurements). We would argue that this is quite natural for any global convergence analysis and does not constitute a novel proof technique. We are certainly not claiming this to be a novelty of our analysis. Rather the novelty is based on the intimate connection to spectral initialization as also evident in the title. We will add further discussion to clarify this.  We also note that the convergence analyses of Riemannian gradient descent [5,7] seem to center around leveraging the geometric properties of the loss function that appears in dictionary learning and perhaps it might be more relevant to discuss them in lines 50-54 as an example of cases where landscape analysis can lead to global convergence analysis. We agree that adding this nuance to that discussion is indeed important. Thanks for pointing us to these.

---

> > ### Comment · Reviewer_UAzE · 2021-08-21
> > **thanks**
> >
> > Dear authors,
> >
> > Thank you for your thorough rebuttal to my review, and for the intriguing comments around tensor decomposition/spectral initialization.
> >
> > I am happy with your proposed revisions and clarifications, and feel they address my complaints -- I will increase my score. Particularly with regards to the connection/motivation from NTK-type methods / DL, I think it will be highly clarifying to have it pointed out that He initialization corresponds to "small" in the picture you are drawing in the introduction -- besides this, (I think this is a minor/semantic point, but) if He initialization is considered "small", I feel like what you are calling "NTK-like" is more accurately described as "lazy regime" in the sense of Chizat Bach and Oyallon (which we can achieve just by scaling the neural network's output); the "kernel" part of NTK seems to suggest kernel-like performance to me in the sense of Jacot et al. (for example, [9] in my last message proves a restricted generalization result when He initialization is used via NTK tools).
> >
> > I do agree with your perception of when it is appropriate to compare to related works on matrix sensing, and must add that the presentation in the full version is appropriately robust. But I feel it is essential to have the "full story" presented in the "short", 9 page version as well; and given that the model studied here is one of overparameterized matrix sensing, I think these results must be mentioned in the main paper. Your proposal for how to execute this sounds appropriate.

---

> > > ### Author Response · Authors · 2021-08-24
> > > **Response**
> > >
> > > Thanks for your nice comments and raising the score!
> > >
> > > We agree that "lazy regime" seems to be more accurate than "NTK" and we will use this terminology in the final version of our paper.
> > >
> > > Are there any other concerns which we can address that would raise your score?

---

### Decision · Program_Chairs · 2021-09-27

**Decision:**

Accept (Poster)

**Comment:**

This paper provides and analyzes a setting where small random initialization has a certain implicit spectral bias.  Reviewers are positive and I recommend acceptance.  That said, the reviewers had many detailed concerns, which led to extensive feedback discussions as below; I request the authors address these points carefully in their revisions.